

**Heterogeneous reactions of mineral dust aerosol: implications for**
**tropospheric oxidation capacity**
Mingjin Tang,[1,*] Xin Huang,[2] Keding Lu,[3] Maofa Ge,[4] Yongjie Li,[5] Peng Cheng,[6] Tong Zhu,[3,*]
Aijun Ding,[2] Yuanhang Zhang,[3] Sasho Gligorovski,[1] Wei Song,[1] Xiang Ding,[1] Xinhui Bi,[1]
Xinming Wang[1,7,*]
[1] State Key Laboratory of Organic Geochemistry and Guangdong Key Laboratory of
Environmental Protection and Resources Utilization, Guangzhou Institute of Geochemistry,
Chinese Academy of Sciences, Guangzhou, China
[2] Joint International Research Laboratory of Atmospheric and Earth System Sciences
(JirLATEST),School of Atmospheric Sciences, Nanjing University, Nanjing, China
[3] State Key Joint Laboratory of Environmental Simulation and Pollution Control, College of
Environmental Sciences and Engineering, Peking University, Beijing, China
[4] Beijing National Laboratory for Molecular Sciences, State Key Laboratory for Structural
Chemistry of Unstable and Stable Species, Institute of Chemistry, Chinese Academy of
Sciences, Beijing, China
[5] Department of Civil and Environmental Engineering, Faculty of Science and Technology,
University of Macau, Avenida da Universidade, Taipa, Macau, China
[6] Institute of Mass Spectrometer and Atmospheric Environment, Jinan University, Guangzhou,
China
[7] Center for Excellence in Regional Atmospheric Environment, Institute of Urban
Environment, Chinese Academy of Sciences, Xiamen 361021, China
Correspondence: Mingjin Tang (mingjintang@gig.ac.cn), Tong Zhu (tzhu@pku.edu.cn),
Xinming Wang (wangxm@gig.ac.cn)





## 26 Abstract

Heterogeneous reactions of mineral dust aerosol with trace gases in the atmosphere could
directly and indirectly affect tropospheric oxidation capacity, in addition to aerosol
composition and physicochemical properties. In this article we provide a comprehensive and
critical review of laboratory studies of heterogeneous uptake of OH, $NO_3$, $O_3$, and their directly
related species as well (including $HO_2$, $H_2O_2$, HCHO, HONO, and $N_2O_5$) by mineral dust
particles. Atmospheric importance of heterogeneous uptake as sinks for these species are
assessed i) by comparing their lifetimes with respect to heterogeneous reactions with mineral
dust to lifetimes with respect to other major loss processes and ii) by discussing relevant field
and modelling studies. We have also outlined major open questions and challenges in
laboratory studies of heterogeneous uptake by mineral dust and discussed research strategies
to address them in order to better understand the effects of heterogeneous reactions with
mineral dust on tropospheric oxidation capacity.



# 1 Introduction

## 1.1 Mineral dust in the atmospheres

Mineral dust, emitted from arid and semi-arid regions with an annual flux of ~2000 Tg per year, is one of the most abundant types of aerosol particles in the troposphere (Textor et al., 2006; Huneeus et al., 2011; Ginoux et al., 2012). After being emitted into the atmosphere, mineral dust aerosol has an average lifetime of a few days in the troposphere and can be transported over several thousand kilometers, thus having important impacts globally (Prospero, 1999; Uno et al., 2009; Huneeus et al., 2011). Mineral dust aerosol has a myriad of significant impacts on atmospheric chemistry and climate. For example, dust aerosol particles can influence the radiative balance of the Earth system directly by scattering and absorbing solar and terrestrial radiation (Balkanski et al., 2007; Jung et al., 2010; Lemaitre et al., 2010; Huang et al., 2015b), and indirectly by serving as cloud condensation nuclei (CCN) to form cloud droplets (Koehler et al., 2009; Kumar et al., 2009; Twohy et al., 2009) and ice nucleation particles (INP) to form ice particles (DeMott et al., 2003; Hoose and Moehler, 2012; Murray et al., 2012; Ladino et al., 2013; DeMott et al., 2015). Mineral dust particles are believed to be the dominant ice nucleation particles in the troposphere (Hoose et al., 2010; Creamean et al., 2013; Cziczo et al., 2013), therefore having a large impact on the radiative balance, precipitation, and the hydrological cycle (Rosenfeld et al., 2001; Lohmann and Feichter, 2005; Rosenfeld et al., 2008). In addition, deposition of mineral dust is a major source for several important nutrient elements (e.g., Fe and P) in remote regions such as open ocean waters and the Amazon (Jickells et al., 2005; Mahowald et al., 2005; Mahowald et al., 2008; Boyd and Ellwood, 2010; Nenes et al., 2011; Shi et al., 2012), strongly affecting several biogeochemical cycles and the climate system of the Earth (Jickells et al., 2005; Mahowald, 2011; Schulz et al., 2012). The impacts of mineral dust aerosol on air quality, atmospheric visibility, and public





health have also been widely documented (Prospero, 1999; Mahowald et al., 2007; Meng and
Lu, 2007; De Longueville et al., 2010; de Longueville et al., 2013; Giannadaki et al., 2014).

It is worthy being emphasized that impacts of mineral dust aerosol on various aspects

of atmospheric chemistry and climate depend on its mineralogy (Journet et al., 2008; Crowley
et al., 2010a; Formenti et al., 2011; Highwood and Ryder, 2014; Jickells et al., 2014; Morman
and Plumlee, 2014; Fitzgerald et al., 2015; Tang et al., 2016a), which shows large geographical
and spatial variability (Claquin et al., 1999; Ta et al., 2003; Zhang et al., 2003; Jeong, 2008;
Nickovic et al., 2012; Scheuvens et al., 2013; Formenti et al., 2014; Journet et al., 2014; Scanza
et al., 2015).

Mineral dust particles can undergo heterogeneous and/or multiphase reactions during

their transport (Dentener et al., 1996; Usher et al., 2003a; Crowley et al., 2010a). These
reactions will modify the composition of dust particles (Matsuki et al., 2005; Ro et al., 2005;
Sullivan et al., 2007; Shi et al., 2008; Li and Shao, 2009; He et al., 2014) and subsequently
change their physicochemical properties, including hygroscopicity, CCN and IN activities
(Krueger et al., 2003b; Sullivan et al., 2009b; Chernoff and Bertram, 2010; Ma et al., 2012;
Tobo et al., 2012; Sihvonen et al., 2014; Wex et al., 2014; Kulkarni et al., 2015), and the
solubility of Fe and P, and etc. (Meskhidze et al., 2005; Vlasenko et al., 2006; Duvall et al.,
2008; Nenes et al., 2011; Shi et al., 2012; Ito and Xu, 2014). The effects of heterogeneous and
multiphase reactions on the hygroscopicity and CCN and IN activities of dust particles have
been comprehensively summarized by a very recent review paper (Tang et al., 2016a), and the
impacts of atmospheric aging processes on the Fe solubility of mineral dust has also been
reviewed (Shi et al., 2012).

Heterogeneous reactions of mineral dust in the troposphere can also remove or produce

a variety of reactive trace gases, directly and/or indirectly modifying the gas phase
compositions of the troposphere and thus changing its oxidation capacity. The global impact



of mineral dust aerosol on tropospheric chemistry through heterogeneous reactions was
proposed in the mid-1990s by a modelling study (Dentener et al., 1996), which motivated many
following laboratory, field, and modelling work (de Reus et al., 2000; Bian and Zender, 2003;
Usher et al., 2003a; Bauer et al., 2004; Crowley et al., 2010a; Zhu et al., 2010; Wang et al.,
2012; Nie et al., 2014). It should be noted that the regional impact of heterogeneous reactions
of mineral dust aerosol was even recognized earlier (Zhang et al., 1994). It has also been
suggested that dust aerosol could indirectly impact tropospheric chemistry by affecting
radiative fluxes and thus photolysis rates (Liao et al., 1999; Bian and Zender, 2003; Jeong and
Sokolik, 2007; Real and Sartelet, 2011).
A few minerals (e.g., $TiO_2$) with higher refractive indices, compared to stratospheric
sulfuric acid particles, have been proposed as potentially suitable materials (Pope et al., 2012;
Tang et al., 2014e; Weisenstein et al., 2015) instead of sulfuric acid and its precursors, to be
delivered into the stratosphere in order to scatter more solar radiation back into space, as one
of solar radiation management methods for climate engineering (Crutzen, 2006).
Heterogeneous uptake of reactive trace gases by minerals is also of interest in this aspect for
assessment of impacts of particle injection on stratospheric chemistry and especially
stratospheric ozone (Pope et al., 2012; Tang et al., 2014e; Tang et al., 2016b). In addition, some
minerals, such as $CaCO_3$ and $TiO_2$, are widely used as raw materials in construction, and their
heterogeneous interactions with reactive trace gases can be important for local outdoor and
indoor air quality (Langridge et al., 2009; Raff et al., 2009; Ammar et al., 2010; Baergen and
Donaldson, 2016; George et al., 2016) and deterioration of construction surfaces (Lipfert, 1989;
Webb et al., 1992; Striegel et al., 2003; Walker et al., 2012).
**1.2 An introduction to heterogeneous kinetics**
The rates of atmospheric heterogeneous reactions are usually described or
approximated as pseudo-first-order reactions. The pseudo-first-order removal rate of a trace





gas (X), $k_I(X)$, due to the heterogeneous reaction with mineral dust, depends on its average
molecular speed, $c(X)$, the surface area concentration of mineral dust aerosol, $S_a$, and the uptake
coefficient, $\gamma$, given by Eq. (1) (Crowley et al., 2010a; Kolb et al., 2010; Ammann et al., 2013;
Tang et al., 2014b):
$$k_I(X) = 0.25 \cdot c(X) \cdot S_A \cdot \gamma \quad (1)$$
The uptake coefficient is the net probability that a molecule X is actually removed from the gas
phase upon collision with the surface, equal to the ratio of number of molecules removed from
the gas phase to the total number of gas-surface collisions (Crowley et al., 2010a).
Heterogeneous reaction of a trace gas (X) will lead to depletion of X close to the surface,
and thus the effective uptake coefficient, $\gamma_{eff}$, will be smaller than the true uptake coefficient, $\gamma$,
as described by Eq. (2) (Crowley et al., 2010a; Davidovits et al., 2011; Tang et al., 2014b):
$$\frac{1}{\gamma_{eff}} = \frac{1}{\gamma} + \frac{1}{\Gamma_{diff}} \quad (2)$$
where $\Gamma_{diff}$ represents the gas phase diffusion limitation. For the uptake onto spherical particles,
Eq. (3) (the Fuchs-Sutugin equation) can be used to calculate $\Gamma_{diff}$ (Tang et al., 2014b; Tang et
al., 2015):
$$\frac{1}{\Gamma_{diff}} = \frac{0.75 + 0.286 Kn}{Kn \cdot (Kn+1)} \quad (3)$$
where $Kn$ is the Knudsen number, given by Eq. (4)
$$Kn = \frac{2\lambda(X)}{d_p} = \frac{6D(X)}{c(X) \cdot d_p} \quad (4)$$
where $\lambda(X)$, $D(X)$ and $d_p$ are the mean free path of X, the gas phase diffusion coefficient of X,
and the particle diameter, respectively. Experimentally measured gas phase diffusion
coefficients of trace gases with atmospheric relevance have been recently compiled and
evaluated (Tang et al., 2014b; Tang et al., 2015); if not available, they can be estimated using
Fuller's semi-empirical method (Fuller et al., 1966; Tang et al., 2015). A new method has also
been proposed to calculate $Kn$ without the knowledge of $D(X)$, given by Eq. (5):





$$Kn = \frac{2}{d_p} \cdot \frac{\lambda_P}{P} \qquad (5)$$
where $P$ is the pressure in atm and $\lambda_P$ is the pressure-normalized mean free path which is equal
to 100 nm·atm (Tang et al., 2015).
**1.3 Scope of this review**

Usher et al. (2003a) provided the first comprehensive review in this field, and

heterogeneous reactions of mineral dust with a myriad of trace gases, including nitrogen oxides,
$SO_2$, $O_3$, and some organic compounds are included. After that, the IUPAC Task Group on
Atmospheric Chemical Kinetic Data Evaluation published the first critical evaluation of kinetic
data for heterogeneous reactions of solid substrates including mineral dust particles (Crowley
et al., 2010a), and kinetic data for heterogeneous uptake of several trace gases (including $O_3$,
$H_2O_2$, $NO_2$, $NO_3$, $HNO_3$, $N_2O_5$, and $SO_2$) onto mineral dust have been recommended. It should
be pointed out that in addition to this and other review articles published by Atmospheric
Chemistry and Physics, the IUPAC task group keeps updating recommended kinetic data
online (http://iupac.pole-ether.fr/). We note that a few other review papers and monographs
have also mentioned atmospheric heterogeneous reactions of mineral dust particles (Cwiertny
et al., 2008; Zhu et al., 2011; Chen et al., 2012; Rubasinghege and Grassian, 2013; Shen et al.,
2013; Burkholder et al., 2015; Ge et al., 2015; George et al., 2015; Akimoto, 2016), in a less
comprehensive manner compared to Usher et al. (2003a) and Crowley et al. (2010). For
example, Cwiertny et al. (2008) reviewed heterogeneous reactions and heterogeneous
photochemical reactions of $O_3$ and $NO_2$ with mineral dust. Atmospheric heterogeneous
photochemistry was summarized by Chen et al. (2012) for $TiO_2$ and by George et al. (2015)
for other minerals. Heterogeneous reactions of mineral dust with a few volatile organic
compounds (VOCs), such as formaldehyde, acetone, methacrolein, methyl vinyl ketone, and
organic acids, have been covered by a review article on heterogeneous reactions of VOCs (Shen
et al., 2013). The NASA-JPL data evaluation panel has compiled and evaluated kinetic data for





heterogeneous reactions with alumina (Burkholder et al., 2015). In a very recent paper, Ge et
al. (2015) summarized previous studies on heterogeneous reactions of mineral dust with $NO_2$,
$SO_2$, and monocarboxylic acids, with work conducted by scientists in China emphasized. In his
monograph entitled Atmospheric Reaction Chemistry, Akimoto (2015) briefly discussed some
heterogeneous reactions of mineral dust particles in the troposphere. Roles heterogeneous
chemistry of aerosol particles (including mineral dust) play in haze formation in China were
outlined (Zhu et al., 2011), and effects of surface adsorbed water and thus relative humidity
(RH) on heterogeneous reactions of mineral dust have also been discussed by a recent feature
article (Rubasinghege and Grassian, 2013).

After the publication of the two benchmark review articles (Usher et al., 2003a;

Crowley et al., 2010a), much advancement has been made in this field. For example,
heterogeneous uptake of $HO_2$ radicals by mineral dust particles had not been explored at the
time when Crowley et al. (2010a) published the IUPAC evaluation, and in the last few years
this reaction has been investigated by two groups (Bedjanian et al., 2013b; Matthews et al.,
2014). A large number of new studies on the heterogeneous reactions of mineral dust with $H_2O_2$
(Wang et al., 2011; Zhao et al., 2011b; Romanias et al., 2012b; Yi et al., 2012; Zhou et al.,
2012; Romanias et al., 2013; Zhao et al., 2013; El Zein et al., 2014; Zhou et al., 2016) and $N_2O_5$
(Tang et al., 2012; Tang et al., 2014a; Tang et al., 2014c; Tang et al., 2014e) have emerged.
Therefore, a review on atmospheric heterogeneous reaction of mineral dust is both timely and
necessary.

Furthermore, the novelty of our current review, which distinguishes it from previous

reviews in the same/similar fields (Usher et al., 2003a; Cwiertny et al., 2008; Crowley et al.,
2010a; Zhu et al., 2011; Chen et al., 2012; Shen et al., 2013; Ge et al., 2015; George et al.,
2015), is the fact that atmospheric relevance and significance of laboratory studies are
illustrated, discussed, and emphasized. We hope that this paper will be useful not only for those





whose expertise is laboratory work but also for experts in field measurements and atmospheric
modelling. The following approaches are used to achieve this goal: 1) lifetimes of reactive trace
gases with respect to heterogeneous uptake by mineral dust, calculated using preferred uptake
coefficients and typical mineral dust mass concentrations, are compared to their lifetimes in
the troposphere (discussed in Section 2.1) in order to discuss the significance of heterogeneous
reactions as atmospheric sinks for these trace gases; 2) atmospheric importance of these
heterogeneous reactions are further discussed by referring to representative box, regional, and
global modelling studies reported previously; 3) we also describe two of the largest challenges
in the laboratory studies of heterogeneous reactions of mineral dust particles (Section 2.2), and
explain why reported uptake coefficients show large variability and how we interpret and use
these kinetic data. In fact, the major expertise of a few coauthors of this review paper is field
measurements and/or modelling studies, and their contribution should largely increase the
readability of this paper for the entire atmospheric chemistry community regardless of the
academic background of individual readers.

OH, $NO_3$, and $O_3$ are the most important gas phase oxidants in the troposphere, and

their contribution to tropospheric oxidation capacity has been well recognized (Brown and
Stutz, 2012; Stone et al., 2012). $HO_2$ radicals are closely linked with OH radicals (Stone et al.,
2012). $H_2O_2$, HCHO and HONO are important precursors for OH radicals in the troposphere
(Stone et al., 2012), and they may also be important oxidants in the aqueous phase (Seinfeld
and Pandis, 2006). Tropospheric $N_2O_5$ is found to be in dynamic equilibrium with $NO_3$ radicals
(Brown and Stutz, 2012). Therefore, in order to provide a comprehensive view of implications
of heterogeneous reactions of mineral dust particles for tropospheric oxidation capacity, not
only heterogeneous uptake of OH, $NO_3$, and $O_3$ but also heterogeneous reactions of $HO_2$, $H_2O_2$,
HCHO, HONO, and $N_2O_5$ are included. Cl atoms (Spicer et al., 1998; Osthoff et al., 2008;
Thornton et al., 2010; Phillips et al., 2012; Liao et al., 2014; Wang et al., 2016) and stable





Criegee radicals (Mauldin III et al., 2012; Welz et al., 2012; Percival et al., 2013; Taatjes et al.,
2013) are proposed to be potentially important oxidants in the troposphere, thought their
atmospheric significance is to be systematically assessed (Percival et al., 2013; Taatjes et al.,
2014; Simpson et al., 2015). In addition, their heterogeneous reactions with mineral dust have
seldom been explored. Therefore, heterogeneous uptake of Cl atoms (and their precursors such
as $ClNO_2$) and stable Criegee radicals by mineral dust is not included here.

In Section 2, a brief introduction to tropospheric chemistry of OH, $HO_2$, $H_2O_2$, $O_3$,

HCHO, HONO, $NO_3$, and $N_2O_5$ (8 species in total) is provided first. After that, we describe
two major challenges in laboratory studies of heterogeneous reactions of mineral dust particles,
and then discuss their implications in reporting and interpreting kinetic data. Following this in
Section 3, we review previous laboratory studies of heterogeneous reactions of mineral dust
particles with these eight reactive trace gases. Uncertainties for each individual reactions are
discussed, and future work required to reduce these uncertainties is suggested. In addition,
atmospheric importance of these reactions is discussed by 1) comparing their lifetimes with
respect to heterogeneous uptake to typical lifetimes in the troposphere and 2) discussing
representative modelling studies at various spatial and temporal scales. Finally in Section 4 we
outline key challenges which preclude better understanding of impacts of heterogeneous
reactions of mineral dust on tropospheric oxidation capacity and discuss how they can be
addressed by future work.
**2 Background**

In first part of this section we provide a brief introduction of production and removal

pathways, chemistry, and lifetimes of OH, $HO_2$, $H_2O_2$, $O_3$, HCHO, HONO, $NO_3$, and $N_2O_5$ in
the troposphere. In the second part we describe two of the largest challenges in laboratory
investigation of heterogeneous reactions of mineral dust particles and discuss their implications
for reporting, interpreting, and using uptake coefficients.



### 2.1 Sources and sinks of tropospheric oxidants


Figure 1 shows a simplified schematic diagram of atmospheric chemistry of major free

radicals in the troposphere. Sources, sinks, and atmospheric lifetimes of these radicals and their
important precursors are discussed below.

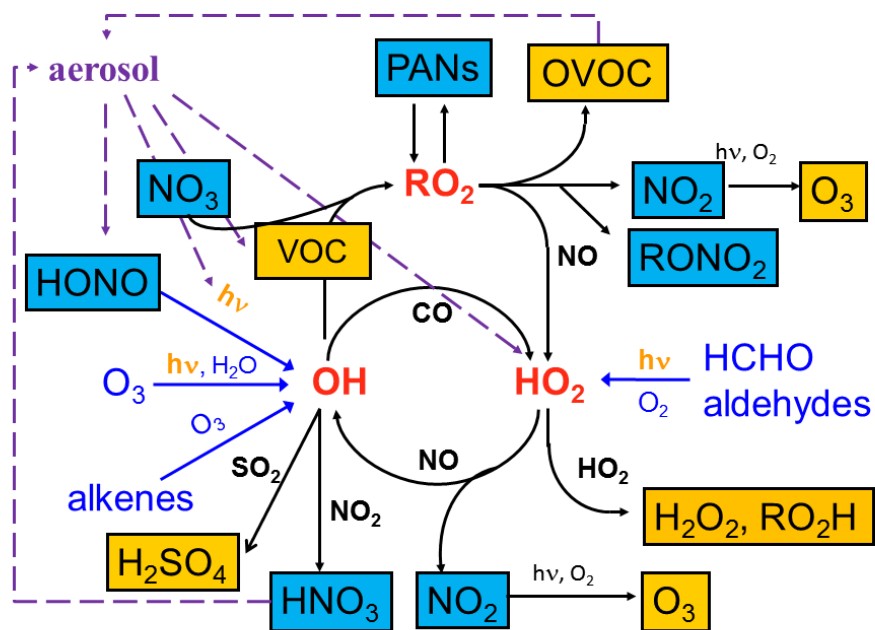


**Figure 1.** Simplified schematic diagram of chemistry of major free radicals in the troposphere.

### 2.1.1 OH, HO$_2$, and H$_2$O$_2$

Large amounts of OH ($10^6$ - $10^7$ molecule cm$^{-3}$) and HO$_2$ radicals ($10^8$ - $10^9$ cm$^{-3}$) have

been observed and predicted for the lower troposphere (Stone et al., 2012). The first major
primary source of OH radicals in the troposphere is the reaction of water vapor with O($^1$D)
(R1), which is produced from photolysis of O$_3$ by UV radiation with wavelengths smaller than
325 nm (R2) (Atkinson et al., 2004; Burkholder et al., 2015):

O($^1$D) + H$_2$O → OH + OH (R1)





$$O_3 + h\nu \ (\lambda < 325 \ \text{nm}) \rightarrow O_2 + O(^1D) \quad (R2)$$
In polluted urban areas, another two primary sources of OH and $HO_2$ radicals, i.e. photolysis
of HONO and HCHO, become significant (Seinfeld and Pandis, 2006) and sometimes even
dominate the primary production of OH (Su et al., 2008):
$$HONO + h\nu \ (\lambda < 400 \ \text{nm}) \rightarrow NO + OH \quad (R3)$$
$$HCHO + h\nu \ (\lambda < 340 \ \text{nm}) \rightarrow H + HCO \quad (R4a)$$
$$H + O_2 + M \rightarrow HO_2 + M \quad (R4b)$$
$$HCO + O_2 \rightarrow HO_2 + CO \quad (R4c)$$
Photolysis of higher oxygenated volatile organic compounds (OVOCs) such as di-carbonyl
compounds has also been suggested as important primary sources for HOx radicals in
megacities in China (Lu et al., 2012; Lu et al., 2013) and Mexico (Dusanter et al., 2009). Under
twilight conditions as well as during winter time, ozonolysis of alkenes and photolysis of
OVOCs have been found to be dominant primary sources of OH and $HO_2$ (Geyer et al., 2003;
Heard et al., 2004; Kanaya et al., 2007b; Edwards et al., 2014; Lu et al., 2014).
After initiated by primary production channels described above, OH radicals further
react with volatile organic compounds (VOCs) to generate organic peroxy radicals ($RO_2$). $RO_2$
radicals are then converted to $HO_2$ radicals by reacting with NO (R5) and the produced $HO_2$
radicals are finally recycled back to OH via reaction with NO (R6).
$$RO_2 + NO \rightarrow HO_2 + NO_2 \quad (R5)$$
$$HO_2 + NO \rightarrow OH + NO_2 \quad (R6)$$
Due to these chain reactions, ambient OH levels are sustained and emitted reductive trace gas
compounds (e.g., VOCs and NO) are catalytically oxidized (Seinfeld and Pandis, 2006). These
chain reactions are terminated by reaction of OH with $NO_2$ (R7, in which M is the third-body
molecule) at high NOx conditions and by cross reaction of $HO_2$ with $RO_2$ and self-reaction of
$HO_2$ radicals (R8) at low NOx conditions.



$$OH + NO_2 + M \rightarrow HNO_3 + M \quad (R7)$$
$$HO_2 + HO_2 \rightarrow H_2O_2 + O_2 \quad (R8)$$
In recent years, a new OH regeneration mechanism, which has not been completely elucidated
so far, has been identified for low NOx environments including both forested (Lelieveld et al.,
2008) and rural areas (Hofzumahaus et al., 2009; Lu et al., 2012). This new mechanism is found
to stabilize the observed OH-$j$(O$^1$D) relationships and enables a type of maximum efficiency
of OH sustainment under low NOx conditions (Rohrer et al., 2014).
Table 1 summarizes representative lifetimes of OH and HO$_2$ radicals in the troposphere
as determined by previous field campaigns. The OH lifetime is an important parameter to
characterize HOx chemistry as well as VOC reactivity in the troposphere. As a result, it has
been widely measured at different locations using a variety of experimental methods (Sinha et
al., 2008; Ingham et al., 2009), as discussed by a very recent paper (Yang et al., 2016b). OH
lifetimes in clean environments, like open ocean and remote continental areas, are dominated
by reactions with CO, CH$_4$, and HCHO, summed up to values of about 0.5-1 s (Ehhalt, 1999;
Brauers et al., 2001). OH lifetimes in forested areas, mainly contributed by oxidation of
biogenic VOCs, are typically in the range of 0.01-0.05 s (Ingham et al., 2009; Nölscher et al.,
2012). In urban areas, OH lifetimes are determined by anthropogenically emitted hydrocarbons,
NOx, CO, and biogenic VOCs as well, and they are typically smaller than 0.1 s (Ren et al.,
2003; Mao et al., 2010b; Lu et al., 2013).
Compared to OH radicals, lifetimes of HO$_2$ radicals have been much less investigated
and are mainly determined by ambient NO concentrations when NO is larger than 10 pptv
(parts per trillion by volume). Therefore, the lower limit of HO$_2$ lifetimes, on the order of 0.1 s,
often appear in polluted urban areas (Ren et al., 2003; Kanaya et al., 2007a; Lu et al., 2012).
The upper limit of HO$_2$ lifetimes, up to 1000-2000 s, is often observed in clean regions and
sometimes also in urban areas during nighttime (Holland et al., 2003; Lelieveld et al., 2008;



Whalley et al., 2011). In addition, heterogeneous uptake of $HO_2$ radicals have been frequently
considered in the budget analysis of HOx radicals for marine and polluted urban regions
(Abbatt et al., 2012).

**Table 1.** Summary of typical lifetimes of OH, $HO_2$, $NO_3$ and $N_2O_5$ in the troposphere reported
by field measurements.

| Time | Location | Lifetimes | Reference |
|---|---|---|---|
| OH radicals | | | |
| OCT-NOV 1996 | Tropical Atlantic Ocean | 1 s | (Brauers et al., 2001) |
| AUG 1994 | Mecklenburg Vorpommern, Germany | 0.5 s | (Ehhalt, 1999) |
| JUL-AUG 1998 | Pabstthum (rural Berlin), Germany | 0.15-0.5 s | (Mihelcic et al., 2003) |
| AUG-SEP 2000 | Houston, US | 0.08-0.15 s | (Mao et al., 2010b) |
| JUN-AUG 2001 | New York, US | 0.04-0.06 s | (Ren et al., 2003) |
| AUG 2007 | Tokyo, Japan | 0.01-0.1 s | (Chatani et al., 2009) |
| JUL 2006 | Backgarden (rural Guangzhou), China | 0.008-0.1 s | (Lou et al., 2010) |
| AUG 2006 | Yufa (rural Beijing), China | 0.01-0.1s | (Lu et al., 2013) |
| APR-MAY 2008 | Borneo, Malaysia | 0.015-0.1 s | (Ingham et al., 2009) |
| JUL-AUG 2010 | Hyytiala, Finland | 0.01-0.5 s | (Nölscher et al., 2012) |
| $HO_2$ radicals | | | |
| JUL-AUG 1998 | Pabstthum (rural Berlin), Germany | 3-500 s | (Holland et al., 2003) |
| JUN-AUG 2001 | New York, US | 0.1-1.5 s | (Ren et al., 2003) |
| JUL-AUG 2004 | Tokyo, Japan | 0.05-1000 s | (Kanaya et al., 2007a) |
| JUL 2006 | Backgarden (rural Guangzhou), China | 0.1-500 s | (Lu et al., 2012) |
| AUG 2006 | Yufa (rural Beijing), China | 0.06-500 s | (Lu et al., 2013) |
| OCT 2005 | Suriname | 500-1000 s | (Lelieveld et al., 2008) |
| APR-MAY 2008 | Borneo, Malaysia | 20-2000 s | (Whalley et al., 2011) |



| | | NO₃ radicals | |
|---|---|---|---|
| OCT 1996 | Helgoland, Germany | 10-1000 s | (Martinez et al., 2000) |
| JUL-AUG 1998 | Berlin, Germany | 10-500 s | (Geyer et al., 2001) |
| JUL-AUG 2002 | US east coast | typically a few min, up to 20 min | (Aldener et al., 2006) |
| MAY 2008 | Klein Feldberg, Germany | up to ~1500 s | (Crowley et al., 2010b) |
| AUG-SEP 2011 | Klein Feldberg, Germany | up to 1 h, with an average value of ~200 s | (Sobanski et al., 2016) |
| | | $N_2O_5$ | |
| OCT 1996 | Helgoland, Germany | hundred to thousand seconds | (Martinez et al., 2000) |
| JAN 2004 | Contra Costa, California, US | 600-1800 s | (Wood et al., 2005) |
| JUL-AUG 2002 | US east coast | up to 60 min | (Aldener et al., 2006) |
| NOV 2009 | Fairbank, Alaska, US | ~6 min on average | (Huff et al., 2011) |
| NOV-DEC 2013 | Hong Kong, China | from <0.1 h to 13 h | (Brown et al., 2016) |


Formation and removal of gas phase $H_2O_2$ in the troposphere is closely linked with the
HOx radical chemistry. Tropospheric $H_2O_2$ is mainly produced from self-reaction of $HO_2$
radicals (R8) and this process is further enhanced by the presence of water vapor (Stockwell,
1995). In addition to dry and wet deposition, another two pathways, i.e. photolysis (R9) and
the reaction with OH (R10), dominate the removal of $H_2O_2$ in the troposphere:

$H_2O_2 + hv\ (\lambda < 360\ nm) \rightarrow OH + OH$    (R9)

$H_2O_2 + OH \rightarrow H_2O + HO_2$    (R10)

Typical $J(H_2O_2)$ daily maximum values are ~$7.7 \times 10^{-6}$ s$^{-1}$ for solar zenith angle of 0°
and ~$6.0 \times 10^{-6}$ s$^{-1}$ in the northern mid-latitude (Stockwell et al., 1997), corresponding to
$\tau_{phot}(H_2O_2)$ ($H_2O_2$ lifetimes with respect to photolysis) of 33-56 h (or 1.5-2 days). The rate
constant for the bimolecular reaction of $H_2O_2$ with OH radicals is $1.7 \times 10^{-12}$ cm$^3$ molecule$^{-1}$ s$^{-1}$
at room temperature, and its temperature dependence is quite small (Atkinson et al., 2004).



Concentrations of OH radicals in the troposphere are usually in the range of $(1\text{-}10)\times10^6$
molecule $cm^{-3}$, and thus $\tau_{OH}(H_2O_2)$ ($H_2O_2$ lifetimes with respect to reaction with OH radicals)
are estimated to be around 16-160 h. Dry deposition rates of $H_2O_2$ were determined to be ~5
$cm\ s^{-1}$ (Hall and Claiborn, 1997), and an assumed boundary height of 1 km gives $\tau_{dry}(H_2O_2)$
($H_2O_2$ lifetimes with respect to dry deposition) of 5-6 h. Therefore, dry deposition is a major
sink for near surface $H_2O_2$. We do not estimate $H_2O_2$ lifetimes with respect to wet deposition
because wet deposition rates depend on amount of precipitation which shows large spatial and
temporal variation. Heterogeneous uptake of $H_2O_2$ by ambient aerosols as well as fog and rain
droplets is also considered to be a significant sink for $H_2O_2$, especially when the ambient $SO_2$
concentrations are high (de Reus et al., 2005; Hua et al., 2008).

As mentioned previously, HONO and HCHO are two important precursors for OH

radicals, and therefore their removal (as well as production) significantly affects tropospheric
oxidation capacity. The typical $J$(HONO) daily maximum value for the northern mid-latitude
is ~$1.63\times10^{-3}\ s^{-1}$ (Stockwell et al., 1997), corresponding to $\tau_{phot}$(HONO) of about 10 min. This
is supported by field measurements which suggest that lifetimes of HONO due to photolysis
during the daytime are typically in the range of 10-20 min (Alicke et al., 2003; Li et al., 2012).
The second order rate constant for the reaction of HONO with OH radicals is $6.0\times10^{-12}\ cm^3$
$molecule^{-1}\ s^{-1}$ at 298 K (Atkinson et al., 2004), giving $\tau_{OH}$(HONO) of ~280 min (~4.6 h) if OH
concentration is assumed to be $1\times10^7$ molecule $cm^{-3}$. Dry deposition velocities of HONO
reported by previous work show large variability, ranging from 0.077 to 3 $cm\ s^{-1}$ (Harrison and
Kitto, 1994; Harrison et al., 1996; Stutz et al., 2002), and thus $\tau_{dry}$(HONO) are estimated to be
in the range of ~9 h to several days if a boundary height of 1 km is assumed. Therefore,
photolysis is the main sink for HONO in the troposphere and the contribution from dry
deposition and reaction with OH is quite minor.



The second order rate constant for the reaction of HCHO with OH radicals is $8.5 \times 10^{-12}$
$cm^3$ $molecule^{-1}$ $s^{-1}$ at 298 K (Atkinson et al., 2006), and $\tau_{OH}(HCHO)$ is calculated to be ~200
min (~3.3 h) if OH concentration is assumed to be $1 \times 10^7$ molecule $cm^{-3}$. The typical $J$(HCHO)
daily maximum value for the northern mid-latitude is $\sim 5.67 \times 10^{-5}$ $s^{-1}$ (Stockwell et al., 1997),
giving $\tau_{phot}(HCHO)$ of about 300 min (~5 h). The dry deposition velocity for HCHO was
measured to be 1.4 cm $s^{-1}$ (Seyfioglu et al., 2006), corresponding to $\tau_{dry}(HCHO)$ of ~20 h if the
boundary layer height is assumed to be 1 km. To summarize, lifetimes of HCHO in the
troposphere are estimated to be a few hours, with photolysis and reaction with OH radicals
being major sinks.
**2.1.2 O$_3$**
After being emitted, NO is converted to $NO_2$ in the troposphere through its reactions
with $O_3$ (R11) and peroxy radicals (R5, R6). $NO_2$ is further photolyzed to generate $O_3$ (R12),
and NO oxidation processes through R5 and R6 are the reason for $O_3$ increase in the
troposphere (Wang and Jacob, 1998).
$$O_3 + NO \rightarrow NO_2 + O_2 \quad (R11)$$
$$NO_2 + O_2 + h\nu \,(\lambda < 420 \text{ nm}) \rightarrow O_3 + NO \quad (R12)$$
Tropospheric $O_3$ is mainly destroyed via its photolysis (R1) and the subsequent reaction of $O^1D$
with $H_2O$ (R2). Other important removal pathways include dry deposition, reaction with $NO_2$
(to produce $NO_3$ radicals) (R13), and ozonolysis of alkenes, etc.
$$NO_2 + O_3 \rightarrow NO_3 + O_2 \quad (R13)$$
In addition, the loss of $NO_2$ through reaction with OH (R7) and the loss of peroxy radicals
through their self-reactions (R8) would be a significant term of $O_3$ losses in large scales.
Therefore, it is anticipated that both the formation and destruction of $O_3$ is closely related with
gas phase HOx and NOx radical chemistry.



Several processes remove $O_3$ from the troposphere. The first one is the photolysis of $O_3$
to produce $O^1D$ (R1) and the subsequent reaction of $O^1D$ with $H_2O$ (R2); therefore, the removal
rate of $O_3$ through this pathway depends on solar radiation and RH. $\tau_{pho}(O_3)$ is typically in the
range of 1.8-10 days in the troposphere (Stockwell et al., 1997). Ozonolysis of alkenes is
another significant sink for $O_3$ under high VOCs conditions, and $\tau_{alkene}(O_3)$ with respect to
reaction with alkenes are estimated to be 3-8 h for urban and forested areas (Shirley et al., 2006;
Kanaya et al., 2007b; Whalley et al., 2011; Lu et al., 2013; Lu et al., 2014). $O_3$ lifetimes in the
remote troposphere are primarily determined by $O_3$ photolysis (and the subsequent reaction of
$O^1D$ with $H_2O$) and reactions of $O_3$ with $HO_2$ and OH. For typical conditions ($j(O^1D)$, $H_2O$,
$HO_2$, OH, temperature, and pressure) over northern mid-latitude oceans, $O_3$ lifetimes are
calculated to be a few days in summer, 1-2 weeks in spring/autumn, and about a month for in
winter, using the GEOS-Chem model (to be published). $O_3$ dry deposition has been extensively
studied and as a rule of thumb, 1 cm s$^{-1}$ is taken as its dry deposition rate (Wesely and Hicks,
2000). Consequently, $\tau_{dry}(O_3)$ is calculate to be ~28 h, assuming a boundary height of 1 km.
Reactions with NO and $NO_2$ will further contribute to the removal of $O_3$ in the troposphere at
night. The second-order rate constants are $1.9 \times 10^{-14}$ cm$^3$ molecule$^{-1}$ s$^{-1}$ for the reaction of $O_3$
with NO and $3.5 \times 10^{-17}$ cm$^3$ molecule$^{-1}$ s$^{-1}$ for its reaction with $NO_2$ at 298 K (Atkinson et al.,
2004), and $O_3$ lifetimes are calculated to be ~29 h and ~32 h in the presence of 20 pptv NO and
10 ppbv (parts per billion by volume) $NO_2$, respectively.
Moreover, heterogeneous processes may also strongly influence the budget of $O_3$
through impacts on sources and sinks of HOx and NOx (Dentener et al., 1996; Jacob, 2000;
Zhu et al., 2010), the production of halogen radicals (Thornton et al., 2010; Phillips et al., 2012;
Wang et al., 2016), and possibly also direct removal of $O_3$ due to heterogeneous uptake (de
Reus et al., 2000).




**2.1.3 NO$_3$ radicals (and N$_2$O$_5$)**


Oxidation of NO$_2$ by O$_3$ (R13) is the dominant source for NO$_3$ radicals in the
troposphere. NO$_3$ radicals further react with NO$_2$ to form N$_2$O$_5$ (R14), which can thermally
dissociate back to NO$_3$ and NO$_2$ (R15) (Wayne et al., 1991; Brown and Stutz, 2012).
$$NO_2 + O_3 \rightarrow NO_3 + O_2 \quad (R13)$$
$$NO_2 + NO_3 + M \rightarrow N_2O_5 + M \ (R14)$$
$$N_2O_5 + M \rightarrow NO_2 + NO_3 + M \ (R15)$$
The equilibrium between NO$_3$ and N$_2$O$_5$ is usually reached within several seconds under typical
tropospheric conditions. Therefore, NO$_3$ radicals are considered to be in dynamic equilibrium
with N$_2$O$_5$, as confirmed by a number of field measurements (Brown and Stutz, 2012, and
references therein). As a results, NO$_3$ and N$_2$O$_5$ are discussed together here. Recently reactions
of Criegee radicals with NO$_2$ are proposed as another source for NO$_3$ radicals (Ouyang et al.,
2013), though atmospheric significance of this source has not been systematically assessed yet
(Sobanski et al., 2016).
Photolysis of NO$_3$ (R17) and its reaction with NO (R16) are both very fast (Wayne et
al., 1991), and atmospheric chemistry of NO$_3$ (and thus N$_2$O$_5$) is only important during
nighttime, though the daytime presence of NO$_3$ and N$_2$O$_5$ in the troposphere has also been
reported (Brown and Stutz, 2012). Therefore, for a sink to be important for NO$_3$ or N$_2$O$_5$, the
lifetime with respect to this sink should be comparable to or shorter than half one day.
$$NO_3 + NO \rightarrow NO_2 + NO_2 \ (R16)$$
$$NO_3 + (\lambda < 11080 \text{ nm}) \rightarrow NO + O_2 \ (R17a)$$
$$NO_3 + (\lambda < 587 \text{ nm}) \rightarrow NO_2 + O \ (R17b)$$
The predominant sinks for tropospheric NO$_3$ and N$_2$O$_5$ include reactions with
unsaturated volatile organic compounds (VOCs), reaction with dimethyl sulfite in the marine
and coastal troposphere, and heterogeneous uptake by aerosol particles and cloud droplets



(Brown and Stutz, 2012). The gas phase reaction of $N_2O_5$ with water vapor was investigated
by a laboratory study (Wahner et al., 1998), and several field measurements have suggested
that this reaction is unlikely to be significant in the troposphere (Brown et al., 2009; Crowley
et al., 2010b; Brown and Stutz, 2012). Lifetimes of $NO_3$ and $N_2O_5$ during nighttime depend on
a variety of atmospheric conditions (including concentrations of VOCs and aerosols, aerosol
composition and mixing state, and RH etc.) (Brown and Stutz, 2012), exhibiting large spatial
and temporal variations. As shown in Table 1, $NO_3$ lifetimes typically range from tens of
seconds to 1 h, while $N_2O_5$ lifetimes are usually longer, spanning from <10 min to several hours.
**2.2 Laboratory studies of atmospheric heterogeneous reactions of mineral dust**
**particles**
Kinetics of heterogeneous reactions can be determined by measuring the decay and/or
production rates of trace gases in the gas phase (Hanisch and Crowley, 2001; Usher et al.,
2003b; Liu et al., 2008a; Vlasenko et al., 2009; Pradhan et al., 2010a; Tang et al., 2012; Zhou
et al., 2014). Alternatively, reaction rates can also be measured by detecting changes in particle
composition (Goodman et al., 2000; Sullivan et al., 2009a; Li et al., 2010; Tong et al., 2010;
Ma et al., 2012; Kong et al., 2014). A number of experimental techniques have been developed
and utilized to investigate heterogeneous reactions of mineral dust particles, as summarized in
Table 1. It should be emphasized that this list is far from being complete and only techniques
mentioned in this review paper are included. These techniques can be classified into three
groups according to the way particles under investigation exist: 1) particle ensembles deposited
on a substrate, 2) an ensemble of particles as an aerosol, and 3) single particles, either levitated
or deposited on a substrate. Detailed description of these techniques can be found in several
previous review articles and monographies (Usher et al., 2003a; Cwiertny et al., 2008; Crowley
et al., 2010a; Kolb et al., 2010; Akimoto, 2016) and thus is not repeated here. Instead, in this
paper we intend to discuss two critical issues in determining and reporting uptake coefficients





for heterogeneous reactions of mineral dust particles, i.e. 1) surface area available for
heterogeneous uptake and 2) time dependence of heterogeneous kinetics.

For experiments in which single particles are used, usually surface techniques,

including Raman spectroscopy (Liu et al., 2008b; Zhao et al., 2011a), scanning electron
microscopy (SEM) (Krueger et al., 2003a; Laskin et al., 2005b), and secondary ion mass
spectroscopy (SIMS) (Harris et al., 2012), can be utilized to characterize their compositional
and morphological changes simultaneously. Nevertheless, it is still non-trivial to derive
quantitative information for most of surface techniques. In addition to being deposited on a
substrate, single particles can also be levitated by an electrodynamic balance (Lee and Chan,
2007; Pope et al., 2010) or optical levitation (Tong et al., 2011; Krieger et al., 2012; Rkiouak
et al., 2014), and Raman spectroscopy can be used to measure the compositional changes of
levitated particles (Lee et al., 2008; Tang et al., 2014a).

**Table 2:** Abbreviations of experimental techniques used by previous laboratory studies to
investigate heterogeneous reactions of mineral dust. Only techniques mentioned in this review
paper are included.

| abbreviation | full name |
| --- | --- |
| AFT | aerosol flow tube |
| CIMS | chemical ionization mass spectrometry |
| CLD | chemiluminescence detector |
| CRDS | cavity ring-down spectroscopy |
| CRFT | coated rod flow tube |
| CWFT | coated wall flow tube |
| DRIFTS | diffuse reflectance infrared Fourier transform spectroscopy |
| EC | environmental chamber |
| KC | Knudsen cell reactor |
| IC | ion chromatography |
| LIF | laser induced fluorescence |
| MS | mass spectrometry |





| T-FTIR | transmission FTIR |
|---|---|


### 2.2.1 Surface area available for heterogeneous uptake

As described by Eq. (1), surface area concentration is required to derive uptake

coefficients from measured pseudo first order reaction rates. However, it can be a difficult task
to obtain surface area concentrations for particles. In fact, variation in estimated surface area
available for heterogeneous uptake is one of the main reasons why large differences in uptake
coefficients have been reported by different groups for the same reaction system of interest.

For experiments in which aerosol particles are used, surface area concentrations are

typically derived from size distribution measured using an aerodynamic particle sizer (APS) or
scanning mobility particle sizer (SMPS). Because of the non-sphericity of mineral dust
particles, it is not straightforward to convert aerodynamic and mobility diameters to surface
area. In some aerosol chamber studies, surface areas available for heterogeneous uptake are
assumed to be equal to the BET surface areas of dust particles introduced into the chamber
(Mogili et al., 2006b; Mogili et al., 2006a; Chen et al., 2011b). Some dust particles are porous,
making their BET surface areas much larger than the corresponding geometrical surface areas.
$\gamma(N_2O_5)$ for airborne $SiO_2$ particles reported by two previous studies (Mogili et al., 2006b;
Wagner et al., 2009) differed by almost two orders of magnitude. Tang et al. (2014a) suggested
that such a large difference is mainly due to the fact that different methods were used to
calculate surface area available for heterogeneous uptake. Specifically, Mogili et al. (2006a)
used the BET surface area, while Wagner et al. (2009) used Stoke diameters derived from APS
measurements to calculate the surface area. Tang et al. (2014a) further found that if the same
method is used to calculate surface area concentrations, $\gamma(N_2O_5)$ reported by the two studies
(Mogili et al., 2006b; Wagner et al., 2009) agree fairly well.

This issue becomes even more severe for experiments using mineral dust particles

deposited on a substrate. In these experiments the surface area available for heterogeneous



483 uptake is assumed to be either the projected area of dust particles (equal to the geometrical

484 surface area of the sample holder) or the BET surface area of the dust sample. Multiple layers

485 of powdered dust samples are typically deposited on a substrate. Consequently, it is not

486 uncommon that the BET surface area is several orders of magnitude larger than the projected

487 area (Nicolas et al., 2009; Liu et al., 2010; Tong et al., 2010). The surface area actually available

488 for heterogeneous uptake falls between the two extreme cases and varies for different studies.

489 For a very fast heterogeneous reaction it is likely that only the topmost few layers of a powdered

490 sample are accessible for the reactive trace gases, whereas more underlying layers become

491 available for slower uptake processes. Therefore, uptake coefficients reported by experiments

492 using aerosol samples, if available, are preferred and used in this study to estimate the

493 atmospheric importance of heterogeneous reactions. We note that a similar strategy has also

494 been adopted by the IUPAC task group (Crowley et al., 2010a).

495  In theory, transport of gaseous molecules within the interior space of the powdered

496 sample coupled to reaction with particle surface can be described by mathematical models. The

497 KML (Keyser-Moore-Leu) model initially developed to describe diffusion and reaction of

498 gaseous molecules in porous ice (Keyser et al., 1991; Keyser et al., 1993) has been used to

499 derive uptake coefficients for heterogeneous reactions of mineral dust particles. One major

500 drawback of the KML model (and other models with similar principles but different

501 complexities) is that it can be difficult to measure or accurately calculate diffusion constants of

502 reactive trace gases through powdered samples (Underwood et al., 2000).





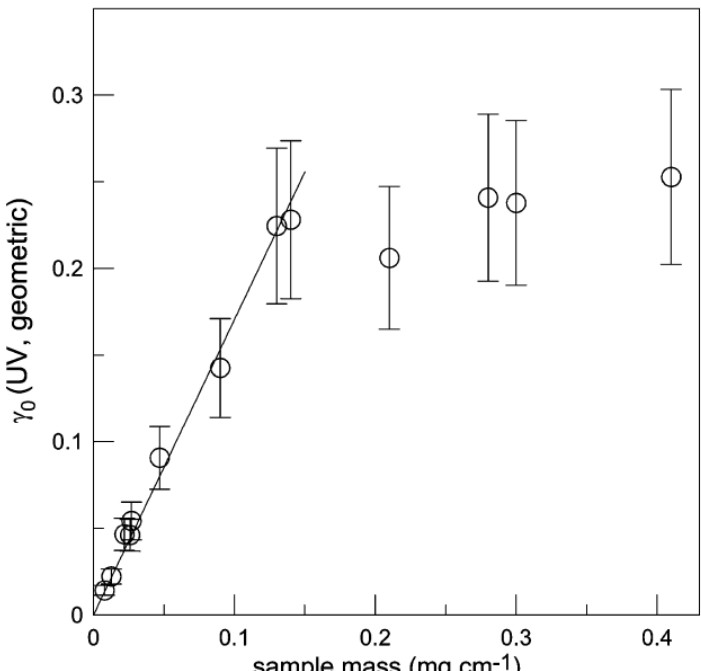

**Figure 2.** Projected area based uptake coefficients of $H_2O_2$ on irradiated $TiO_2$ particles as a

function of $TiO_2$ sample mass (per cm length of the support tube onto which $TiO_2$ particles

were deposited). Reprinted with permission from (Romanias et al., 2012b). Copyright 2012

American Chemical Society.


Grassian and coworkers developed a simple method to calculate surface area available

for heterogeneous uptake (Underwood et al., 2000; Li et al., 2002). If the thickness of a
powdered sample is smaller than the interrogation depth of the reactive trace gas (i.e. depth of
the sample which can actually be reached by the reactive trace gas), all the particles should be
accessible for heterogeneous uptake. In this case, uptake coefficients calculated using the
projected area should exhibit a linear mass dependence. The linear mass dependent (LMD)
regime can be experimentally determined, with an example shown in Figure 2. Figure 2
suggests that when the $TiO_2$ sample mass is <0.15 mg cm$^{-1}$, the projected area based uptake





coefficients depend linearly on the sample mass. If measurements are carried out within the
LMD regime, surfaces of all the particles are available for heterogeneous uptake and the BET
surface area should be used to calculate uptake coefficients (Underwood et al., 2000; Romanias
et al., 2012b; Bedjanian et al., 2013b).
Another way to circumvent the problem due to diffusion within interior space of
powdered samples is to use particles less than one layer (Hoffman et al., 2003a; Hoffman et al.,
2003b). This experimental strategy was used to investigate heterogeneous reactions of NaCl
with $HNO_3$, $N_2O_5$, and $ClONO_2$, and a mathematical model was developed to calculate the
effective surface area exposed to reactive trace gases (Hoffman et al., 2003a; Hoffman et al.,
2003b). Nevertheless, to our knowledge this method has not yet be used by laboratory studies
of heterogeneous reaction of mineral dust particles.
**2.2.2 Time dependence of heterogeneous kinetics**
When exposed to reactive trace gases, mineral dust surface may become deactivated
and thus gradually lose its heterogeneous reactivity. Figure 3 shows three representative
examples of decays of a reactive trace gas, X, after exposure to mineral dust particles. For the
case shown in Figure 3a, no surface active sites are consumed and the uptake rate is
independent of reaction time. Figure 3b displays another case in which surface reactive sites
may be consumed and heterogeneous uptake will cease after some exposure. In addition, as
shown in Figure 3c, an initial large uptake rate gradually decreases with time to a non-zero
constant value for longer exposure (i.e. the heterogeneous reaction reaches a "steady state").





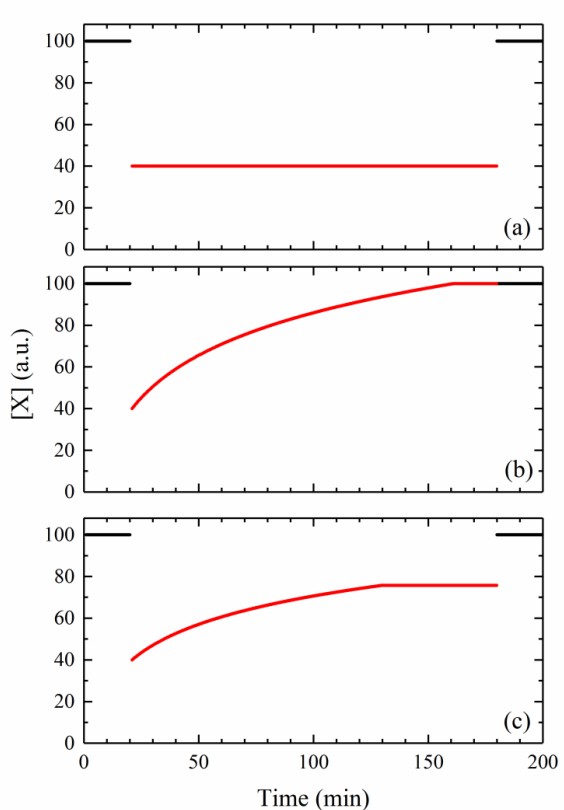


**Figure 3.** Synthetic data of the decay of a trace gas, X, due to heterogeneous reaction when it

is exposed to mineral dust particles from 20 to 180 min: a) not surface deactivation; b) complete

surface deactivation; c) partial surface deactivation. Black curves represent the concentration

of X without exposure to mineral dust particles (i.e. initial [X]), and red curves represent the

evolution of measured [X] during exposure of X to mineral dust particles.

In atmospheric chemistry community, heterogeneous reactions are usually treated as

pseudo-first-order processes (with respect to reactive trace gases), as implied by Eq. (1).

However, deactivation of mineral dust surfaces has been reported for a variety of trace gases

by experiments using particle ensembles deposited on a substrate (Underwood et al., 2001;



Hanisch and Crowley, 2003a; Ndour et al., 2009; Tang et al., 2010; Zhou et al., 2012; Romanias
et al., 2013; Liu et al., 2015). Therefore, uptake coefficients are normally set to be time
dependent (instead of assuming to be a constant), such that Eq. (1) is still valid for time
dependent heterogeneous kinetics. Many studies (Michel et al., 2003; Seisel et al., 2005;
Karagulian et al., 2006; Wang et al., 2011; El Zein et al., 2014) have reported initial and/or
steady-state uptake coefficients ($\gamma_0$ and $\gamma_{ss}$, respectively). What makes interpreting reported
uptake coefficients more difficult is that even for the same heterogeneous reaction, $\gamma_0$ and $\gamma_{ss}$
may exhibit dependence on experimental conditions (e.g., dust sample mass, trace gas
concentration, temperature, and etc.). For example, it takes less time for a reaction to reach
steady-state when higher concentrations are used for the same reactive trace gas. In many cases,
surface may be completely deactivated given sufficient reaction time. Furthermore, $\gamma_0$ is usually
reported as the first measurable uptake coefficient, which largely depends on the response time
(and time resolution) of the instrument used to detect the trace gas.
In aerosol flow tube experiments, on the other hand, exposure time of mineral dust
aerosol particles to trace gases are very short (typically <1 min). Therefore, significant surface
deactivation is not observed and decays of trace gases can usually be well described by pseudo-
first-order kinetics with time independent uptake coefficients (Vlasenko et al., 2006; Pradhan
et al., 2010a; Tang et al., 2012; Matthews et al., 2014).
Ideally laboratory studies of heterogeneous reactions should be carried out at or at least
close to atmospherically relevant conditions, such that experimental results can be directly used.
However, due to experimental challenges, laboratory studies are usually performed at much
shorter time scales (from <1 min to a few hours, compared to average residence time of several
days for mineral dust aerosol) and with much higher trace gas concentrations. Alternatively,
measurements can be conducted over a wide range of experimental conditions in order that
fundamental physical and chemical processes can be deconvoluted and corresponding rate



constants can be determined (Kolb et al., 2010; Davidovits et al., 2011; Pöschl, 2011). With
more accurate kinetic data, kinetic models which integrate these fundamental processes can be
constructed and applied to predict uptake coefficients for atmospherically relevant condition
(Ammann and Poschl, 2007; Pöschl et al., 2007; Shiraiwa et al., 2012; Berkemeier et al., 2013).
Unfortunately, measurements of this type are resource-demanding. In practice laboratory
studies of heterogeneous kinetics are usually carried out under very limited experimental
conditions. Therefore, there is a great need to invest more resource in fundamental laboratory
research.
**3 Heterogeneous reactions of mineral dust particles with tropospheric**
**oxidants and their direct precursors**

The importance of a heterogeneous reaction for removal of a trace gas, X, is determined

by the uptake coefficient and the aerosol surface area concentration, as suggested by Eq. (1). It
also depends on the rates of other removal processes in competition, although it is not
uncommon that this aspect has not been fully taken into account. In this section, previous
laboratory studies of heterogeneous reactions of mineral dust particles with OH, $HO_2$, $H_2O_2$,
$O_3$, HCHO, HONO, $NO_3$, and $N_2O_5$ are summarized, analyzed, and discussed. After that,
lifetimes of each trace gases with respect to their heterogeneous reactions with mineral dust are
calculated, using uptake coefficients listed in Table 2, followed by discussion of relative
importance of heterogeneous reactions for their removal in the troposphere. In addition, we
also discuss representative modeling studies to further demonstrate and illustrate the
importance of these heterogeneous reactions.

Uptake coefficients which are used in this paper to calculate lifetimes with respect to

heterogeneous reactions with mineral dust particles are shown in Table 2. The IUPAC Task
Group on Atmospheric Chemical Kinetic Data Evaluation has been compiling and evaluating
kinetic data for atmospheric heterogeneous reactions (Crowley et al., 2010a), and preferred





uptake coefficients are also recommended. It should be noted that uptake coefficients listed in
Table 2 do not intend to compete with those recommended by the IUPAC task group. Instead,
some of our values are largely based on their recommended values if available and proper.

**Table 3:** Uptake coefficients used in this work to calculate lifetimes of OH, HO$_2$, H$_2$O$_2$, O$_3$,
HCHO, HONO, NO$_3$, and N$_2$O$_5$ with respect to heterogeneous reactions with mineral dust
aerosol.

| species | uptake coefficient | species | uptake coefficient |
|---------|--------------------|---------|--------------------|
| OH | 0.2 | HCHO | $1\times10^{-5}$ |
| HO$_2$ | 0.038 | HONO | $1\times10^{-6}$ |
| H$_2$O$_2$ | $1\times10^{-3}$ | NO$_3$ | 0.018 |
| O$_3$ | $4.5\times10^{-6}$ | N$_2$O$_5$ | 0.020 |


The pseudo-first-order loss rate depends on the aerosol surface area concentration,
which depends on aerosol number concentration and its size distribution. Although particle
sizing instruments such as aerodynamic particle sizer (APS) and scanning particle mobility
sizer (SMPS) are commercially available, particle mass concentrations are still more widely
measured and reported. Therefore, it is convenient to calculate lifetimes based on mass
concentration instead of surface area concentration. This calculation requires information of
particle size and density. For simplicity dust aerosol particles are assumed to have an average
particles diameter of 1 μm and a density of 2.7 g cm$^{-3}$. Consequently, the lifetime of X with
respect to its heterogeneous reaction with mineral dust, $\tau_{het}(X)$, can be described by Eq. (6)
(Wagner et al., 2008; Tang et al., 2010; Tang et al., 2012):
$$\tau_{het}(X) = \frac{1.8\times10^8}{\gamma_{eff}(X)\cdot c(X)\cdot L} \quad (6)$$

where $\gamma_{eff}(X)$ is the effective uptake coefficient of X, $c(X)$ is the average molecular speed of X
(cm s$^{-1}$), and $L$ is the mineral dust loading (i.e. mass concentration) in μg m$^{-3}$. Mass





concentrations of mineral dust aerosol particles in the troposphere show high variability,
ranging from a few $\mu g\ m^{-3}$ in background regions such as north Atlantic to >1000 $\mu g\ m^{-3}$ during
extreme dust storms (Prospero, 1979; Zhang et al., 1994; de Reus et al., 2000; Gobbi et al.,
2000; Alfaro et al., 2003). To take into account this spatial and temporal variation, mass
concentrations of 10, 100, and 1000 $\mu g\ m^{-3}$ are used in this paper to assess atmospheric
significance of heterogeneous reactions with mineral dust for the removal of trace gases.

**3.1 OH and HO$_2$ radicals**

**3.1.1 OH radicals**

Heterogeneous uptake of OH radicals by mineral dust particles was first investigated
using a coated wall flow tube with detection of OH radicals by electron paramagnetic resonance
(EPR) (Gershenzon et al., 1986). The uptake coefficient was reported to be 0.04±0.02 for $Al_2O_3$
and 0.0056±0.0020 for $SiO_2$, independent of temperature in the range of 253-348 K
(Gershenzon et al., 1986). Using laser induced fluorescence (LIF), Suh et al. (2000) measured
concentration changes of OH radicals after the gas flow was passed through a wire screen
loaded with $TiO_2$ (anatase or rutile), $\alpha$-$Al_2O_3$, or $SiO_2$ under dry conditions. It is shown that the
uptake coefficients, $\gamma$(OH), increased with temperature from ~310 K to ~350 K for all the three
oxides, being $(2\text{-}4)\times10^{-4}$ for $TiO_2$, $(2\text{-}4)\times10^{-3}$ for $SiO_2$, and $(5\text{-}6)\times10^{-3}$ for $\alpha$-$Al_2O_3$ (Suh et al.,
2000). Unfortunately, most of the results reported by Suh et al. (2000) are only presented
graphically. In an earlier study (Bogart et al., 1997), $\gamma$(OH) was reported to be 0.41±0.04 at
300 K on deposited $SiO_2$ films, decreasing with temperature. OH($X^2\Pi$) radicals used by Bogart
et al. were generated in a 20:80 tetraethoxysilane/$O_2$ plasmas and their atmospheric relevance
is not very clear; therefore, this study is not included in Table 1 or further discussed.
The average $\gamma$(OH) was determined to be 0.20 for $Al_2O_3$ at room temperature under dry
conditions (Bertram et al., 2001), using a coated wall flow tube coupled to chemical ionization
mass spectrometry (CIMS). In a following study, the RH dependence of $\gamma$(OH) on $SiO_2$ and





Al$_2$O$_3$ at room temperature was investigated (Park et al., 2008). It is found that $\gamma$(OH) increased
from 0.032±0.007 at 0% RH to 0.098±0.022 at 33% RH for SiO$_2$ and from 0.045±0.005 at 0%
RH to 0.084±0.012 at 38% RH for Al$_2$O$_3$ (Park et al., 2008).






**Table 4:** Summary of previous laboratory studies on heterogeneous reactions of mineral dust with OH and HO$_2$ radicals

| Trace gases | Dust | Reference | T (K) | Concentration (molecule cm$^{-3}$) | Uptake coefficients | Techniques |
|---|---|---|---|---|---|---|
| OH | TiO$_2$ | Suh et al., 2000 | 308 to 350 | ~4×10$^{12}$ | (2-4)×10$^{-4}$, increasing with temperature | LIF |
| | SiO$_2$ | Gershenzon et al., 1986 | 253-343 | <2×10$^{12}$ | 0.0056±0.002, independent of temperature | CWFT-EPR |
| | | Suh et al., 2000 | 308 to 350 | ~4×10$^{12}$ | (2-4)×10$^{-3}$, increasing with temperature | LIF |
| | | Park et al, 2008 | room temperature | ~4×10$^{11}$ | 0.032±0.007 at 0% RH and 0.098±0.022 at 33% RH | CWFT-CIMS |
| | Al$_2$O$_3$ | Gershenzon et al., 1986 | 253-343 | <2×10$^{12}$ | 0.04±0.02, independent of temperature | CWFT-EPR |
| | | Suh et al., 2000 | 308 to 350 | ~4×10$^{12}$ | (5-6)×10$^{-3}$, increasing with temperature | LIF |
| | | Bertram et al, 2001 | room temperature | (1-100)×10$^{9}$ | 0.20 | CWFT-CIMS |
| | | Park et al, 2008 | room temperature | ~4×10$^{11}$ | 0.045±0.005 at 0%RH and 0.084±0.012 at 38% RH | CWFT-CIMS |
| | ATD | Bedjanian et al., 2013a | 275-320 | (0.4-5.2)×10$^{12}$ | 0.20 at 0% RH, showing a negative RH dependence but no dependence on temperatures | CRFT-MS |
| HO$_2$ | ATD | Bedjanian et al., 2013b | 275-320 | (0.35-3.3)×10$^{12}$ | 0.067±0.004 at 0% RH, showing a negative RH dependence (0.02-94%) but no dependence on temperature. | CRFT-MS |
| | | Matthews et al., 2014 | 291±2 | (3-10)×10$^{8}$ | 0.018±0.006 when HO$_2$ concentration was 3×10$^{8}$ molecule cm$^{-3}$ and 0.031±0.008 when HO$_2$ concentration was 3×10$^{8}$ molecule cm$^{-3}$. No RH (5-76%) dependence was observed. | AFT-FAGE |





Recently a coated rod flow tube was used to investigate uptake of OH radicals by
Arizona test dust (ATD) particles (Bedjanian et al., 2013a) as a function of temperature (275-
320 K) and RH (0.03-25.9%). Gradual surface deactivation was observed, and the initial uptake
coefficient was found to be independent of temperature and decrease with increasing RH, given
by Eq. (7):

$\gamma_0 = 0.2/(1 + RH^{0.36})$    (7)

with an estimated uncertainty of ±30%. Please note that uptake coefficients reported by
Bedjanian et al. (2013a) are based on the geometrical area of the rod coated with ATD particles
and thus should be considered as the upper limit. No effect of UV radiation, with $J(NO_2)$ up to
0.012 s$^{-1}$, was observed (Bedjanian et al., 2013a). In addition, $H_2O$ and $H_2O_2$ were found to be
the major and minor products in the gas phase respectively (Bedjanian et al., 2013a), as shown
in Figure 4.

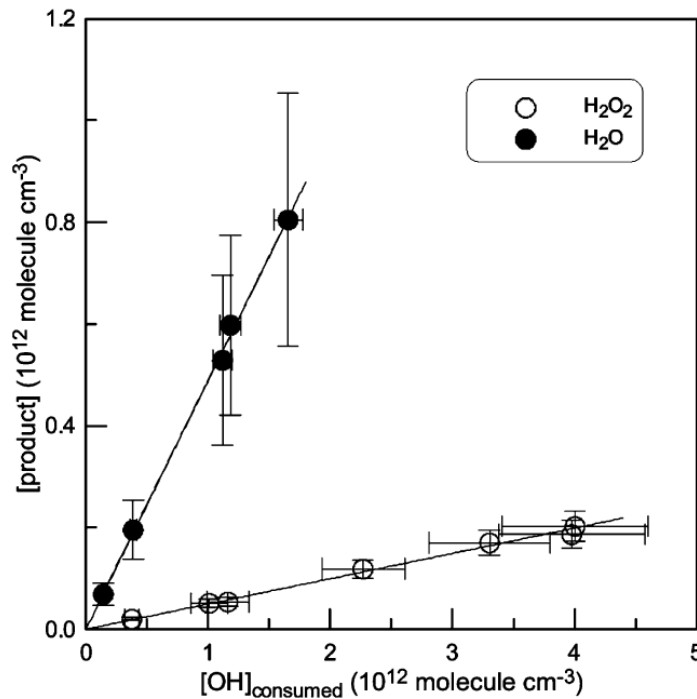






**Figure 4.** Concentrations of $H_2O$ (solid circles) and $H_2O_2$ (open circles) produced in the gas
phase due to heterogeneous reaction of OH radicals with ATD particles. Reprinted with
permission from Bedjanian et al. (2013a). Copyright 2013 American Chemical Society.

As shown in Figure 5, $\gamma$(OH) reported by previous flow tube studies, except that onto

$SiO_2$ particles reported by Gershenzon et al. (1986), show reasonably good agreement,
considering that different minerals were used. Reported $\gamma$(OH) are larger than 0.02 in general,
suggesting that mineral dust exhibits relatively large reactivity towards OH radicals.
Discrepancies are also identified from data presented in Figure 5, with the most evident one
being the effect of RH. Park et al. (2008) found that $\gamma$(OH) increased significantly with RH for
both $SiO_2$ and $Al_2O_3$, while Bedjanian et al. (2013b) suggested that $\gamma$(OH) showed a negative
dependence on RH. It is not clear yet whether different minerals used by these two studies can
fully account for the different RH dependence observed. Furthermore, a positive dependence
of $\gamma$(OH) on temperature was found by Suh et al. (2000) for $TiO_2$, $\alpha$-$Al_2O_2$, and $SiO_2$, while
Bogart et al. (1997) reported a negative temperature effect for deposited $SiO_2$ film and no
significant dependence on temperature was found for ATD (Bedjanian et al., 2013a).

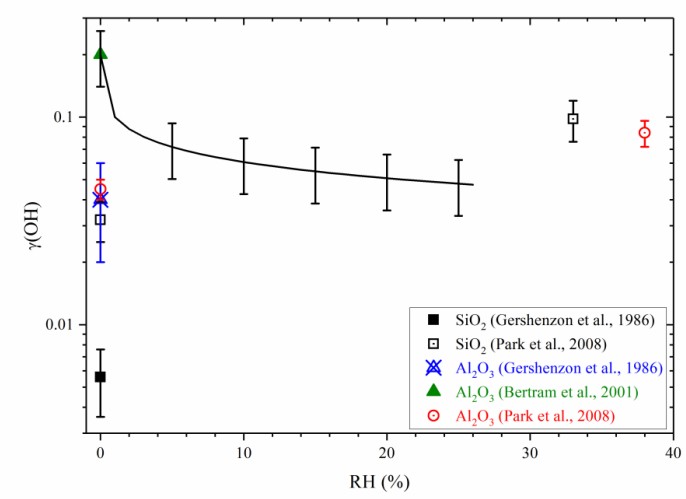






**Figure 5.** Uptake coefficients of OH radicals for different minerals at room temperature, as
reported by different studies. The plotted RH dependence of $\gamma$(OH) for ATD (solid curve) is
based on the parameterization reported by Bedjanian et al. (2013a), i.e. Eq. (7).

A $\gamma$(OH) value of 0.2, reported by Bedjanian et al. (2013a) for ATD, is used in our

present work to evaluate the importance of heterogeneous uptake of OH radicals by mineral
dust aerosol. According to Eq. (6), dust mass loadings of 10, 100, and 1000 μg m$^{-3}$ correspond
to $\tau_{het}$(OH) of ~25 min, 150 s, and 15 s with respect to heterogeneous uptake by mineral dust.
As discussed in Section 2.1.1, lifetimes of tropospheric OH are in the range of 1 s or less in
very clean regions and <0.1 s in polluted and forested areas, much shorter than $\tau_{het}$(OH). Even
if $\gamma$(OH) is assumed to be 1, for uptake by 1 μm particles $\gamma_{eff}$(OH) is calculated to be 0.23,
which is only 15% larger than what we use to calculate $\tau_{het}$(OH). Therefore, it can be concluded
that heterogeneous reaction with mineral dust aerosol is not a significant sink for OH radicals
in the troposphere.
**3.1.2 HO$_2$ radicals**

To the best of our knowledge, only two previous studies have investigated

heterogeneous uptake of HO$_2$ radicals by mineral dust particles. Bedjanian et al. (2013b) used
a coated rod flow tube to study the interaction of HO$_2$ radicals with ATD film as a function of
temperature and RH. Surface deactivation was observed, and $\gamma_0$, based on the geometrical area
of dust films, was determined to be 0.067±0.004 under dry conditions (Bedjanian et al., 2013b).
The initial uptake coefficient, independent of temperature, was found to decrease with RH,
given by Eq. (8):

$$\gamma_0 = 1.2/(18.7 + RH^{1.1}) \quad (8)$$

with an estimated uncertainty of ±30%. UV radiation, with $J$(NO$_2$) ranging from 0 to 0.012 s$^{-1}$,
did not affect uptake kinetics significantly. In addition, the yield of H$_2$O$_2$(g), defined as the



ratio of formed $H_2O_2(g)$ molecules to consumed $HO_2$ radicals, was determined to be <5%
(Bedjanian et al., 2013b).
In the second study (Matthews et al., 2014), an aerosol flow tube was deployed to
measure $\gamma(HO_2)$ onto ATD aerosol particles at 291±2 K, with $HO_2$ detection via the
fluorescence assay by gas expansion technique. No significant effect of RH in the range of 5-
76% was observed, and $\gamma(HO_2)$ was reported to be 0.031±0.008 for $[HO_2]$ of $3\times10^8$ molecule
$cm^{-3}$ and 0.018±0.006 for $[HO_2]$ of $1\times10^9$ molecule $cm^{-3}$ (Matthews et al., 2014). In addition,
$\gamma(HO_2)$ was found to decrease with increasing reaction time. The negative dependence of $\gamma(HO_2)$
on $[HO_2]$ and reaction time implies that ATD surface is gradually deactivated upon exposure
to $HO_2$ radicals, as directly observed by Bedjanian et al. (2013a).

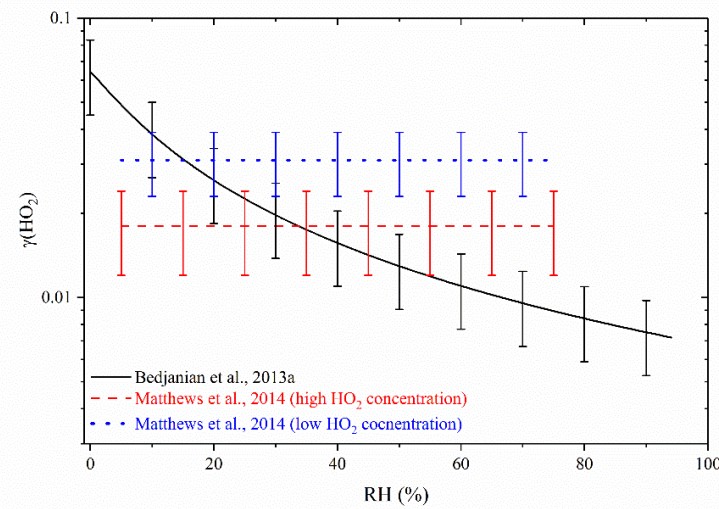


**Figure 6.** RH dependence of $\gamma(HO_2)$ for ATD reported by two previous studies. Solid curve,
reported by Bedjanian et al. (2013b) with initial $[HO_2]$ in the range of $(0.35-3)\times10^{12}$ molecule
$cm^{-3}$; dashed and dotted curve, reported by Matthews et al. (2014) with initial $[HO_2]$ of $1\times10^9$
and $3\times10^8$ molecule $cm^{-3}$, respectively. Numerical data for $\gamma(HO_2)$ at different RH were not
provided by Matthews et al. (2014), and thus in this figure we plot their reported average $\gamma(HO_2)$



together with their estimated uncertainties. The plotted RH dependence of $\gamma(HO_2)$ reported by
Bedjanian et al. (2013b) is based on their proposed parameterization, i.e. Eq. (8).

Figure 6 shows the effect of RH on $\gamma(HO_2)$ for ATD particles. A quick look at Figure 6
could lead to the impression that $\gamma(HO_2)$ reported by two previous studies (Bedjanian et al.,
2013b; Matthews et al., 2014) agree relatively well, especially considering that two very
different experimental techniques were used. Nevertheless, Matthews et al. (2014), who
conducted their measurements with initial $[HO_2]$ which are 3-4 orders of magnitude lower than
those used by Bedjanian et al. (2013a), found a significant negative dependence of $\gamma(HO_2)$ on
initial $[HO_2]$. If this trend can be further extrapolated to higher initial $[HO_2]$, one may expect
that if carried out with initial $[HO_2]$ similar to those used by Bedjanian et al. (2013a), Matthews
et al. (2014) may find much smaller $\gamma(HO_2)$. In addition, these two studies also suggest very
different RH effects, as evident from Figure 6. We also note that ATD is the only one type of
mineral dust onto which heterogeneous uptake of $HO_2$ radicals was investigated, and the effect
of mineralogy is not clear at all yet. Therefore, our understanding of heterogeneous reactions
of $HO_2$ radicals with mineral dust particles is very limited.
Apart from these included in Table 4, the uptake of $HO_2$ by analogues of meteoric
smoke particles was also examined at room temperature (James et al., 2017), using an aerosol
flow tube. At $(10\pm1)\%$ RH, the uptake coefficient was determined to be $0.069\pm0.012$ for olivine
$(MgFeSiO_4)$, $0.073\pm0.004$ for fayalite $(Fe_2SiO_4)$, and $0.0043\pm0.0004$ for forsterite $(Mg_2SiO_4)$,
respectively. It appears that meteoric smoke particles which do not contain Fe, these which
contain Fe show much larger heterogeneous reactivity towards $HO_2$ radicals. The experimental
result indicates a catalytic role of Fe in $HO_2$ uptake, as supported by electronic structure
calculations (James et al., 2017). Though its tropospheric relevance is limited, this study



provides valuable mechanistic insights into heterogeneous reaction of mineral dust with $HO_2$
radicals.

For reasons discussed in Section 2.2.1, $\gamma(HO_2)$ reported by Matthews et al. (2014) using

aerosol samples are used to calculate $\tau_{het}(HO_2)$ with respect to uptake onto mineral dust.
Another reason that the data reported by Matthews et al. (2014) are preferred is that $[HO_2]$ used
in this study were low enough to be of direct atmospheric relevance. As a result, $\gamma(HO_2)$
measured at lower initial $[HO_2]$ ($3\times10^8$ molecule cm$^{-3}$), equal to $0.031\pm0.008$, is adopted in our
current work to assess the significance of $HO_2$ uptake by mineral dust. Using Eq. (6), $\tau_{het}(HO_2)$
is estimated to be 2.2, 22, and 222 min for dust mass concentrations of 1000, 100, and 10 μg m$^{-3}$,
respectively. Typical $HO_2$ lifetimes in the troposphere, as summarized in Table 1, show large
variability, ranging from <1 s (Ren et al., 2003) to >30 min (Whalley et al., 2011). Therefore,
dust aerosol with moderate mass concentrations could be a significant tropospheric $HO_2$ sink
except regions with very high NO levels.

The importance of heterogeneous uptake as a $HO_2$ sink in the troposphere has also been

demonstrated by several more sophisticated modelling studies. For example, it is found that
while standard gas phase chemical mechanism used by the GEOS-Chem model would
overestimate $HO_2$ and $H_2O_2$ concentrations observed in the Arctic troposphere in the spring,
including heterogeneous reaction of $HO_2$ with an average $\gamma(HO_2)$ of >0.1 in the model could
better reproduce the measured concentrations and vertical profiles of $HO_2$ and $H_2O_2$ (Mao et
al., 2010a). Though not directly relevant for mineral dust aerosol, this study provided strong
evidence that heterogeneous uptake can be an important but yet not fully recognized sink for
tropospheric $HO_2$ radicals (Mao et al., 2010a). Using a global tropospheric model, Macintyre
and Evans (2011) analyzed the sensitivity of model output to $\gamma(HO_2)$ values used in the model.
A global average $\gamma(HO_2)$ of 0.028 was derived from available laboratory studies (Macintyre
and Evans, 2011), and large regional differences in modelled $O_3$ were observed between



simulations using $\gamma(HO_2)$ parameterization developed by Macintyre and Evans (2011) and
those using a constant $\gamma(HO_2)$ of 0.2. This results highlights the importance of accurate
determination of $\gamma(HO_2)$ under different tropospheric conditions (e.g., aerosol composition, RH,
and temperature).
The impact of $HO_2$ uptake by mineral dust has also been investigated by several
modelling studies. For example, an observation constrained box model study (Matthews et al.,
2014) suggested that heterogeneous reaction with mineral dust could result in >10% reduction
in $HO_2$ concentrations in Cape Verde, using a $\gamma(HO_2)$ of 0.038. A WRF-Chem simulation,
using $\gamma(HO_2)$ reported by Bedjanian et al. (2013a), showed that heterogeneous uptake by
mineral dust could reduce $HO_2$ concentrations by up to 40% over northern India during a pre-
monsoon dust storm (Kumar et al., 2014).
One may assume that heterogeneous reaction of $HO_2$ with aerosol particles leads to the
formation of $H_2O_2$ (Graedel et al., 1986; Thornton and Abbatt, 2005). A second channel
without $H_2O_2$ formation, i.e. simple decomposition of $HO_2$ radicals to $H_2O$ and $O_2$, may also
be important (Bedjanian et al., 2013b; Mao et al., 2013a). Atmospheric impacts can be very
different for these two mechanisms. While the second pathway represents a net sink for $HO_2$
in the troposphere, the first channel only converts $HO_2$ to $H_2O_2$ via heterogeneous reaction and
is thus of limited efficacy as a net sink for HOx because $H_2O_2$ can undergo photolysis to
generate OH radicals.
The relative importance of these two mechanisms has been explored by modelling
studies. Mao et al. (2010a) found that only including the first reaction channel (with $H_2O_2$
production) will overestimate $H_2O_2$ in the Arctic, while only considering the second channel
(without $H_2O_2$ production) would cause underestimation of $H_2O_2$. Consequently, it seems that
both channels have non-negligible contributions in the troposphere (Mao et al., 2010a).
Significant differences in modelled OH, $HO_2$, $O_3$, and sulfate concentrations have been found



by a global model study when including two mechanisms separately (Macintyre and Evans,
2011). One experimental study (Bedjanian et al., 2013b) measured gas phase products for
heterogeneous reaction of $HO_2$ radicals with ATD particles and found that gaseous $H_2O_2$
formed in this reaction is minor but probably non-negligible. Considering the importance of
mechanisms of heterogeneous reaction of $HO_2$ with mineral dust, further experimental work is
required. Furthermore, mineralogy and RH may also impact the yield of $H_2O_2(g)$, but these
effects are not clear yet.
**3.2 $H_2O_2$**

Pradhan et al. (2012a, 2012b) utilized an aerosol flow tube to investigate heterogeneous

interaction of $H_2O_2$ with airborne $TiO_2$, Gobi dust, and Saharan dust particles at $295\pm2$ K, and
$H_2O_2$ was detected by CIMS. A negative dependence of $\gamma(H_2O_2)$ on RH was observed for $TiO_2$,
with $\gamma(H_2O_2)$ decreasing from $(1.53\pm0.11)\times10^{-3}$ at 15% RH to $(6.47\pm0.74)\times10^{-4}$ at 40% RH and
$(5.04\pm0.58)\times10^{-4}$ at 70% RH (Pradhan et al., 2010a). In contrast, $H_2O_2$ uptake kinetics
displayed positive dependence on RH for Gobi and Saharan dust, with $\gamma(H_2O_2)$ increasing from
$(3.33\pm0.26)\times10^{-4}$ at 15% RH to $(6.03\pm0.42)\times10^{-4}$ at 70% RH for Gobi dust and from
$(6.20\pm0.22)\times10^{-4}$ at 15% RH to $(9.42\pm0.41)\times10^{-4}$ at 70% RH for Saharan dust (Pradhan et al.,
2010b). It appears that heterogeneous reactivity of Saharan dust towards $H_2O_2$ is significantly
higher than Gobi dust.

Heterogeneous interaction of gaseous $H_2O_2$ with $SiO_2$ and $\alpha$-$Al_2O_3$ particles was

investigated at $298\pm1$ K, using transmission FTIR to probe particle surfaces and a HPLC-based
offline technique to measure gaseous $H_2O_2$ (Zhao et al., 2011b). It is found that most of $H_2O_2$
molecules were physisorbed on $SiO_2$ surface and a small amount of molecularly adsorbed $H_2O_2$
underwent thermal decomposition. In contrast, catalytic decomposition occurred to a large
fraction of $H_2O_2$ uptaken by $\alpha$-$Al_2O_3$, though some $H_2O_2$ molecules were also physisorbed on
the surface (Zhao et al., 2011b). The uptake coefficient, based on the BET surface area, was





found to be independent of initial $H_2O_2$ concentrations (1.27-13.8 ppmv) while largely affected
by RH (Zhao et al., 2011b). $\gamma(H_2O_2)$ decreased from $(1.55\pm0.14)\times10^{-8}$ at 2% RH to
$(0.81\pm0.11)\times10^{-8}$ at 21% RH for $SiO_2$ particles, and further increase in RH (up to 76%) did not
affect the uptake kinetics (Zhao et al., 2011b). A similar dependence of $\gamma(H_2O_2)$ on RH was
also observed for $\alpha$-$Al_2O_3$: $\gamma(H_2O_2)$ decreased from $(1.21\pm0.04)\times10^{-7}$ at 2% RH to
$(0.84\pm0.07)\times10^{-7}$ at 21% RH, and the effect of RH was not significant for RH in the range of
21-76% (Zhao et al., 2011b). Compared to $SiO_2$, $\alpha$-$Al_2O_3$ appears to be much more reactive
towards $H_2O_2$.

In a following study, using the same experimental setup, Zhao et al. (2013) explored

heterogeneous interaction of $H_2O_2$ with fresh, $HNO_3$-processed, and $SO_2$-processed $CaCO_3$
particles. The uptake of $H_2O_2$ on fresh $CaCO_3$ particles was drastically reduced with increasing
RH, indicating that $H_2O_2$ and $H_2O$ compete for surface reactive sites. In addition, about 85-90%
of $H_2O_2$ molecules uptaken by fresh $CaCO_3$ particles undergo decomposition (Zhao et al.,
2013). Unfortunately no uptake coefficients were reported (Zhao et al., 2013). Pretreatment of
$CaCO_3$ particles with $HNO_3$ or $SO_2$ can significantly affect their heterogeneous reactivity
towards $H_2O_2$. The effect of $HNO_3$ pretreatment increases with surface coverage of nitrate
(formed on $CaCO_3$ particles), showing an interesting dependence on RH. Pretreatment of
$CaCO_3$ with $HNO_3$ reduced its heterogeneous reactivity by 30-85% at 3% RH, while it led to
enhancement of reactivity towards $H_2O_2$ by 20-60% at 25% RH, a factor of 1-3 at 45% RH,
and a factor of 3-8 at 75% RH (Zhao et al., 2013). At low RH, formation of $Ca(NO_3)_2$ on the
surface could deactivate $CaCO_3$; however, $Ca(NO_3)_2$ may exit as an aqueous film at higher RH
(Krueger et al., 2003b; Liu et al., 2008b), consequently leading to large enhancement of $H_2O_2$
uptake. Compared to fresh $CaCO_3$, $SO_2$-processed particles always exhibit much higher
reactivity towards $H_2O_2$, and enhancement factors, increasing with RH, were observed to fall
into the range of 3-10 (Zhao et al., 2013).





Heterogeneous uptake of $H_2O_2$ by several oxides was investigated at 298 K using a

Knudsen cell reactor with $H_2O_2$ measured by a quadrupole mass spectrometer (Wang et al.,

2011). $\gamma_0(H_2O_2)$, based on the BET surface area of sample powders, was determined to be

$(1.00\pm0.11)\times10^{-4}$ for α-$Al_2O_3$, $(1.66\pm0.23)\times10^{-4}$ for MgO, $(9.70\pm1.95)\times10^{-5}$ for $Fe_2O_3$, and

$(5.22\pm0.90)\times10^{-5}$ for $SiO_2$, respectively (Wang et al., 2011). Surface deactivation occurred for

all the surfaces, though complete surface saturation was only observed for $SiO_2$ after extended

$H_2O_2$ exposure. This may indicate that the uptake of $H_2O_2$ by α-$Al_2O_3$, MgO, and $Fe_2O_3$ are of

catalytic nature to some extent (Wang et al., 2011).

Continuous wave CRDS was employed to detect the depletion of $H_2O_2$ and formation

of $HO_2$ radicals in the gas phase above $TiO_2$ films which were exposed to gaseous $H_2O_2$ and

illuminated by a light-emitting diode at 375 nm (Yi et al., 2012). Three different $TiO_2$ samples

were investigated, including Degussa P25 $TiO_2$, Aldrich anatase, and Aldrich rutile. $H_2O_2$

decays did not occur in the absence of $TiO_2$. In addition, production of $HO_2$ radicals was only

observed in the presence of $H_2O_2$, and the presence of $O_2$ did not have a significant effect.

Therefore, Yi et al. (2012) suggested that the production of $HO_2$ radicals is due to the

photodecomposition of $H_2O_2$ on $TiO_2$ surfaces. Decays of $H_2O_2$ and formation of $HO_2$ are

found to vary with $TiO_2$ samples (Yi et al., 2012). Photo-degradation of $H_2O_2$ is fast for P25

$TiO_2$ samples and much slower for anatase and rutile; furthermore, significant production of

$HO_2$ radicals in the gas phase was observed for anatase and rutile but not for P25 $TiO_2$.

However, no uptake coefficients were reported by Yi et al. (2012).

Zhou et al. (2012) first explored the temperature dependence of heterogeneous

reactivity of mineral dust towards $H_2O_2$, using a Knudsen cell reactor coupled to a quadrupole

mass spectrometer. The uptake kinetics show negative temperature dependence, with $\gamma_0(H_2O_2)$

(BET surface area based) decreasing from $(12.6\pm2.52)\times10^{-5}$ at 253 K to $(6.08\pm1.22)\times10^{-5}$ at

313 K for $SiO_2$ and from $(7.11\pm1.42)\times10^{-5}$ at 253 K to $(3.00\pm0.60)\times10^{-5}$ at 313 K for $CaCO_3$



(Zhou et al., 2012). Complete surface deactivation was observed for both dust samples after
long exposure to $H_2O_2$ (Zhou et al., 2012). In a following study, the effects of temperature on
the uptake of $H_2O_2$ by ATD and two Chinese dust samples were also investigated (Zhou et al.,
2016). $\gamma_0(H_2O_2)$, based on the BET surface area, was observed to decrease with temperature,
from $(2.71\pm0.54)\times10^{-4}$ at 253 K to $(1.47\pm0.29)\times10^{-4}$ at 313 K for ATD, and from
$(3.56\pm0.71)\times10^{-4}$ at 253 K to $(2.19\pm0.44)\times10^{-4}$ at 313 K for Inner Mongolia desert dust, and
from $(7.34\pm1.47)\times10^{-5}$ at 268 K to $(4.46\pm0.889)\times10^{-4}$ at 313 K for Xinjiang sierozem (Zhou et
al., 2016). In addition, loss of heterogeneous reactivity towards $H_2O_2$ was observed for all the
three dust samples (Zhou et al., 2016).





**Table 5:** Summary of previous laboratory studies on heterogeneous reactions of mineral dust with $H_2O_2$

| Dust | Reference | $T$ (K) | Concentration (molecule cm⁻³) | Uptake coefficient | Techniques |
|---|---|---|---|---|---|
| $TiO_2$ | Pradhan et al., 2010a | 295±2 | ~4.1×10¹² | $(1.53\pm0.11)\times10^{-3}$ at 15% RH, $(6.47\pm0.74)\times10^{-4}$ at 40% RH, and $(5.04\pm0.58)\times10^{-4}$ at 70% RH | AFT-CIMS |
| | Romanias et al., 2012a | 275-320 | $(0.17-120)\times10^{12}$ | Under dark conditions at 275 K, $\gamma_0$ was determined to be $(4.1\pm1.2)\times10^{-3}$ at 0% RH, $(5.1\pm1.5)\times10^{-4}$ at 20% RH, $(3.4\pm1.0)\times10^{-4}$ at 40% RH, $(2.7\pm0.8)\times10^{-4}$ at 60% RH, and $(2.3\pm0.7)\times10^{-4}$ at 80% RH. Surface deactivation was observed under dark conditions, and UV illumination could enhance the steady state uptake of $H_2O_2$. | CRFT-MS |
| | Yi et al., 2012 | not stated | $(3\pm1)\times10^{13}$ | No uptake coefficients were not reported. | CRDS |
| $SiO_2$ | Zhao et al., 2011 | 298±1 | $(3.2-34.5)\times10^{13}$ | $\gamma(H_2O_2)$ decreased from $(1.55\pm0.14)\times10^{-8}$ at 2% RH to $(0.81\pm0.11)\times10^{-8}$ at 21% RH, and further increase in RH (up to 76%) did not affect uptake kinetics. | T-FTIR, HPLC |
| | Wang et al., 2011 | 298 | $(1-25)\times10^{11}$ | $\gamma_0$: $(5.22\pm0.90)\times10^{-5}$ | KC-MS |
| | Zhou et al., 2012 | 253-313 | $(0.37-3.7)\times10^{12}$ | Under dry conditions, $\gamma_0$ decreased from $(12.6\pm2.52)\times10^{-5}$ at 253 K to $(6.08\pm1.22)\times10^{-5}$ at 313 K. | KC-MS |
| $Al_2O_3$ | Zhao et al., 2011 | 298±1 | $(3.2-34.5)\times10^{13}$ | $\gamma(H_2O_2)$ decreased from $(1.21\pm0.04)\times10^{-7}$ at 2% RH to $(0.84\pm0.07)\times10^{-7}$ at 21% RH, and the effect of RH was not significant for RH in the range of 21-76%. | T-FTIR, HPLC |
| | Wang et al., 2011 | 298 | $(1-25)\times10^{11}$ | $\gamma_0$: $(1.00\pm0.11)\times10^{-4}$ ; $\gamma_{ss}$: $1.1\times10^{-5}$ | KC-MS |
| | Romanias et al., 2013 | 268-320 | $(0.16-12.6)\times10^{12}$ | At 280 K, $\gamma_0$ was determined to be $(1.1\pm0.3)\times10^{-3}$ at 0% RH, $(1.2\pm0.3)\times10^{-4}$ at 10% RH, $(3.5\pm1.0)\times10^{-5}$ at 40% RH, and $(2.1\pm0.6)\times10^{-5}$ at 70% RH, showing a negative dependence on RH. No significant effect was observed for UV illumination. | CRFT-MS |





| | Reference | T (K) | | Comment | Method |
|---|---|---|---|---|---|
| $Fe_2O_3$ | Wang et al., 2011 | 298 | $(1\text{-}25)\times10^{11}$ | $\gamma_0$: $(9.70\pm1.95)\times10^{-4}$; $\gamma_{ss}$: $5.5\times10^{-5}$ | KC-MS |
| | Romanias et al., 2013 | 268-320 | $(0.16\text{-}12.6)\times10^{12}$ | At 280 K, $\gamma_0$ was determined to be $(1.1\pm0.3)\times10^{-3}$ at 0% RH, $(1.7\pm0.5)\times10^{-4}$ at 10% RH, $(6.7\pm2.0)\times10^{-5}$ at 40% RH, and $(4.5\pm1.4)\times10^{-5}$ at 70% RH, showing a negative dependence on RH. No significant effect was observed for UV illumination. | CRFT-MS |
| $CaCO_3$ | Zhou et al., 2012 | 253-313 | $(0.37\text{-}3.7)\times10^{12}$ | Under dry conditions, $\gamma_0$ decreased from $(7.11\pm1.42)\times10^{-5}$ at 253 K to $(3.00\pm0.60)\times10^{-5}$ at 313 K. | KC-MS |
| | Zhao et al., 2013 | 298±1 | $1.3\times10^{14}$ | The uptake of $H_2O_2$ on fresh $CaCO_3$ particles decreased drastically with RH. Pretreatment with $SO_2$ always enhances its reactivity towards $H_2O_2$, whereas exposure to $HNO_3$ could either enhance or suppress $H_2O_2$ uptake, depending on RH. Numerical values for uptake coefficients were reported. | T-FTIR, HPLC |
| ATD | El Zein et al., 2014 | 268-320 | $(0.18\text{-}5.1)\times10^{12}$ | Under dark conditions at 275 K, $\gamma_0$ was determined to be $(4.8\pm1.4)\times10^{-4}$ at 0% RH, $(5.8\pm1.8)\times10^{-5}$ at 20% RH, $(3.9\pm1.2)\times10^{-5}$ at 40% RH, and $(3.0\pm0.9)\times10^{-5}$ at 60% RH. Surface deactivation was observed under dark conditions, and UV illumination could enhance the steady state uptake of $H_2O_2$. | CRFT-MS |
| | Zhou et al., 2016 | 253-313 | $(0.26\text{-}1.2)\times10^{12}$ | Under dry conditions, $\gamma_0$ decreased with temperature, from $(2.71\pm0.54)\times10^{-4}$ at 253 K to $(1.47\pm0.29)\times10^{-4}$ at 313 K. | KC-MS |
| Saharan dust | Pradhan et al., 2012b | 295±2 | $\sim4.2\times10^{12}$ | $\gamma(H_2O_2)$ increased from $(6.20\pm0.22)\times10^{-4}$ at 15% RH to $(9.42\pm0.41)\times10^{-4}$ at 70% RH. | AFT-CIMS |
| Gobi dust | Pradhan et al., 2012b | 295±2 | $\sim4.2\times10^{12}$ | $\gamma(H_2O_2)$ increased from $(3.33\pm0.26)\times10^{-4}$ at 15% RH to $(6.03\pm0.42)\times10^{-4}$ at 70% RH. | AFT-CIMS |
| Chinese dust | Zhou et al., 2016 | 253-313 | $(0.26\text{-}1.2)\times10^{12}$ | Under dry conditions, $\gamma_0$ decreased with temperature, from $(3.56\pm0.71)\times10^{-4}$ at 253 K to $(2.19\pm0.44)\times10^{-4}$ at 313 K for Inner Mongolia desert dust and | KC-MS |





| MgO | Wang et al., 2011 | 298 | $(1\text{-}25)\times10^{11}$ | $\gamma_0$: $(1.66\pm0.23)\times10^{-4}$; $\gamma_{ss}$: $1.6\times10^{-5}$. | from $(7.34\pm1.47)\times10^{-4}$ at 268 K to $(4.46\pm0.89)\times10^{-4}$ at 313 K for Xinjiang sierozem. | KC-MS |



A coated rod flow tube was coupled to a quadrupole mass spectrometer to investigate
heterogeneous reactions of $H_2O_2$ with a variety of mineral dust particles as a function of initial
$H_2O_2$ concentrations, irradiance intensity, RH, and temperature (Romanias et al., 2012b;
Romanias et al., 2013; El Zein et al., 2014). Under dark conditions, quick surface deactivation
was observed for $TiO_2$. When $[H_2O_2]_0$ was $<1\times10^{12}$ molecule $cm^{-3}$, $\gamma_0$ was found to be
independent of $[H_2O_2]_0$; however, when $[H_2O_2]_0$ was above this threshold, a negative
dependence of $\gamma_0$ on $[H_2O_2]_0$ occurred. At 275 K, $\gamma_0$ (based on BET surface area) depended on
RH (up to 82%), given by (Romanias et al., 2012b):
$$\gamma_0(dark) = 4.1 \times 10^{-3}/(1 + RH^{0.65}) \quad (9)$$
The uncertainty was estimated to be ±30%.
UV illumination could lead to photocatalytic decomposition of $H_2O_2$ on $TiO_2$ surface.
The steady state uptake coefficient, $\gamma_{ss}(UV)$, increasing linearly with illumination intensity, was
found to be independent of RH and depended inversely on $[H_2O_2]_0$ (Romanias et al., 2012b).
When $[H_2O_2]_0$ is $\sim5\times10^{11}$ molecule $cm^{-3}$ and $J(NO_2)$ for UV illumination is 0.012 $s^{-1}$, the
dependence of $\gamma_{ss}(UV)$ on temperature (275-320 K) at 0.3% RH can be described by (Romanias
et al., 2012b):
$$\gamma_{ss}(UV) = (7.2 \pm 1.9) \times 10^{-4} \times \exp[(460 \pm 80)/T] \quad (10)$$
It has also been found that NO added into the gas flow was converted to $NO_2$ during
heterogeneous reaction of $H_2O_2$ with $TiO_2$. As shown in Figure 7, the ratio of consumed NO to
formed $NO_2$ is close to 1. This indirect evidence suggests that $HO_2$ radicals (which could
convert NO to $NO_2$) were found in the gas phase due to photocatalytic reaction of $H_2O_2$ with
$TiO_2$ particles (Romanias et al., 2012b).





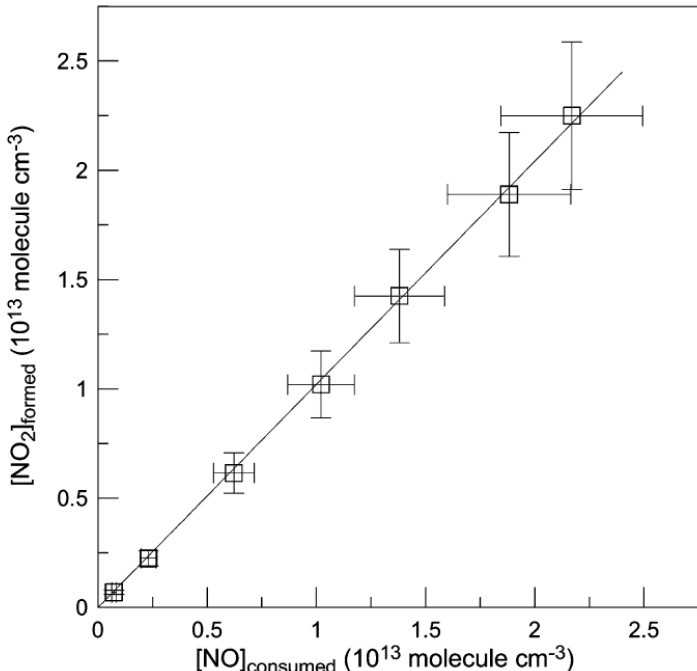


**Figure 7.** Consumed NO versus formed $NO_2$ in the heterogeneous reaction of $H_2O_2$ with $TiO_2$
particles under illumination. Reprinted with permission from Romanias et al. (2012a).
Copyright 2012 American Chemical Society.

Gradual surface deactivation was also observed for uptake of $H_2O_2$ by ATD particles.

$\gamma_0$, independent of $[H_2O_2]_0$ in the range of $(0.18\text{-}5.1) \times 10^{12}$ molecule $cm^{-3}$ and irradiation for
$J(NO_2)$ up to 0.012 $s^{-1}$, was observed to decrease with RH and temperature (El Zein et al., 2014).
At 275 K, the dependence of $\gamma_0$ on RH (up to 69%) can be described by (El Zein et al., 2014):

$$\gamma_0 = 4.8 \times 10^{-4}/(1 + RH^{0.66}) \quad (11)$$

At 0.35% RH, the effect of temperature on $\gamma_0$ is given by (El Zein et al., 2014):

$$\gamma_0 = 3.2 \times 10^{-4}/[1 + 2.5 \times 10^{10} \times \exp\left(-\frac{7360}{T}\right)] \quad (12)$$

It has also been found that $\gamma_{ss}$, independent of RH and $T$, decreased with $[H_2O_2]_0$ under dark
and irradiated conditions, given by (El Zein et al., 2014):





$$\gamma_{ss}(dark) = 3.8 \times 10^{-5} \times ([H_2O_2]_0)^{-0.6} \quad (13)$$
UV irradiation could enhance heterogeneous reactivity of ATD towards $H_2O_2$. For example,
when $J(NO_2)$ was equal to 0.012 s$^{-1}$, $\gamma_{ss}$(dark) and $\gamma_{ss}$(UV) were determined to be
$(0.95\pm0.30)\times10^{-5}$ and $(1.85\pm0.55)\times10^{-5}$, respectively (El Zein et al., 2014).
Romanias et al. (2013) examined heterogeneous interactions of $H_2O_2$ with $\gamma$-$Al_2O_3$ and
$Fe_2O_3$, and found that both surfaces were gradually deactivated after exposure to $H_2O_2$. $\gamma_0$,
independent of $[H_2O_2]_0$ in the range of $(0.15\text{-}16.6)\times10^{12}$ molecule cm$^{-3}$, was found to vary with
RH and temperature (Romanias et al., 2013). At 280 K, the dependence of $\gamma_0$ on RH (up to
73%) can be given by
$$\gamma_0(Al_2O_3) = 1.10 \times 10^{-3}/(1 + RH^{0.93}) \quad (14)$$
$$\gamma_0(Fe_2O_3) = 1.05 \times 10^{-3}/(1 + RH^{0.73}) \quad (15)$$
At 0.3% RH, the dependence of $\gamma_0$ on temperature ($T$) in the range of 268-320 K can be
described by:
$$\gamma_0(Al_2O_3) = 8.7 \times 10^{-4}/[1 + 5.0 \times 10^{13} \times \exp(-9700/T)] \quad (16)$$
$$\gamma_0(Fe_2O_3) = 9.3 \times 10^{-4}/[1 + 3.6 \times 10^{14} \times \exp(-10300/T)] \quad (17)$$
In contract to $TiO_2$ and ATD, no significant effects of UV irradiation with $J(NO_2)$ up to
0.012 s$^{-1}$ were observed for $\gamma$-$Al_2O_3$ and $Fe_2O_3$ (Romanias et al., 2013).
**3.2.1 Discussion of previous laboratory studies**
The dependence of $\gamma(H_2O_2)$ on RH, measured at room temperature, is plotted in Figure
8 for different dust particles. Uptake coefficients reported by Zhao et al. (2011b) are several
orders of magnitude smaller than those reported by other studies, and therefore they are not
included in Figure 8. For studies using dust particles supported on substrates, $\gamma_0(H_2O_2)$ are
plotted.





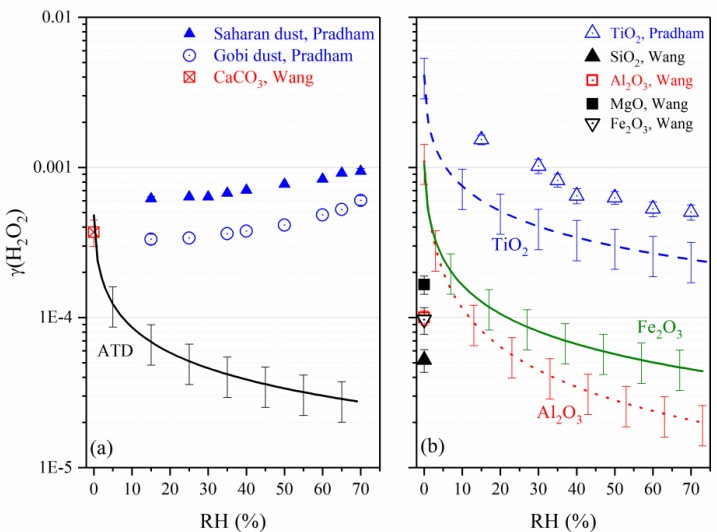


**Figure 8.** RH dependence of $\gamma(H_2O_2)$ for mineral dust particles as reported by previous studies

(Pradhan et al., 2010a; Pradhan et al., 2010b; Wang et al., 2011; Romanias et al., 2012b;

Romanias et al., 2013; El Zein et al., 2014).


Figure 8 suggests that different minerals show various heterogeneous reactivity towards

$H_2O_2$, and the effects of RH also appear to be different. Two previous studies have investigated

heterogeneous uptake of $H_2O_2$ by $TiO_2$ at different RH under dark conditions, one using an

aerosol flow tube (Pradhan et al., 2010a) and the other using coated rod flow tube (Romanias

et al., 2012b). For $TiO_2$, $\gamma(H_2O_2)$ reported by Romanias et al. (2012a) are around 40-50% of

those determined by Pradhan et al. (2010a) over 10-75% RH. The agreement is quite good

considering the fact that two very different techniques were used. Wang et al. (2011) and

Romanias (2013) examined heterogeneous reactions of $H_2O_2$ with $Fe_2O_3$ and $Al_2O_3$. Their

reported $\gamma_0(H_2O_2)$ differ significantly, though BET surface area was used by both studies to

calculate uptake coefficients. This may be largely explained by the variation of interrogation

depth of $H_2O_2$ molecules under investigation in different studies, as discussed in Section 2.2.1.





Experiments in which aerosol samples are used can largely overcome the difficulty in
estimating surface area available for heterogeneous uptake. Up to now only two studies
(Pradhan et al., 2010a; Pradhan et al., 2010b) used aerosol flow tubes, and more aerosol flow
tube studies will help better constrain $\gamma(H_2O_2)$ onto mineral dust particles.

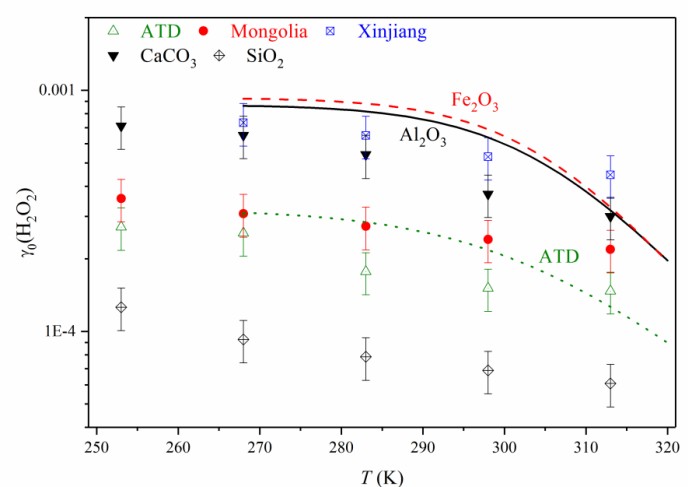


**Figure 9.** Temperature dependence of $\gamma_0(H_2O_2)$ for mineral dust particles under dark conditions
as reported by previous studies. Upward triangles: ATD (Zhou et al., 2016); circles: Inner
Mongolia desert dust (Zhou et al., 2016); squares: Xinjiang sierozem (Zhou et al., 2016);
downward triangles: $CaCO_3$ (Zhou et al., 2012); diamonds: $SiO_2$ (Zhou et al., 2012); olive
circle: ATD (El Zein et al., 2014); solid black curve: $Al_2O_3$ (Romanias et al., 2013); dashed red
curve: $Fe_2O_3$ (Romanias et al., 2013).

The effects of temperature on heterogeneous reactions of $H_2O_2$ with mineral dust have
also been explored. As shown in Figure 9, $\gamma_0(H_2O_2)$ decrease with increasing temperature. Zhou
et al. (2012, 2016) suggest that $\gamma_0(H_2O_2)$ are reduced by a factor of ~2 for all the five minerals
they investigated when temperature increases from 253 K to 313 K. Romanias et al. (2013) and
El Zein et al. (2014) reported larger temperature impacts, with $\gamma_0(H_2O_2)$ reduced by a factor of





~4 when temperature increases from 268 to 320 K. These studies show that the temperature
effect is significant and should be taken into account when assessing the importance of
heterogeneous uptake of $H_2O_2$ by mineral dust in the troposphere. It should also be pointed out
that the effect of temperature on heterogeneous reactions of $H_2O_2$ with airborne mineral dust
particles has never been investigated.

In addition, it has been suggested that uptake of $H_2O_2$ by mineral dust can affect

heterogeneous oxidation of other trace gases (Zhao et al., 2011b; Zhao et al., 2013; Huang et
al., 2015a). For examples, heterogeneous uptake of $H_2O_2$ could convert sulfite formed by the
adsorption of $SO_2$ on $CaCO_3$ particles to sulfate, and this conversion is enhanced by adsorbed
water (Zhao et al., 2013). Similarly, Huang et al. (2015a) found that the presence of $H_2O_2$ could
enhance the uptake of $SO_2$ on Asian mineral dust, Tengger desert dust, and ATD, and the
enhancement factors, varying with dust mineralogy and RH, can be as large as a factor of ~6.
Heterogeneous oxidation of methacrolein on kaolinite, $\alpha$-$Al_2O_3$, $\alpha$-$Fe_2O_3$, and $TiO_2$ (but not on
$CaCO_3$) is largely accelerated by the presence of $H_2O_2$, which also changes the oxidation
products (Zhao et al., 2011b).
**3.2.2 Atmospheric implication**

For reasons we have discussed in Section 2.2.1, $\gamma(H_2O_2)$ reported by studies using

aerosol samples (Pradhan et al., 2010a; Pradhan et al., 2010b) are preferred. Since Saharan dust
is the most abundant mineral dust particles in the troposphere, in our work we use $\gamma(H_2O_2)$
reported by Pradhan et al. (2010b) for Saharan dust to assess the atmospheric importance of
heterogeneous uptake of $H_2O_2$. $\gamma(H_2O_2)$ onto Saharan dust depends on RH, increasing from
$6.2\times10^{-4}$ at 15% to $9.4\times10^{-4}$ at 70% RH. For simplicity, a $\gamma(H_2O_2)$ value of $1\times10^{-3}$, very close
to that at 70%, is used here to calculate $\tau_{het}(H_2O_2)$. When dust mass concentrations are 10, 100,
and 1000 $\mu g$ $m^{-3}$, $\tau_{het}(H_2O_2)$ are calculated to be 120, 12, and 1.2 h, using Eq. (6). Typical
$\tau(H_2O_2)$ are estimated to be 33-56 h with respect to photolysis and 16-160 h with respect to





reaction with OH radicals. Therefore, heterogeneous uptake by mineral dust particles can be a
significant sink for $H_2O_2$ when dust mass concertation is as low as 10 µg m$^{-3}$.
Several modelling studies have also discussed and evaluated the contribution of
heterogeneous uptake by mineral dust to the removal of $H_2O_2$ in the troposphere. Pradhan et al.
(2010b) determined $\gamma(H_2O_2)$ for Saharan dust as a function of RH experimentally and then
included this reaction in a box model based on the MCM. It has been found that heterogeneous
uptake by mineral dust could reduce simulated $H_2O_2$ concentrations by up to ~40%, and its
impacts on total peroxy organic radicals, OH, $O_3$, and NOx are small but non-negligible
(Pradhan et al., 2010b). In another box model study, $\gamma(H_2O_2)$ onto Saharan dust was varied in
order to reproduce $H_2O_2$ concentrations measured in July/August 2002 at Tenerife (de Reus et
al., 2005). It is found that using $\gamma(H_2O_2)$ of $5 \times 10^{-4}$, which agrees very well with these measured
by Pradhan et al. (2010b), could reach the best agreement between measured and simulated
$H_2O_2$ concentrations (de Reus et al., 2005).
In addition to the uncertainties in $\gamma(H_2O_2)$ related to the effects of mineralogy, RH, and
temperature, products formed in heterogeneous reactions of $H_2O_2$ with mineral dust are not
entirely clear. Three pathways have been proposed, including i) simple partitioning of $H_2O_2$
onto dust particles (Zhao et al., 2011b; Zhao et al., 2013), ii) surface decomposition of $H_2O_2$ to
$H_2O$ and $O_2$, and iii) heterogeneous conversion of $H_2O_2$ to $HO_2$ radicals (Romanias et al., 2012b;
Yi et al., 2012). Branching ratios seem to depend on mineralogy, RH, and probably also UV
illumination (Zhao et al., 2011b; Yi et al., 2012; Zhao et al., 2013); however, our knowledge
in this aspect is very limited. Since these three different pathways may have very different
impacts on tropospheric oxidation capacity, product distribution in heterogeneous reactions of
$H_2O_2$ with mineral dust deserves further investigation.





### 3.3 $O_3$


Heterogeneous reactions of $O_3$ with $Al_2O_3$, $CaCO_3$, and Saharan dust were explored
using a fluidized bed reactor more than two decades ago, and substantial $O_3$ decays were
observed after interactions with dust power in the reactor (Alebić-Juretić et al., 1992). This
study did not report uptake coefficients and thus is not included in Table 4. Uptake coefficients
in the range of $(1-100)\times10^{-11}$ were reported for $Al_2O_3$ (Hanning-Lee et al., 1996). Since their
experiments were carried out with $O_3$ concentrations in the range of $(5-200)\times10^{15}$ molecule
$cm^{-3}$ which are several orders of magnitude higher than typical $O_3$ levels in the troposphere,
this work is also not included in Table 4.
A Knudsen cell reactor was used by Grassian and co-workers (Michel et al., 2002;
Michel et al., 2003; Usher et al., 2003b) to study heterogeneous reactions of $O_3$ with fresh and
aged mineral dust particles. Measurements were carried out in the linear mass dependent
regime (see Section 2.2.1 for more explanations of the linear mass dependent regime), and thus
the BET surface areas of dust samples were used to calculate uptake coefficients. In the first
study (Michel et al., 2002), $\gamma_0(O_3)$ was determined to be $(1.8\pm0.7)\times10^{-4}$ for $\alpha$-$Fe_2O_3$, $(8\pm5)\times10^{-5}$
for $\alpha$-$Al_2O_3$, $(5\pm3)\times10^{-5}$ for $SiO_2$, $(2.7\pm0.9)\times10^{-5}$ for China loess, $(6\pm3)\times10^{-5}$ for ground
Saharan dust, and $(4\pm2)\times10^{-6}$ for sieved Saharan dust at 296 K when $[O_3]_0$ was $1.9\times10^{11}$
molecule $cm^{-3}$. In a following study, Michal et al. (2003) systematically investigated
heterogeneous reactions of $O_3$ with several mineral dust particles, and progressive surface
deactivation was observed for all the dust samples. At $295\pm1$ K and $[O_3]_0$ of $(1.9\pm0.6)\times10^{11}$
molecule $cm^{-3}$, $\gamma_0(O_3)$ were reported to be $(2.0\pm0.3)\times10^{-4}$ for $\alpha$-$Fe_2O_3$, $(1.2\pm0.4)\times10^{-4}$ for 25
$\mu m$ $\alpha$-$Al_2O_3$, $(6.3\pm0.9)\times10^{-5}$ for $SiO_2$, $(3\pm1)\times10^{-5}$ for kaolinite, $(2.7\pm0.8)\times10^{-5}$ for China loess,
$(6\pm2)\times10^{-5}$ for ground Saharan dust, and $(2.7\pm0.9)\times10^{-6}$ for ground Saharan dust, respectively.
$\gamma_0(O_3)$ was also measured for 1 $\mu m$ $\alpha$-$Al_2O_3$, and with the experimental uncertainties it shows
no difference with that for 25 $\mu m$ $\alpha$-$Al_2O_3$. The steady-state uptake coefficients, $\gamma_{ss}$, were





determined to be $2.2 \times 10^{-5}$ for α-$Fe_2O_3$, $7.6 \times 10^{-6}$ for α-$Al_2O_3$, and $6 \times 10^{-6}$ for ground Saharan
dust. The effect of initial $O_3$ concentration in the range of $(1-10) \times 10^{11}$ molecule $cm^{-3}$ on $\gamma_0(O_3)$
is insignificant for either α-$Al_2O_3$ or α-$Fe_2O_3$. In addition, $\gamma_0(O_3)$ was found to have a very weak
dependence on temperature (250-330 K) for α-$Al_2O_3$, with an activation energy of $7\pm4$ kJ $mol^{-1}$
(Michel et al., 2003).

Heterogeneous processing of mineral dust particles by other trace gases could affect $O_3$

uptake. It has been observed that $\gamma_0(O_3)$ was reduced by ~70% after pretreatment of α-$Al_2O_3$
with $HNO_3$ and increased by 33% after pretreatment with $SO_2$ (Usher et al., 2003b). Similarly,
functionalization of $SiO_2$ with a C8 alkene would increase its heterogeneous reactivity towards
$O_3$ by 40% whereas its heterogeneous reactivity was reduced by about 40% if functionalized
by a C8 alkane (Usher et al., 2003b). The presence of $O_3$ can also promote heterogeneous
oxidation of other trace gases on mineral dust surface (Ullerstam et al., 2002; Hanisch and
Crowley, 2003b; Li et al., 2006; Chen et al., 2008; Wu et al., 2011), including NO, $SO_2$,
methacrolein, methyl vinyl ketone, and etc.





**Table 6:** Summary of previous laboratory studies on heterogeneous reactions of mineral dust with $O_3$

| Dust | Reference | T (K) | Concentration (molecule cm$^{-3}$) | Uptake coefficient | Techniques |
|---|---|---|---|---|---|
| $Al_2O_3$ | Michel et al., 2002 | 296 | $1.9\times10^{11}$ | $\gamma_0$: $(8\pm5)\times10^{-5}$ | KC-MS |
| | Michel et al., 2003 | 250-330 | $(1\text{-}10)\times10^{11}$ | At 296 K, $\gamma_0$ was determined to be $(1.2\pm0.4)\times10^{-4}$ and $\gamma_{ss}$ was determined to be $7.6\times10^{-6}$. A very weak temperature dependence was observed. | KC-MS |
| | Usher et al., 2003b | $295\pm1$ | $1.9\times10^{11}$ | Compared to fresh particles, $\gamma_0$ were reduced by 72% to $(3.4\pm0.6)\times10^{-5}$ when the surface coverage of $HNO_3$ was $(6\pm3)\times10^{14}$ molecule cm$^{-2}$ and increased by 33% to $(1.6\pm0.2)\times10^{-4}$ when the surface coverage of $SO_2$ was $(1.5\pm0.3)\times10^{14}$ molecule cm$^{-2}$. | KC-MS |
| | Sullivan et al., 2004 | room temperature | $(1\text{-}10)\times10^{13}$ | $\gamma(O_3)$ decreased from $\sim1\times10^{-5}$ to $\sim1\times10^{-6}$ when initial $O_3$ concentration increased from $1\times10^{13}$ to $1\times10^{14}$ molecule cm$^{-3}$. | static reactor |
| | Mogili et al., 2006a | room temperature | $1\times10^{15}$ | $\gamma(O_3)$ decreased from $(3.5\pm0.9)\times10^{-8}$ at <1% RH to $(4.5\pm1.1)\times10^{-9}$ at 19% RH. | EC |
| | Chen et al., 2011a | room temperature | $\sim1.9\times10^{15}$ | Irradiation from a solar simulation could enhance $O_3$ uptake by $\alpha$-$Al_2O_3$, but no uptake coefficient was reported. | EC |
| | Chen et al., 2011b | room temperature | $(2\text{-}3)\times10^{15}$ | Uptake of $O_3$ by $\alpha$-$Al_2O_3$ was insignificant under both dark and irradiated conditions. | EC |





Atmospheric Chemistry and Physics Discussions — Open Access — EGU

| | Reference | Temperature | | Description | Method |
|---|---|---|---|---|---|
| Saharan dust | Michel et al., 2002 | 296 | $1.9\times10^{11}$ | $\gamma_0$ was determined to be $(6\pm3)\times10^{-5}$ for ground Saharan dust and $(4\pm2)\times10^{-6}$ for sieved Saharan dust. | KC-MS |
| | Hanisch and Crowley, 2003 | 296 | $(0.54\text{-}84)\times10^{11}$ | $\gamma_0 = 3.5\times10^{-4}$ and $\gamma_{ss} = 4.8\times10^{-5}$ when $[O_3]_0 = 5.4\times10^{10}$ molecule cm$^{-3}$; $\gamma_0 = 5.8\times10^{-5}$ and $\gamma_{ss} = 1.3\times10^{-5}$ when $[O_3]_0 = 2.8\times10^{11}$ molecule cm$^{-3}$; $\gamma_0 = 5.5\times10^{-6}$ and $\gamma_{ss} = 2.2\times10^{-6}$ when $[O_3]_0 = 8.4\times10^{12}$ molecule cm$^{-3}$. | KC-MS |
| | Michel et al., 2003 | 295±1 | $(1.9\pm0.6)\times10^{11}$ | For ground Saharan dust, $\gamma_0$: $(6\pm2)\times10^{-5}$ and $\gamma_{ss}$: $6\times10^{-6}$. For sieved Saharan dust, $\gamma_0$: $(2.7\pm0.9)\times10^{-6}$. | KC-MS |
| | Chang et al., 2005 | room temperature | $(0.2\text{-}10)\times10^{13}$ | $\gamma(O_3)$ decreased from $6\times10^{-6}$ to $\sim2\times10^{-7}$ when $[O_3]$ increased from $2\times10^{12}$ to $1\times10^{14}$ molecule cm$^{-3}$. | static reactor |
| | Karagulian and Rossi, 2006 | 298±2 | $(3.5\text{-}10)\times10^{12}$ | $\gamma_0 = (9.3\pm2.6)\times10^{-2}$ and $\gamma_{ss} = (6.7\pm1.3)\times10^{-3}$ when $[O_3]_0 = 3.5\times10^{12}$ molecule cm$^{-3}$; $\gamma_0 = (3.7\pm1.8)\times10^{-3}$ and $\gamma_{ss} = (3.3\pm2.5)\times10^{-3}$ when $[O_3]_0 = 1.0\times10^{13}$ molecule cm$^{-3}$. Reported uptake coefficients were based on the projected surface area. | KC-MS |
| $Fe_2O_3$ | Michel et al., 2002 | 296 | $1.9\times10^{11}$ | $\gamma_0$: $(1.8\pm0.7)\times10^{-4}$ | KC-MS |
| | Michel et al., 2003 | 295±1 | $(1\text{-}10)\times10^{11}$ | $\gamma_0$: $(2.0\pm0.3)\times10^{-4}$; $\gamma_{ss}$: $2.2\times10^{-5}$ | KC-MS |
| | Mogili et al., 2006a | room temperature | $(1.8\text{-}8.5)\times10^{14}$ | When $[O_3]_0$ was $7.9\times10^{14}$ molecule cm$^{-3}$, $\gamma(O_3)$ decreased from $(1.0\pm0.3)\times10^{-7}$ at <1% RH to $(1.2\pm0.3)\times10^{-8}$ at 23% RH and to $(2.5\pm0.6)\times10^{-9}$ at 58% RH. When $[O_3]_0$ was $2.1\times10^{14}$ molecule cm$^{-3}$, | EC |



| Material | Reference | Temperature | Value | Result | Method |
|---|---|---|---|---|---|
| | Chen et al., 2011a | room temperature | $\sim 1.9 \times 10^{15}$ | $\gamma(O_3)$ decreased from $(5.0\pm1.2)\times10^{-8}$ at <1% RH to $(2.0\pm0.5)\times10^{-8}$ at 21% RH and to $(9.0\pm2.3)\times10^{-9}$ at 43% RH. Irradiation from a solar simulation could enhance the $O_3$ uptake by $\alpha$-$Fe_2O_3$, but no uptake coefficient was reported. | EC |
| | Chen et al., 2011b | room temperature | $(2\text{-}3)\times10^{15}$ | Under dark conditions, $\gamma(O_3)$ decreased from $(4.1\pm0.2)\times10^{-7}$ at <2% RH to $(2.7\pm0.1)\times10^{-7}$ at 21% RH. When irradiated, $\gamma(O_3)$ decreased from $(6.6\pm0.3)\times10^{-7}$ at <2% RH to $(5.5\pm0.3)\times10^{-7}$ at 12% RH and to $(1.1\pm0.1)\times10^{-7}$ at 25% RH. | EC |
| $SiO_2$ | Michel et al., 2002 | 296 | $1.9\times10^{11}$ | $\gamma_0$: $(5\pm3)\times10^{-5}$ | KC-MS |
| | Michel et al., 2003 | 295±1 | $(1.9\pm0.6)\times10^{11}$ | $\gamma_0$: $(6.3\pm0.9)\times10^{-5}$ | KC-MS |
| | Usher et al., 2003b | 295±1 | $1.9\times10^{11}$ | Compared to fresh particles, $\gamma_0$ was increased by 40% to $(7\pm2)\times10^{-5}$ when the surface coverage of a C8 alkene was $(2\pm1)\times10^{14}$ molecule cm$^{-2}$ and reduced by 40% to $(3\pm1)\times10^{-5}$ when the surface coverage of a C8 alkane was $(2\pm1)\times10^{14}$ molecule cm$^{-2}$. | KC-MS |
| | Nicolas et al., 2009 | 298 | $(1.3\text{-}7.3)\times10^{12}$ | $\gamma(O_3)$ was found to be $<1\times10^{-8}$, showing negative dependence on $[O_3]_0$ and RH. No difference in $\gamma(O_3)$ under dark and illuminated conditions was reported. | CWFT |
| China loess | Michel et al., 2002 | 296 | $1.9\times10^{11}$ | $\gamma_0$: $(2.7\pm0.9)\times10^{-5}$ | KC-MS |
| | Michel et al., 2003 | 295±1 | $(1.9\pm0.6)\times10^{11}$ | $\gamma_0$: $(2.7\pm0.8)\times10^{-5}$ | KC-MS |





| | | | | | |
|---|---|---|---|---|---|
| kaolinite | Michel et al., 2003 | 295±1 | $(1.9\pm0.6)\times10^{11}$ | $\gamma_0$: $(3\pm1)\times10^{-5}$ | KC-MS |
| | Karagulian and Rossi, 2006 | 298±2 | $(2.4\pm0.7)\times10^{12}$ | Projected surface area based: $\gamma_0 = (6.3\pm0.2)\times10^{-2}$ and $\gamma_{ss} = (1.0\pm0.2)\times10^{-2}$; pore diffusion corrected $\gamma_{ss}$: $(2.7\pm0.3)\times10^{-6}$. | KC-MS |
| CaCO$_3$ | Karagulian and Rossi, 2006 | 298±2 | $(5.3\pm0.7)\times10^{12}$ | Projected surface area based: $\gamma_0 = (1.2\pm0.3)\times10^{-2}$ and $\gamma_{ss} = (3.6\pm0.2)\times10^{-3}$; pore diffusion corrected $\gamma_{ss}$: $(7.8\pm0.7)\times10^{-7}$. | KC-MS |
| TiO$_2$ | Nicolas et al., 2009 | 298 | $(1.3\text{-}7.3)\times10^{12}$ | $\gamma(O_3)$ on TiO$_2$/SiO$_2$ decreased with [O$_3$]$_0$ and RH under both dark and illuminated conditions. Under illuminated conditions it increased with TiO$_2$ mass fraction in TiO$_2$/SiO$_2$ and depended almost linearly on irradiance intensity. At 24% RH and [O$_3$]$_0$ of 51 ppbv, $\gamma(O_3)$ on 1 wt% TiO$_2$/SiO$_2$ was reported to be $(2.8\pm0.4)\times10^{-9}$ under dark conditions and $(4.7\pm0.7)\times10^{-8}$ under a near UV irradiance of $3.2\times10^{-8}$ mW cm$^{-2}$. | CWFT |
| | Chen et al., 2011b | room temperature | $(2\text{-}3)\times10^{15}$ | Uptake of O$_3$ was negligible under dark conditions. Under the irradiation of a solar simulator, $\gamma(O_3)$ was determined to be $(2.0\pm0.1)\times10^{-7}$ at <2% RH, $(2.2\pm0.1)\times10^{-7}$ at 12% RH, $(2.4\pm0.1)\times10^{-7}$ at 22% RH, and $(1.9\pm0.1)\times10^{-7}$ at 39% RH, respectively. | EC |
| ATD | Karagulian and Rossi, 2006 | 298±2 | $(3.3\text{-}8.0)\times10^{12}$ | $\gamma_0 = (1.3\pm0.6)\times10^{-2}$ and $\gamma_{ss} = (2.2\pm1.2)\times10^{-3}$ when [O$_3$]$_0$= $3.3\times10^{12}$ molecule cm$^{-3}$; $\gamma_0 = (1.3\pm0.7)\times10^{-2}$ and $\gamma_{ss} = (2.5\pm1.2)\times10^{-3}$ when [O$_3$]$_0$= $8\times10^{12}$ molecule cm$^{-3}$. Reported uptake coefficients were based on the projected surface area. | KC-MS |



| limestone | Karagulian and Rossi, 2006 | 298±2 | (3–20)×10$^{12}$ | $\gamma_0 = (1.3\pm0.2)\times10^{-2}$ and $\gamma_{ss} = (1.6\pm0.5)\times10^{-3}$ when $[O_3]_0 = 3\times10^{12}$ molecule cm$^{-3}$; $\gamma_0 = (2.1\pm0.3)\times10^{-3}$ and $\gamma_{ss} = (2.4\pm0.7)\times10^{-4}$ when $[O_3]_0 = 2\times10^{13}$ molecule cm$^{-3}$. Reported uptake coefficients were based on the projected surface area. | KC-MS |




Another two groups also utilized Knudsen cell reactors to investigate $O_3$ uptake by
mineral dust (Hanisch and Crowley, 2003a; Karagulian and Rossi, 2006). The uptake of $O_3$ by
Saharan dust was investigated over a broad range of $[O_3]_0$ by Hanisch and Crowley (2003), and
$\gamma_0(O_3)$ and $\gamma_{ss}(O_3)$ were determined to be $3.5\times10^{-4}$ and $4.8\times10^{-5}$ when $[O_3]_0$ was $(5.4\pm0.8)\times10^{10}$
molecule $cm^{-3}$, $5.8\times10^{-5}$ and $1.3\times10^{-5}$ when $[O_3]_0$ was $2.8\times10^{11}$ molecule $cm^{-3}$, and $5.5\times10^{-6}$
and $2.2\times10^{-4}$ when $[O_3]_0$ was $(8.4\pm3.4)\times10^{12}$ molecule $cm^{-3}$, showing a negative dependence
on $[O_3]_0$. It should be noted that the KML model (Keyser et al., 1991; Keyser et al., 1993) was
applied by Hanisch and Crowley (2003) to derive the uptake coefficients. Furthermore, they
found that $O_3$ was converted to $O_2$ after reaction with Saharan dust and physisorption was
negligible (Hanisch and Crowley, 2003a).
Karagulian and Rossi et al. (2006) investigated heterogeneous interactions of $O_3$ with
kaolinite, $CaCO_3$, natural limestone, Saharan dust, and ATD. Based on the projected surface
areas of dust samples, their reported $\gamma_0$ are in the range of $(2.3\pm0.4)\times10^{-2}$ to $(9.3\pm2.6)\times10^{-2}$ and
$\gamma_{ss}$ are in the range of $(3.5\pm1.6)\times10^{-5}$ to $(1.0\pm0.2)\times10^{-2}$. These values, summarized in Table 4
together with corresponding $[O_3]_0$, are not repeated here. Pore diffusion corrected $\gamma_{ss}$ were
reported to be $(2.7\pm0.3)\times10^{-6}$ for kaolinite when $[O_3]_0$ was $2.4\times10^{12}$ molecule $cm^{-3}$ and
$(7.8\pm0.7)\times10^{-7}$ for $CaCO_3$ when $[O_3]_0$ was $5.3\times10^{12}$ molecule $cm^{-3}$, more than three orders of
magnitude smaller than those based on the projected surface area (Karagulian and Rossi, 2006).
The uptake of $O_3$ on $\alpha$-$Al_2O_3$ (Sullivan et al., 2004) and Saharan dust (Chang et al.,
2005) was investigated using a static reactor, in which a dust-coated Pyrex tube was exposed
to $O_3$ at room temperature. In the first few tens of seconds after exposure to dust particles, $O_3$
decays followed an exponential manner, and the average decay rates were used to derive uptake
coefficients. $\gamma(O_3)$, based on the BET surface area, was found to decrease with increasing initial
$[O_3]$. For $\alpha$-$Al_2O_3$, $\gamma(O_3)$ decreased from $\sim1\times10^{-5}$ to $\sim1\times10^{-6}$ when $[O_3]$ increased from $1\times10^{13}$
to $1\times10^{14}$ molecule $cm^{-3}$ (Sullivan et al., 2004). For Saharan dust, $\gamma(O_3)$ decreased from $2\times10^{-7}$



to $2 \times 10^{-6}$ for Saharan dust when [$O_3$] increased from $2 \times 10^{12}$ to $1 \times 10^{14}$ molecule $cm^{-3}$, and the
dependence of $\gamma(O_3)$ on [$O_3$] can be described by Eq. (18) (Chang et al., 2005):

$$\gamma(O_3) = 7.5 \times 10^5 \times [O_3]^{-0.90} \quad (18)$$

where [$O_3$] is the $O_3$ concentration in molecule $cm^{-3}$. No significant effect of RH (0-75%) on
uptake kinetics was observed for $\alpha$-$Al_2O_3$ and Saharan dust (Sullivan et al., 2004; Chang et al.,

2005).

An environmental chamber in which $O_3$ was exposed to suspended particles was

deployed to investigate heterogeneous reactions of airborne mineral dust with $O_3$ under dark
and illuminated conditions (Mogili et al., 2006a; Chen et al., 2011a; Chen et al., 2011b). $O_3$
concentrations in the chamber, detected using FTIR or UV/Vis absorption spectroscopy, were
found to decay exponentially with reaction time. As shown in Figure 10, uptake of $O_3$ by $\alpha$-
$Fe_2O_3$ was significantly suppressed at increasing RH, and a negative effect of RH was also
observed for uptake of $O_3$ by $\alpha$-$Al_2O_3$ (Mogili et al., 2006a). In addition, increasing [$O_3$]$_0$
resulted in reduction in $\gamma(O_3)$ for both minerals. Heterogeneous reactivity towards $O_3$ under
similar conditions is higher for $\alpha$-$Fe_2O_3$ when compared to $\alpha$-$Al_2O_3$ (Mogili et al., 2006a). For
$\alpha$-$Fe_2O_3$, when [$O_3$]$_0$ was $7.9 \times 10^{14}$ molecule $cm^{-3}$, $\gamma(O_3)$ decreased from $(1.0\pm0.3) \times 10^{-7}$ at <1%
RH to $(1.2\pm0.3) \times 10^{-8}$ at 23% RH and to $(2.5\pm0.6) \times 10^{-9}$ at 58% RH; when [$O_3$]$_0$ was $2.1 \times 10^{14}$
molecule $cm^{-3}$, $\gamma(O_3)$ was reduced from $(5.0\pm1.2) \times 10^{-8}$ at <1% RH to $(2.0\pm0.5) \times 10^{-8}$ at 21%
RH and to $(9.0\pm2.3) \times 10^{-9}$ at 43% RH. Meanwhile, $\gamma(O_3)$ was observed to decrease from
$(3.5\pm0.9) \times 10^{-8}$ at <1% RH to $(4.5\pm1.1) \times 10^{-9}$ at 19% RH for $\alpha$-$Al_2O_3$ when [$O_3$]$_0$ was $1 \times 10^{15}$
molecule $cm^{-3}$.





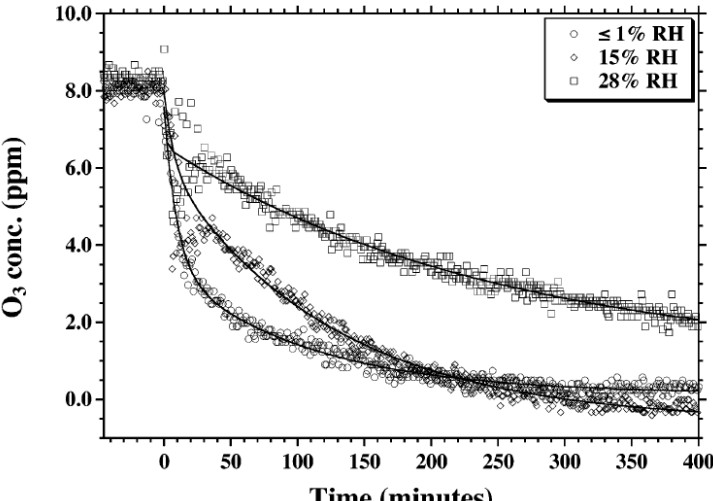


**Figure 10.** Measured $O_3$ decays in an aerosol chamber due to interaction with airborne α-$Fe_2O_3$
particles (starting at 0 min). The solid curves represent exponential fits to the measured $O_3$
concentrations as a function of reaction time. Reprinted with permission from Mogili et al.
(2006b). Copyright 2006 American Chemical Society.

A solar simulator was coupled to the environmental chamber by Chen et al. (2011a),
and irradiation from the solar simulator was found to enhance heterogeneous uptake of $O_3$ by
α-$Fe_2O_3$ and α-$Al_2O_3$; however, no uptake coefficient was reported. In a following study, Chen
et al. (2011b) found that heterogeneous uptake of $O_3$ by α-$Al_2O_3$ was insignificant under both
dark and irradiated conditions. In contract, while the uptake of $O_3$ by $TiO_2$ was negligible under
dark conditions, when irradiated $\gamma(O_3)$ was determined to be $(2.0\pm0.1)\times10^{-7}$ at <2% RH,
$(2.2\pm0.1)\times10^{-7}$ at 12% RH, $(2.4\pm0.1)\times10^{-7}$ at 22% RH, and $(1.9\pm0.1)\times10^{-7}$ at 39% RH,
respectively (Chen et al., 2011b). Photo-enhanced $O_3$ uptake was also observed for α-$Fe_2O_3$
(Chen et al., 2011b). Under dark conditions $\gamma(O_3)$ decreased from $(4.1\pm0.2)\times10^{-7}$ at <2% RH
to $(2.7\pm0.1)\times10^{-7}$ at 21% RH, while when irradiated $\gamma(O_3)$ was reported to be $(6.6\pm0.3)\times10^{-7}$ at
<2% RH, $(5.5\pm0.3)\times10^{-7}$ at 12% RH, and $(1.1\pm0.1)\times10^{-7}$ at 25% RH, respectively.



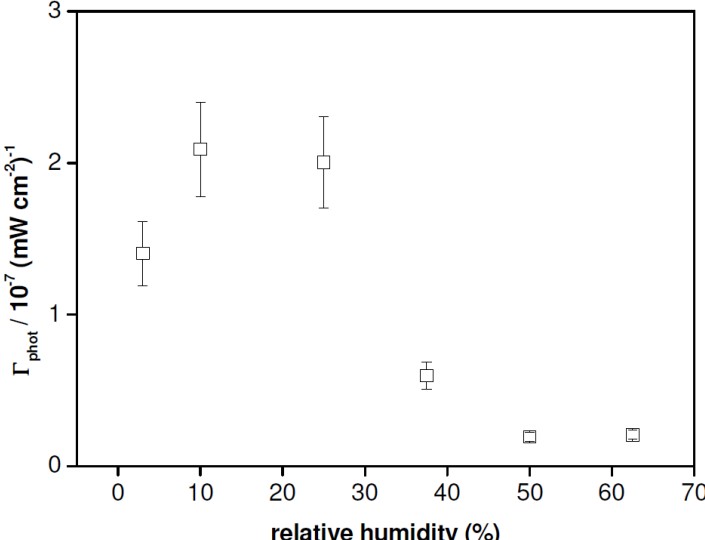


**Figure 11.** Effects of RH on the irradiance-normalized $O_3$ uptake coefficients. The $TiO_2/SiO_2$

films which contained 1 wt% $TiO_2$ were exposed to 37 ppbv $O_3$ at 298 K under irradiance of

$2.7 \times 10^{14}$ photons $cm^{-2}$ $s^{-1}$. Reprinted with permission from Nicolas et al. (2009). Copyright

2009 American Chemical Society.


Photo-enhanced catalytic decomposition of $O_3$ on mineral dust was in fact first reported

by a coated wall flow tube study at 298 K (Nicolas et al., 2009). Under their experimental
conditions ($[O_3]_0$: 50-290 ppbv; RH: 3-60%), the BET surface area based $\gamma_{ss}(O_3)$, was found to
be $<1 \times 10^{-8}$ for $SiO_2$ and $TiO_2/SiO_2$ mixture with $TiO_2$ mass fraction up to 5% under dark
conditions. Near UV irradiation could largely increase the uptake of $O_3$ by $TiO_2/SiO_2$ mixture,
and the effect increased with the $TiO_2$ mass fraction (the effect is insignificant for pure $SiO_2$)
and almost depended linearly on the intensity of UV irradiance (Nicolas et al., 2009). When
RH was 24% and $[O_3]_0$ was 51 ppbv, $\gamma(O_3)$ for $TiO_2/SiO_2$ mixture with a $TiO_2$ mass fraction of
1% was measured to be $(2.8 \pm 0.4) \times 10^{-9}$ under dark conditions and $(4.7 \pm 0.7) \times 10^{-8}$ under near
UV irradiation of $3.0 \times 10^{-8}$ mW $cm^{-2}$. RH was found to play a profound role in heterogeneous





photochemical reaction of $O_3$ with $TiO_2/SiO_2$. Figure 11 shows that the irradiance-normalized
uptake coefficient, defined as the uptake coefficient divided by the irradiance intensity,
increased with RH for RH <20% and then decreased significantly with RH when RH was
further increased. This phenomenon was also observed by Chen et al. (2011b), who found that
under illuminated conditions $\gamma(O_3)$ first increased and then decreased with RH for $TiO_2$ aerosol
particles.

**3.3.1 Discussion**

All the initial $\gamma(O_3)$ reported by previous studies for different minerals are summarized

in Figure 12 as a function of $[O_3]$. Karagulian and Rossi (2006) reported projected area based
$\gamma_0(O_3)$, which are several orders of magnitude larger than values reported by other work. This
is because $O_3$ uptake by mineral dust is relatively slow and some underlying dust layers, if not
all, must be accessible by $O_3$ molecules. Therefore, results reported by Karagulian and Rossi
(2006) are not included in Figure 12. Sullivan et al. (2004) and Chang et al. (2005) measured
$O_3$ decay rates in the first tens of seconds due to interaction with dust particles deposited onto
the inner wall of a Pyrex tube to derive $\gamma(O_3)$. Their reported $\gamma(O_3)$ are in fact the average uptake
coefficients in the first tens of seconds, and can be classified as either $\gamma_0(O_3)$ and $\gamma_{ss}(O_3)$.
Therefore, $\gamma(O_3)$ reported by Sullivan et al. (2004) and Chang et al. (2005) are included in
Figure 12 which summarizes $\gamma_0(O_3)$ and also in Figure 13 which summarizes $\gamma_{ss}(O_3)$.





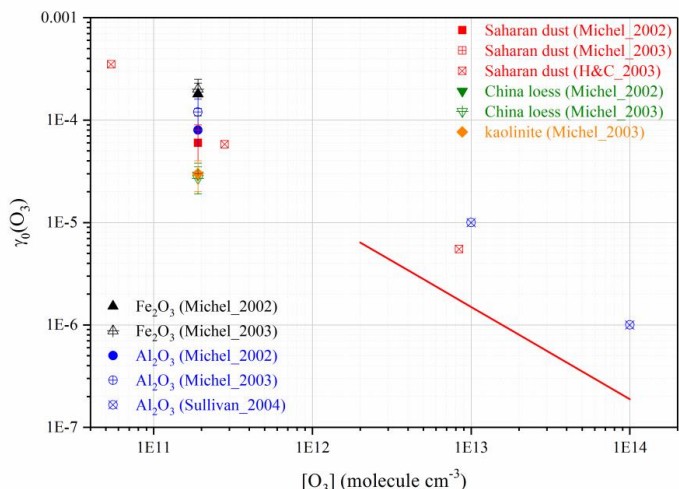


**Figure 12.** Dependence of $\gamma_0(O_3)$ on initial $O_3$ concentrations under dry conditions for different

mineral dust particles as reported by previous studies: Michel_2002 (Michel et al., 2002),

Michel_2003 (Michel et al., 2003), H&C_2003 (Hanisch and Crowley, 2003a), Sullivan_2004

(Sullivan et al., 2004). The red curve represents the dependence of $\gamma(O_3)$ on $[O_3]$ for Saharan

dust reported by Chang et al. (2005).

It should be noted that all the studies included in Figure 12 used dust powder samples

supported on substrates. Significant variation in reported $\gamma_0(O_3)$ is evident from Figure 12. For

examples, $\gamma_0(O_3)$ determined at $[O_3]$ of $\sim 2 \times 10^{11}$ molecule cm$^{-3}$ are differed by a factor of $\sim 10$.

The observed difference in $\gamma_0(O_3)$ may be caused by 1) variability in heterogeneous reactivity

of different minerals and 2) that different experimental methods can lead to different results.

For example, it has been suggested that pretreatment of mineral dust particles (e.g., heating,

grounding, and evacuation) could modify their initial heterogeneous reactivity towards $O_3$

(Hanisch and Crowley, 2003a; Michel et al., 2003). Furthermore, as discussed in Section 2.2,

time resolution in different studies is also different, making interpretation of $\gamma_0$ difficult.





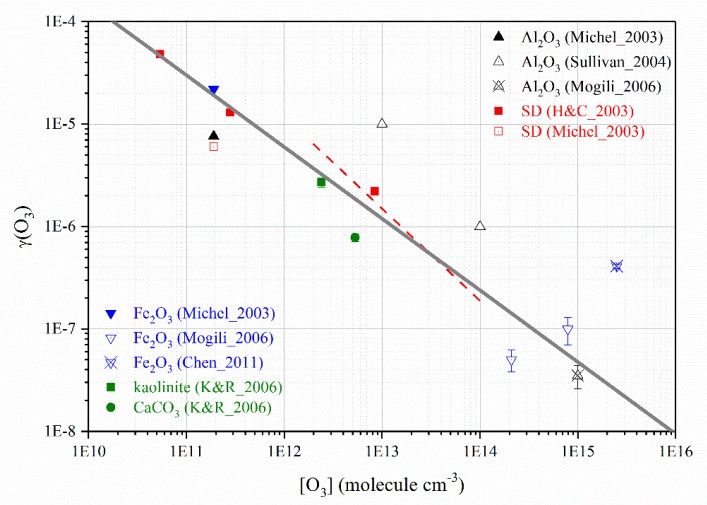

**Figure 13.** Dependence of $\gamma_{ss}(O_3)$ on initial $O_3$ concentrations under dry conditions for different

mineral dust particles: Michel_2003 (Michel et al., 2003), H&C_2003 (Hanisch and Crowley,

2003a), Sullivan_2004 (Sullivan et al., 2004), Mogili_2006 (Mogili et al., 2006a), K&R_2006

(Karagulian and Rossi, 2006), and Chen_2011 (Chen et al., 2011b). The red dashed curve

represents the dependence of $\gamma(O_3)$ on $[O_3]$ for Saharan dust reported by Chang et al. (2005),

and the grey solid curve represents the dependence of $\gamma(O_3)$ on $[O_3]$ for mineral dust particles

recommended by the IUPAC Task Group on Atmospheric Chemical Kinetic Data Evaluation.

Reprinted (with modification) with permission from the IUPAC Task Group on Atmospheric

Chemical Kinetic Data Evaluation ((http://iupac.pole-ether.fr).

In contrast, $\gamma_{ss}(O_3)$ reported by previous studies under dry conditions show fairly good

agreement (as displayed in Figure 13), considering the fact that very different experimental

techniques have been used (for example, aerosol samples were used by Mogili et al. (2006b)

and Chen et al. (2011b) while all the other studies used dust powder samples supported on

substrates). In addition, a rather strong dependence of $\gamma_{ss}(O_3)$ on initial $O_3$ concentration can be

observed. Eq. (19) has been recommended by the IUPAC task group on Atmospheric Chemical



Kinetic Data Evaluation to parameterize the dependence of $\gamma_{ss}(O_3)$ on $[O_3]$ (Crowley et al.,
2010a):
$$\gamma(O_3) = 1500 \times [O_3]^{-0.7} \quad (19)$$
where $[O_3]$ is $O_3$ concentration in molecule $cm^{-3}$. It is quite surprising that $\gamma_{ss}(O_3)$ under dry
conditions are very similar for all the minerals investigated. It can also been observed from
Figure 13 that $\gamma_{ss}(O_3)$ for $\alpha-Al_2O_3$ reported by Sullivan et al. (2004) and for $\alpha-Fe_2O_3$ reported
by Chen et al. (2011b) may be significantly larger than those recommended by Crowley et al.
(2010), and the reason is not very clear yet. It should be pointed out that the work by Sullivan
et al. (2005), though published, was not included in the original figure prepared by the IUPAC
Task Group. In addition, the work by Chen et al. (2011b) was published after the IUACP report
was released online.

Only three previous studies have explored effects of RH on heterogeneous reactions of

$O_3$ with mineral dust, and different results have been reported. While a strong negative effect
of RH on $O_3$ uptake kinetics was observed for $\alpha-Al_2O_3$ and $\alpha-Fe_2O_3$ by Mogili et al. (2006b),
the other two studies (Sullivan et al., 2004; Chang et al., 2005) suggested that the influence of
RH on heterogeneous uptake of $O_3$ by $\alpha-Al_2O_3$ and Saharan dust was insignificant. Further
experimental and theoretical work is required to better understand the effect of RH on $O_3$
uptake by mineral dust. As discussed below, surface adsorbed water may play different roles
in heterogeneous reaction of minerals with $O_3$.

A few other studies (Li et al., 1998; Li and Oyama, 1998; Roscoe and Abbatt, 2005;

Lampimaki et al., 2013) used different surface techniques to monitor mineral dust surfaces
during exposure to $O_3$. These studies did not report uptake coefficients and hence are not
included in Table 4. Nevertheless, they have provided valuable insights into reaction
mechanisms at the molecular level and are worthy of further discussion. A new Raman peak at
884 $cm^{-1}$ was observed after exposure $MnO_2$ to $O_3$, and it is attributed to peroxide species (i.e.



SS-$O_2$) by combining Raman spectroscopy, $^{18}O$ isotope substitution measurements, and ab
initio calculation (Li et al., 1998). Consequently, the following reaction mechanism has been
proposed for heterogeneous reaction of $O_3$ with metal oxides (Li et al., 1998):

$O_3(g) + SS \rightarrow SS\text{-}O + O_2(g)$     (R18a)

$SS\text{-}O + O_3(g) \rightarrow SS\text{-}O_2 + O_2(g)$     (R18b)

where SS represents reactive surface sites towards $O_3$. The intensity of the SS-$O_2$ peak was
found to decrease gradually with time after $O_3$ exposure was terminated, suggesting that SS-
$O_2$ would slowly decompose to $O_2$ (Li et al., 1998):

$SS\text{-}O_2 \rightarrow SS + O_2(g)$     (R18c)

A following study by the same group (Li and Oyama, 1998) suggested that the steady-state and
transient kinetics of heterogeneous decomposition of $O_3$ on $MnO_2$ could be well described by
the aforementioned reaction mechanism (R18a, R18b, and R18c). Reaction R18a is expected
to be of the Eley-Rideal type, because desorption of $O_3$ from mineral surfaces has never been
observed (Hanisch and Crowley, 2003a; Michel et al., 2003; Karagulian and Rossi, 2006) and
thus the Langmuir-Hinshelwood mechanism is unlikely. It is also suggested that reaction R18a
is much faster than the other two steps and the reactivation step (R18c) is slowest (Li et al.,
1998; Li and Oyama, 1998).

The reaction mechanism proposed by Li et al. was supported by several following

studies. For examples, gradual surface passivation was observed for a variety of minerals
(Hanisch and Crowley, 2003a; Michel et al., 2003), suggesting that the number of reactive
surface sites towards $O_3$ is limited, as implied by reactions R18a and R18b. On the other hand,
two previous studies (Hanisch and Crowley, 2003a; Sullivan et al., 2004) observed that surface
reactivation would slowly occur after $O_3$ exposure was stopped, and Michel et al. (2003) found
that heterogeneous uptake of $O_3$ by minerals is of catalytic nature to some extent. These studies
(Hanisch and Crowley, 2003a; Michel et al., 2003; Sullivan et al., 2004) clearly demonstrate





that a slow surface reactivation step exists, consistent with the reaction mechanism (more
precisely, reaction R18c) proposed by Li and coworkers (Li et al., 1998; Li and Oyama, 1998).

Using DRIFTS, Roscoe and Abbatt (2005) monitored the change of alumina during its

heterogeneous interaction with $O_3$ and water vapor. A new IR peak at 1380 cm$^{-1}$, attributed to
SS-O, appeared after alumina was exposed to $O_3$. Because alumina is opaque below 1100 cm$^{-1}$,
the SS-$O_2$ peak, expected to appear at around 884 cm$^{-1}$ (Li et al., 1998), could not be detected
by IR. When alumina was simultaneously exposed $O_3$ and water vapor, the intensity of the
SS-O peak was substantially decreased, compared to the case when exposure to $O_3$ alone. This
suggests that water molecules can be adsorbed strongly to sites which would otherwise react
with $O_3$, thus suppressing the formation of SS-O on the surface (Roscoe and Abbatt, 2005). In
this aspect, increasing RH will reduce heterogeneous reactivity of alumina towards $O_3$. It was
further found that if $O_3$-reacted alumina was exposed to water vapor, the intensity of the SS-O
IR peak would gradually decrease while the amount of surface adsorbed water would increase.
This indicates that SS-O would react with adsorbed water to regenerate reactive surface sites
(i.e. SS as shown in reaction R18a), implying that the presence of water vapor may also
promote $O_3$ uptake by alumina. As we discussed before, previous studies which examined the
effects of RH on heterogeneous reactions of $O_3$ with minerals (Sullivan et al., 2004; Chang et
al., 2005; Mogili et al., 2006a) do not agree with each other. This inconsistence may be (at least
partly) be caused by complex roles which adsorbed water plays in heterogeneous uptake of $O_3$
by mineral dust. Further work is required to elucidate the effect of RH, especially considering
that heterogeneous reaction of $O_3$ with minerals is of interest not only for atmospheric
chemistry but also for indoor air quality and industrial application (Dhandapani and Oyama,

1997).





### 3.3.2 Atmospheric implications

Using the dependence of $\gamma(O_3)$ on $[O_3]$ recommended by Crowley et al. (2010) and assuming an typical $O_3$ concentration of $1.5 \times 10^{12}$ molecule cm$^{-3}$ (~60 ppbv) in the troposphere, $\gamma(O_3)$ is calculated to be $4.5 \times 10^{-6}$. Consequently, lifetimes of $O_3$ with respect to heterogeneous reaction with mineral dust, $\tau_{het}(O_3)$, are estimated to be about 1280, 128, 13 days for dust mass concentrations of 10, 100, and 1000 μg m$^{-3}$, respectively. As discussed in Section 2.1.2, in polluted and forested areas where alkenes are abundant, $O_3$ lifetimes are around several hours; in these regions, $O_3$ removal due to direct heterogeneous uptake by mineral dust is unlikely to be significant. On the other hand, $O_3$ lifetimes in remote free troposphere are in the range of several days to a few weeks; therefore, direct removal of $O_3$ by heterogeneous reaction with mineral dust could play a minor but non-negligible role for some regions in the remote free troposphere heavily impact by mineral dust.

### 3.4 HCHO

The photocatalytic oxidation of HCHO on P25 TiO$_2$ surface was investigated as a function of HCHO concentration and RH (Obee and Brown, 1995). It has been shown that at a given HCHO concentration, oxidation rates of HCHO first increased and then decreased with RH. Noguchi et al. (1998) found that under dark conditions, P25 TiO$_2$ particles showed higher HCHO adsorption capacity (after normalized to surface area) than activated carbon. Under UV illumination, TiO$_2$ thin films could convert HCHO completely to CO$_2$ and H$_2$O, with formic acid (HCOOH) being an intermediate product; furthermore, the dependence of photo-degradation rates on $[HCHO]_0$ could be described by the Langmuir-Hinshelwood model (Noguchi et al., 1998). In another study (Liu et al., 2005), it has also been suggested that kinetics of photocatalytic oxidation of HCHO on TiO$_2$ surface could be described by the Langmuir-Hinshelwood model, and CO was identified as one of the products.



Ao et al. (2004) explored effects of NO, SO₂, and VOCs (including benzene, toluene,
ethylbenzene, and o-xylene) on the photo-degradation of HCHO on P25 TiO₂ particles. Formic
acid was identified as a major reaction intermediate, and HCHO degradation rates and HCOOH
yields both decreased with increasing RH (Ao et al., 2004). In addition, NO could accelerate
HCHO oxidation rates and HCOOH yields, whereas co-presence of SO₂ and VOCs used in this
study was found to inhibit photo-oxidation of HCHO (Ao et al., 2004). DRIFTS was used by
Sun et al. (2010) to investigate adsorption and photo-oxidation of HCHO on TiO₂. It has been
shown that adsorbed HCHO molecules can be rapidly converted to formate on the surface under
UV irradiation, and the presence of water vapor could significantly accelerate oxidation of
HCHO on TiO₂ (Sun et al., 2010).
All the aforementioned studies (Obee and Brown, 1995; Noguchi et al., 1998; Ao et al.,
2004; Liu et al., 2005; Sun et al., 2010) clearly showed that UV illumination could largely
enhance heterogeneous uptake of HCHO by TiO₂ particles, and HCOOH/HCOO⁻, CO₂, CO,
and H₂O were identified as reaction intermediates and/or products. Though these studies
provide useful insights into mechanisms of heterogeneous reaction of HCHO with TiO₂ surface,
they are not listed in Table 5 because no uptake coefficients have been reported. Heterogeneous
reaction of HCHO (10-40 ppbv) with soil samples was investigated using a coated wall flow
tube (Li et al., 2016). At 0% RH, the initial uptake coefficient was determined to be
$(1.1\pm0.05)\times10^{-4}$, gradually decreasing to $(5.5\pm0.4)\times10^{-5}$ within 8 h. Increasing RH would
suppress the uptake of HCHO, and around two thirds of HCHO molecules uptaken by the soil
was reversible (Li et al., 2016). The soil sample used by Li et al. were collected from a
cultivated field site (Mainz, Germany) and may not resemble the composition and mineralogy
of mineral dust aerosol; therefore, this study is not included in Table 7.



**Table 7:** Summary of previous laboratory studies on heterogeneous reactions of mineral dust with HCHO

| Dust | Reference | T (K) | Concentration (molecule cm⁻³) | Uptake coefficient | Techniques |
|---|---|---|---|---|---|
| $TiO_2$ | Xu et al., 2010 | 163-673 | $(1\text{-}20)\times10^{13}$ | At $295\pm2$ K, $\gamma_0$ (based on the BET surface area) were determined to be in the range of $0.5\times10^{-8}$ to $5\times10^{-8}$, increasing linearly with HCHO concentration ($1\times10^{13}$ to $2\times10^{14}$ molecule cm⁻³). UV irradiation and increasing temperature could both accelerate this reaction. | DRIFTS, IC |
|  | Sassine et al., 2010 | 278-303 | $(9\text{-}82)\times10^{10}$ | $\gamma_{ss}$ were determined to range from $(3.00\pm0.45)\times10^{-9}$ to $(2.26\pm0.34)\times10^{-6}$, depending on UV irradiation, HCHO concentration, RH, and temperature. | CWFT |
| $Al_2O_3$ | Carlos-Cuellar et al., 2003 | 295 | room temperature | $\gamma_0$: $(7.7\pm0.3)\times10^{-5}$ | KC-MS |
|  | Xu et al., 2006 | 273-333 | $(1\text{-}10)\times10^{13}$ | At 296 K, $\gamma_0$ was determined to be $(9.4\pm1.7)\times10^{-9}$ based on the BET surface area and $(2.3\pm0.5)\times10^{-5}$ based on the geometrical area for $\alpha$-Al$_2$O$_3$. UV irradiation and increasing temperature could both accelerate this reaction. | DRIFTS, IC |
|  | Xu et al., 2011 | 84-573 | $(1.3\text{-}3.6)\times10^{13}$ | At $295\pm2$ K, $\gamma_0$ was determined to be $(3.6\pm0.8)\times10^{-4}$ based on the geometrical area and $(1.4\pm0.31)\times10^{-8}$ based on the BET surface area for $\gamma$-Al$_2$O$_3$. UV irradiation and increasing temperature could both accelerate this reaction. | DRIFTS, IC |
| $SiO_2$ | Carlos-Cuellar et al., 2003 | 295 | room temperature | $\gamma_0$: $(2.6\pm0.9)\times10^{-7}$ | KC-MS |
|  | Sassine et al., 2010 | 278-303 | $(9\text{-}82)\times10^{10}$ | $\gamma_{ss}$ under dark conditions: $\sim3\times10^{-9}$ | CWFT |
| $Fe_2O_3$ | Carlos-Cuellar et al., 2003 | 295 | room temperature | $\gamma_0$: $(1.1\pm0.5)\times10^{-5}$ | KC-MS |




Carlos-Cuellar et al. (2003) first determined uptake coefficients of HCHO on several

mineral dust particles at room temperature, using a Knudsen cell reactor. Gradual surface
deactivation was observed for all three types of particles, and initial uptake coefficients ($\gamma_0$),
based on the BET surface area, were reported to be $(1.1\pm0.5)\times10^{-4}$ for $\alpha$-Fe$_2$O$_3$, $(7.7\pm0.3)\times10^{-5}$
for $\alpha$-Al$_2$O$_3$, and $(2.6\pm0.9)\times10^{-7}$ for SiO$_2$, respectively (Carlos-Cuellar et al., 2003).

Using DRIFTs and ion chromatography, Xu and co-workers systematically investigated

heterogeneous reactions of HCHO with $\alpha$-Al$_2$O$_3$ (Xu et al., 2006), $\gamma$-Al$_2$O$_3$ (Xu et al., 2011),
and TiO$_2$ particles (Xu et al., 2010) as a function of temperature, UV irradiation, and HCHO
concentration. It has been found that HCHO was first converted to dioxymethylene which was
then oxidized to formate on the surface, and UV irradiation and increasing temperature both
could enhance heterogeneous reactivity of all three types of particles towards HCHO (Xu et al.,
2006; Xu et al., 2010; Xu et al., 2011). $\gamma_0$(HCHO) on $\alpha$-Al$_2$O$_3$ at 293 K was determined to be
$(9.4\pm1.7)\times10^{-9}$ based on the BET surface area of the sample and $(2.3\pm0.5)\times10^{-5}$ based on the
geometrical area of the sample holder (Xu et al., 2006). At room temperature (295$\pm$2 K) and
under dark conditions, $\gamma_0$(HCHO), based on the BET surface area, were determined to be in the
range of $0.5\times10^{-8}$ to $5\times10^{-8}$ for TiO$_2$ (Xu et al., 2010), increasing linearly with HCHO
concentration ($1\times10^{13}$ to $2\times10^{14}$ molecule cm$^{-3}$). Under the same condition, $\gamma_0$(HCHO) was
determined to be $(3.6\pm0.8)\times10^{-4}$ based on the geometrical area and $(1.4\pm0.31)\times10^{-8}$ based on
the BET surface area for $\gamma$-Al$_2$O$_3$ (Xu et al., 2011). The effect of RH was further studied for $\gamma$-
Al$_2$O$_3$ at 295$\pm$2 K, and the dependence of BET surface area based $\gamma_0$(HCHO) on RH is given
by (Xu et al., 2011):

$$\ln[\gamma_0(BET)] = -17.5 - 0.0127 \times RH \quad (20)$$

where RH is in the unit of %.

A coated wall flow tube was deployed to investigate heterogeneous reactions of HCHO

with TiO$_2$ and SiO$_2$ particles, and the effects of UV irradiation, temperature (278-303 K), RH



(6-70 %), and HCHO concentration (3.5-32.5 ppbv) were systematically examined (Sassine et
al., 2010). Under dark conditions, the uptake of HCHO onto $SiO_2$ and $TiO_2$ was very slow,
with BET surface area based $\gamma_{ss}$ being $(3.00\pm0.45)\times10^{-9}$. Nevertheless, its uptake on $TiO_2$ and
$TiO_2/SiO_2$ mixture was largely enhanced by near-UV irradiation (340-420 nm) (Sassine et al.,
2010). For pure $TiO_2$ under the condition of 293 K, 30% RH and 2 ppbv HCHO, $\gamma_{ss}$ depended
linearly on irradiation intensity ($1.9\times10^{15}$ to $2.7\times10^{15}$ photons $cm^{-2}$ $s^{-1}$). The uptake kinetics can
be described by the Langmuir-Hinshelwood model: under the condition of 293 K, 6% RH, and
$2.7\times10^{15}$ photons $cm^{-2}$ $s^{-1}$, $\gamma_{ss}$ decreased from $(6.0\pm0.9)\times10^{-7}$ to $(2.0\pm0.3)\times10^{-7}$ for $TiO_2$ when
[HCHO] increased from 3.5 to 32.5 ppbv (Sassine et al., 2010).

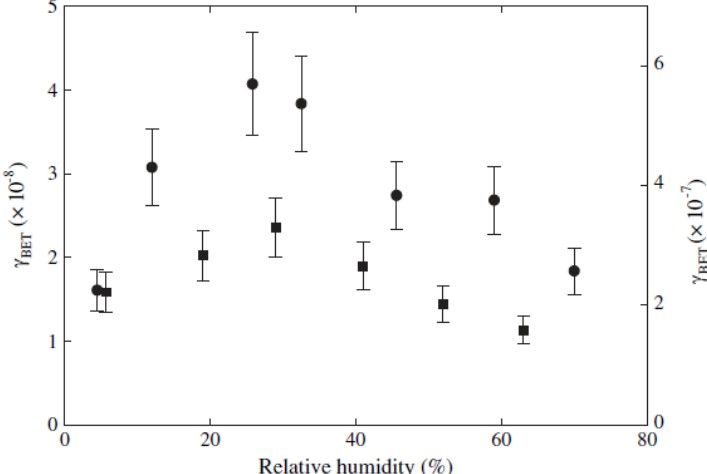


**Figure 14.** Effects of RH on heterogeneous uptake of HCHO by pure $TiO_2$ (circles, right y-
axis) and $TiO_2/SiO_2$ mixture (squares, left y-axis) which contains 5%wt $TiO_2$. Experimental
conditions: 293 K, 11 ppbv HCHO, $2.7\times10^{15}$ photons $cm^{-2}$ $s^{-1}$ illumination. Reprinted with
permission from Sassine et al. (2010). Copyright Elsevier 2010.

In addition, the effects of RH and temperature were also explored. As shown in Figure

14, $\gamma_{ss}$ was found to first increase with RH for $TiO_2$ (and $TiO_2/SiO_2$ mixture as well), reaching



a maximum at ~30%, and then decrease with RH. Under conditions of 30% RH, 11 ppbv
HCHO, and $2.7 \times 10^{15}$ photons $cm^{-2} s^{-1}$, $\gamma_{ss}$ increased from $(1.8 \pm 0.3) \times 10^{-7}$ at 298 K to
$(3.2 \pm 0.5) \times 10^{-7}$ at 303 K (Sassine et al., 2010).
**3.4.1 Discussion and atmospheric implication**
Two previous studies determined BET surface area based $\gamma_0$(HCHO) for $\alpha$-$Al_2O_3$
particles under dry conditions at room temperature, and $\gamma_0$(HCHO) reported by Carlos-Cuellar
et al. (2003) is >3 orders of magnitude larger than that reported by Xu et al. (2006). It is not
very clear yet why such a large difference was found between these two studies. Two studies
(Sassine et al., 2010; Xu et al., 2010) measured $\gamma$(HCHO) for $TiO_2$ particles; however, it is
difficult to make comparison because one study reported $\gamma_0$ (Xu et al., 2010) and the other one
reported $\gamma_{ss}$ (Sassine et al., 2010).
What we can conclude from previous studies as summarized in Table 7 is that our
understanding of atmospheric heterogeneous reaction of HCHO with mineral dust is very
limited. For example, all the previous studies only examined its reactions with oxides, while
clay minerals and authentic dust samples have never been investigated. Second, as discussed
above, large discrepancies are found for uptake coefficients reported by previous studies.
Furthermore, roles of RH in heterogeneous uptake of HCHO by mineral dust are not fully
understood. Last but not least, though several studies have observed that UV illumination could
largely enhance heterogeneous reaction of HCHO with mineral particles, it is non-trivial to
know that compared to dark conditions, to which extent this reaction is accelerated under
irradiation conditions relevant to the troposphere. Therefore, it is difficult to assess the
significance of heterogeneous uptake by mineral dust aerosol particles as a sink for HCHO in
a reliable manner.
An uptake coefficient of $(9.7 \pm 1.4) \times 10^{-6}$ was used by Sassine et al. (2010) to evaluate
the significance of heterogeneous reaction of HCHO with pure $TiO_2$ particles as a sink for





HCHO. This value was linearly extrapolated from their experimental measurements (2 ppbv
HCHO, 293 K, and 30% RH) to realistic solar conditions in the troposphere ($1.21 \times 10^{16}$ photons
$cm^{-2}$ $s^{-1}$). The value used by Sassine et al. (2010) is also adopted here to preliminarily assess
the impact of heterogeneous reaction of HCHO with mineral dust. For simplicity in our work
$\gamma$(HCHO) is set to $1 \times 10^{-5}$ which is only 3% larger than that used by Sassine et al. (2010).
Consequently, $\tau_{het}$(HCHO) are calculated to be about 456, 46, and 4.6 days for mineral dust
mass concentrations of 10, 100, and 1000 $\mu$g $m^{-3}$, respectively. For comparison, as we have
discussed in Section 2.1, typical lifetimes of HCHO are a few hours in the troposphere, with
photolysis and reaction with OH radicals being the two major removal processes. It is quite
clear that $\tau_{het}$(HCHO) are much larger than typical lifetimes of HCHO, and thus heterogeneous
reaction with mineral dust is unlikely to be significant for the removal of HCHO in the
troposphere.
**3.5 HONO**

Bedjanian and coworkers utilized a coated rod flow tube coupled to a mass spectrometer

to investigate heterogeneous reaction of HONO with $TiO_2$, $\gamma$-$Al_2O_3$, $Fe_2O_3$, and ATD particles
under dark and illuminated conditions (El Zein and Bedjanian, 2012; Romanias et al., 2012a;
El Zein et al., 2013a; El Zein et al., 2013b). All these measurements were carried out with dust
mass in the linear mass dependent regime, and thus BET surface area was used to calculate
uptake coefficients. We note that several previous studies have explored heterogeneous
interactions of HONO with Pyrex (Kaiser and Wu, 1977; Ten Brink and Spoelstra, 1998),
borosilicate glass (Syomin and Finlayson-Pitts, 2003), and $TiO_2$-doped commercial paints
(Laufs et al., 2010). However, these studies are not further discussed here because they are not
of direct atmospheric relevance. Uptake of HONO by soil samples was investigated using a
coated-wall flow tube (Donaldson et al., 2014), and uptake coefficients were found to decrease
with RH, from $(2.5\pm0.4) \times 10^{-4}$ at 0% RH to $(1.1\pm0.4) \times 10^{-5}$ at 80% RH. Soil used by Donaldson





et al. were collected from an agricultural field in Indiana and its mineralogical composition
may be quite different from mineral dust aerosol; as a result, this study is not included in Table

8.


**Table 8:** Summary of previous laboratory studies on heterogeneous reactions of mineral dust with HONO

| Dust | Reference | $T$ (K) | Concentration (molecule cm$^{-3}$) | Uptake coefficient | Techniques |
|---|---|---|---|---|---|
| TiO$_2$ | El Zein and Bedjanian, 2012 | 275-320 | $(0.3-3.3)\times10^{12}$ | $\gamma_0$ was determined to be ~$4.2\times10^{-6}$ at 10% RH and 300 K, showing negative dependence on RH (up to 12.6%) and $T$ (275-320 K). | CRFT-MS |
| | El Zein et al., 2013a | 275-320 | $(0.5-5)\times10^{12}$ | Under illuminated condition, $\gamma_0$ increased to ~$3.5\times10^{-4}$ at 10% RH and 280 K, showing negative dependence on RH (up to 60%) and $T$ (275-320 K). Though illumination enhanced HONO uptake compared to dark conditions, further increase in illumination intensity for $J$(NO$_2$) in the range of 0.002-0.012 s$^{-1}$ did not affect $\gamma_0$. | CRFT-MS |
| Al$_2$O$_2$ | Romanias et al., 2012b | 275-320 | $(0.6-3.5)\times10^{12}$ | At 10% RH, $\gamma_0$ was determined to be ~$1.2\times10^{-6}$ and ~$6.2\times10^{-6}$ under dark and illuminated conditions, respectively. $\gamma_0$ was found to increase linearly with $J$(NO$_2$) in the range of 0.002-0.012 s$^{-1}$. In addition, $\gamma_0$ decreased with RH, and no dependence on temperature was observed. | CRFT-MS |
| Fe$_2$O$_3$ | El Zein et al., 2013b | 275-320 | $(0.6-15.0)\times10^{12}$ | No significant effect of UV illumination, with $J$(NO$_2$) up to 0.012 s$^{-1}$, was observed. $\gamma_0$ was determined to be ~$4.1\times10^{-7}$ at 10% RH and 300 K, showing negative dependence on RH (up to 14.4 %) and no dependence on $T$ (275-320 K). | CRFT-MS |
| ATD | El Zein et al., 2013b | 275-320 | $(0.6-15.0)\times10^{12}$ | No significant effect of UV illumination, with $J$(NO$_2$) up to 0.012 s$^{-1}$, was observed. $\gamma_0$ was determined to be ~$9.3\times10^{-7}$ at 10% RH and 275 K, showing negative dependence on RH (up to 84.1%) and no dependence on $T$ (275-320 K). | CRFT-MS |






El Zein and Bedjanian (2012) measured heterogeneous uptake of HONO by $TiO_2$
particles under dark conditions. Upon exposure to HONO, heterogeneous reactivity of $TiO_2$
was progressively reduced, and the steady-state uptake coefficients were at least one order of
magnitude smaller than the corresponding initial uptake coefficients, $\gamma_0$ (El Zein and Bedjanian,
2012). $\gamma_0$, independent of initial HONO concentrations in the range of $(0.3\text{-}3.3)\times10^{12}$ molecule
$cm^{-3}$, showed strong dependence on RH and a slightly negative dependence on temperature.
The effects of temperature (275-320 K) at 0.001% RH and of RH at 300 K on $\gamma_0$ are given by
(El Zein and Bedjanian, 2012):
$$\gamma_0 = (1.4 \pm 0.5) \times 10^{-5} \times \exp[(1405 \pm 110)/T] \quad (21)$$
$$\gamma_0 = 1.8 \times 10^{-5} \times RH^{-0.63} \quad (22)$$
HONO uptaken by $TiO_2$ undergoes chemical conversion on the surface, and molecularly
adsorbed HONO is insignificant (El Zein and Bedjanian, 2012). This was confirmed by gas
phase production analysis, showing that the total yield of NO and $NO_2$ is equal to 1 within the
experimental uncertainties. The yields of NO and $NO_2$ were determined to be 0.42±0.07 and
0.60±0.09, respectively, independent of RH, temperature, and the initial HONO concentration
(El Zein and Bedjanian, 2012).
In a following study, El Zein et al. (2013a) examined the effect of illumination on the
uptake of HONO by $TiO_2$, and found that under illuminated conditions HONO uptake rates
also decreased with reaction time. Compared to dark conditions, HONO uptake was enhanced,
though no difference in the $\gamma_0$ was observed by varying UV illumination from 0.002 to 0.012 $s^{-1}$
(El Zein et al., 2013a). Under illuminated conditions, $\gamma_0$ is independent of initial HONO
concentration but depends inversely on temperature and RH. The effects of temperature (275-
320 K) at 0.002% RH and of RH (0.001-60%) at 280 K can be described by (El Zein et al.,
2013a):
$$\gamma_0 = (3.0 \pm 1.5) \times 10^{-5} \times \exp[(1390 \pm 150)/T] \quad (23)$$



$$\gamma_0 = 6.9 \times 10^{-4} \times RH^{-0.3} \quad (24)$$
Similar to dark conditions, all the HONO molecules removed from the gas phase have been
converted NO and NO$_2$. Yields of NO and NO$_2$ were determined to be 0.48±0.07 and 0.52±0.08,
respectively (El Zein et al., 2013a), independent of RH, temperature, and initial HONO
concentration.
The uptake of HONO by $\gamma$-Al$_2$O$_3$, Fe$_2$O$_3$, and ATD particles was also investigated under
dark and illuminated conditions as a function of temperature and RH. Progressive surface
deactivation was observed in all the experiments. For uptake onto $\gamma$-Al$_2$O$_3$, under both dark
and irradiated conditions $\gamma_0$(HONO) were found to be independent of initial HONO
concentration ($0.3 \times 10^{12}$ to $3.3 \times 10^{12}$ molecule cm$^{-3}$) and temperature (275-320 K), though RH
has a profound influence. Under dark conditions, $\gamma_0$ is given by (Romanias et al., 2012a):
$$\gamma_0 = 4.8 \times 10^{-6} \times RH^{-0.61} \quad (25)$$
for RH in the range of 0.00014% to 10.5%. UV illumination linearly enhances initial HONO
uptake, with $\gamma_0$ under illumination with $J$(NO$_2$) equal to 0.012 s$^{-1}$ given by (Romanias et al.,
2012a):
$$\gamma_0 = 1.7 \times 10^{-5} \times RH^{-0.44} \quad (26)$$
for RH in the range of 0.0003% to 35.4%. NO and NO$_2$ yields were determined to be 0.40±0.06
and 0.60±0.09 for all the experimental conditions.
No significant effects of UV irradiation with $J$(NO$_2$) up to 0.012 s$^{-1}$ were observed for
heterogeneous reaction of HONO with Fe$_2$O$_3$ and ATD particles (El Zein et al., 2013b).
$\gamma_0$(HONO) were found to be independent of initial HONO concentration ($0.6 \times 10^{12}$ to $15.0 \times 10^{12}$
molecule cm$^{-3}$) and temperature (275-320 K), while RH has a significant impact, given by (El
Zein et al., 2013b):
$$\gamma_0 = 1.7 \times 10^{-6} \times RH^{-0.62} \quad (27)$$
for Fe$_2$O$_3$ and RH in the range of 0.0003% to 14.4%, and





$$\gamma_0 = 3.8 \times 10^{-6} \times RH^{-0.61} \quad (28)$$
for ATD and RH in the range of 0.00039% to 84.1%. NO and $NO_2$ yields, independent of
experimental conditions, were reported to be 0.40±0.06 and 0.60±0.09, respectively (El Zein
et al., 2013b).

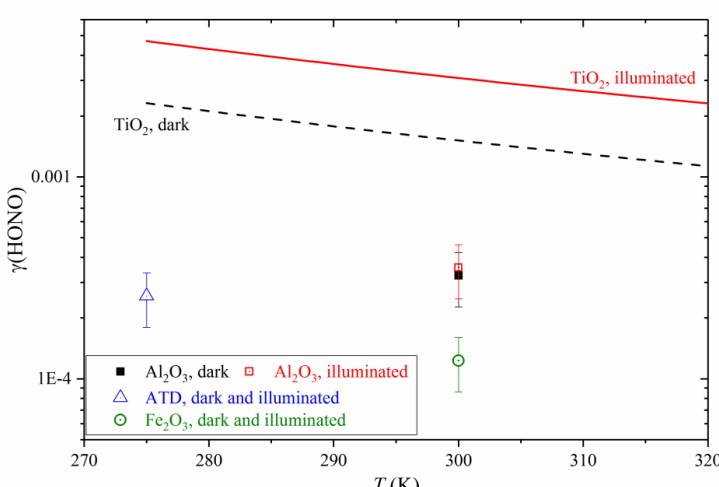


**Figure 15.** Temperature dependence of $\gamma_0$(HONO) for $TiO_2$ (El Zein and Bedjanian, 2012; El
Zein et al., 2013a), $Al_2O_3$ (Romanias et al., 2012a), ATD (El Zein et al., 2013b) and $Fe_2O_3$ (El
Zein et al., 2013b) under dark and illuminated conditions. Data at 0.001% RH were presented
except for illuminated $TiO_2$ at 0.002% RH. Please note that no significant temperature (275-
320 K) effect was found for $Al_2O_3$, ATD, and $Fe_2O_3$. In addition, no difference in uptake
kinetics was observed between dark and illuminated conditions for ATD and $Fe_2O_3$.

The dependence of $\gamma_0$(HONO) on temperature is displayed in Figure 15 for different
mineral dust under dark and illuminated conditions. No significant effect of temperature was
observed for uptake onto $Al_2O_3$, $Fe_2O_3$, and ATD. When temperature increases from 275 K to
320 K, $\gamma_0$(HONO) is reduced by a factor of about 2 under both dark and illuminated conditions
for $TiO_2$. It is interesting to note that UV illumination has different impacts on HONO uptake



for different minerals. HONO uptake onto $Al_2O_3$ is enhanced by UV radiation, and the extent
of enhancement shows linear dependence on illumination intensity for $J(NO_2)$ in the range of
0.002-0.012 $s^{-1}$ (Romanias et al., 2012a). In contrast, photo-enhancement was found to be
insignificant for ATD and $Fe_2O_3$ with $J(NO_2)$ up to 0.012 $s^{-1}$ (El Zein et al., 2013b). Significant
enhancement in $\gamma_0(HONO)$ was observed for illuminated $TiO_2$ with $J(NO_2)$ of 0.002 $s^{-1}$ when
compared to dark conditions, especially at evaluated RH as shown in Figure 16; however,
further increase in illumination intensity with $J(NO_2)$ up to 0.012 $s^{-1}$ did not lead to further
increase in $\gamma_0(HONO)$ (El Zein et al., 2013a). In addition, we note that NO and $NO_2$ yields were
found to be ~0.40 and 0.60 for all the four types of minerals investigated, independent of
experimental conditions.

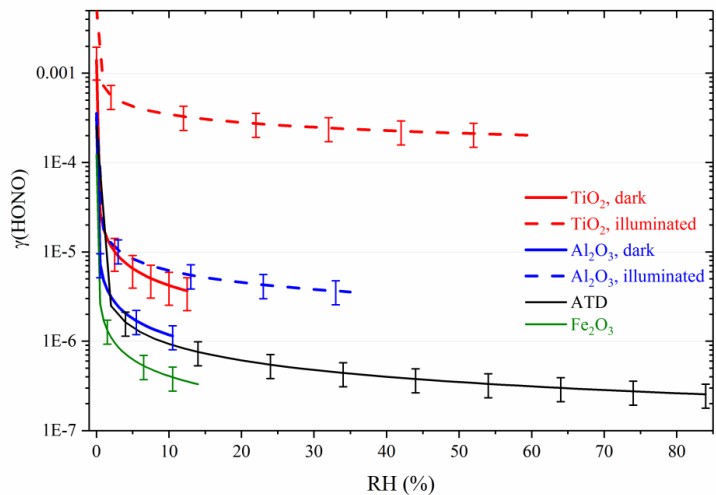


**Figure 16.** RH dependence of $\gamma_0(HONO)$ for $TiO_2$ (El Zein and Bedjanian, 2012; El Zein et al.,
2013a), $Al_2O_3$ (Romanias et al., 2012a), ATD (El Zein et al., 2013b) and $Fe_2O_3$ (El Zein et al.,
2013b) under dark and illuminated conditions at around room temperature.

Figure 16 shows effects of RH on $\gamma_0(HONO)$ at around room temperature for $TiO_2$,

$Al_2O_3$, ATD, and $Fe_2O_3$. Most of measurements were only carried out at low RH (<15%), and



thus their atmospheric relevance is rather limited. Experiments using ATD and illuminated
$TiO_2$ particles were conducted at RH over a wide range, and a negative dependence of
$\gamma_0$(HONO) on RH was observed. When RH increases from 10% to 60%, $\gamma_0$(HONO) is reduced
by ~66% and ~42% for ATD and illuminated $TiO_2$, respectively.
**3.5.1 Discussion and atmospheric implication**
All the fours studies, as shown in Figures 15 and 16, were carried out by the same group.
Furthermore, heterogeneous interactions of HONO with authentic dust and clay minerals which
are the major components for tropospheric dust, have not been explored yet. Future studies can
provide more scientific insights to reaction mechanisms and better quantify uptake kinetics.
In this work we use $\gamma_0$(HONO) for ATD, the only authentic dust sample investigated,
to preliminarily assess the significance of heterogeneous uptake by mineral dust as a HONO
sink. As shown in Figure 16, $\gamma_0$(HONO) decreases from $9.3 \times 10^{-7}$ at 10% to $2.6 \times 10^{-7}$ at 80%. A
$\gamma$(HONO) value of $1 \times 10^{-6}$ is adopted here to calculate $\tau_{het}$(HONO) with respect to
heterogeneous reaction with mineral dust. This may represent an upper limit for its atmospheric
significance, because i) at typical RH found in the troposphere, $\gamma_0$(HONO) should be $<1 \times 10^{-6}$
according to the work by El Zein et al. (2013b); ii) surface deactivation was observed, and thus
the average $\gamma$(HONO) should be smaller than $\gamma_0$(HONO) (El Zein et al., 2013b). Using Eq. (6),
$\tau_{het}$(HONO) is calculated to be ~57 days for dust mass concentration of 1000 μg m$^{-3}$ which can
only occur during dust storms. For comparison, typical HONO lifetimes in the troposphere are
estimated to be 10-20 min, with the major sink being photolysis (in Section 2.1). Therefore,
heterogeneous uptake by mineral dust is a negligible sink for HONO in the troposphere.
**3.6 $N_2O_5$ and $NO_3$ radicals**
$N_2O_5$ and $NO_3$ in the troposphere are in the dynamic equilibrium, as introduced in
Section 2.1.3. Therefore, their heterogeneous reactions with mineral dust are discussed together
in this section.





**3.6.1 $N_2O_5$**

Heterogeneous reaction of $N_2O_5$ with mineral dust particles was investigated for the

first time by Seisel et al. (2005), using DRIFTS and a Knudsen cell reactor coupled to quadruple
mass spectrometry. The initial uptake coefficient of $N_2O_5$ on Saharan dust was determined to
be $0.080\pm0.003$ at 298 K, and slowly decreased to a steady-state value of $0.013\pm0.003$ (Seisel
et al., 2005). Formation of nitrate on dust particles was observed, and $N_2O_5$ uptake was
suggested to proceed with two mechanisms, i.e. heterogeneous hydrolysis and its reaction with
surface OH groups (Seisel et al., 2005). A Knudsen cell reactor was also used by Karagulian
et al. (2006) to investigate heterogeneous uptake of $N_2O_5$ by several different types of mineral
dust. Both the initial and steady-state uptake coefficient were found to decrease with increasing
initial $N_2O_5$ concentrations. When $N_2O_5$ concentration was $(4.0\pm1.0)\times10^{11}$ molecule cm$^{-3}$, $\gamma_0$
and $\gamma_{ss}$ were determined to be $0.30\pm0.08$ and $0.20\pm0.05$ for Saharan dust, $0.12\pm0.04$ and
$0.021\pm0.006$ for $CaCO_3$, $0.20\pm0.06$ and $0.11\pm0.03$ for ATD, $0.16\pm0.04$ and $0.021\pm0.006$ for
kaolinite, and $0.43\pm0.13$ and $0.043\pm0.013$ for natural limestone, respectively. When $N_2O_5$
concentration increased to $(3.8\pm0.5)\times10^{12}$ molecule cm$^{-3}$, $\gamma_0$ and $\gamma_{ss}$ were determined to be
$0.090\pm0.026$ and $0.059\pm0.016$ for Saharan dust, $0.033\pm0.010$ and $0.0062\pm0.0018$ for $CaCO_3$,
$0.064\pm0.019$ and $0.016\pm0.004$ for ATD, $0.14\pm0.04$ and $0.022\pm0.006$ for kaolinite, and
$0.011\pm0.003$ and $0.0022\pm0.0006$ for natural limestone, respectively (Karagulian et al., 2006).
Formation of $HNO_3$ in the gas phase was detected, with production yield varying with dust
mineralogy. The postulated reason is that partitioning of formed $HNO_3$ between gas and
particle phases may vary for different dust samples (Karagulian et al., 2006).

Wagner et al. (2008) utilized a Knudsen cell reactor to study heterogeneous uptake of

$N_2O_5$ by Saharan dust, ATD, and $CaCO_3$ particles at $296\pm2$ K. Interestingly, surface
deactivation was only observed for $CaCO_3$ under their experimental conditions. Therefore, $\gamma_0$
and $\gamma_{ss}$ are equal for the other two types of dust, being $0.037\pm0.012$ for Saharan dust and





0.022±0.008 for ATD, respectively (Wagner et al., 2008). The initial uptake coefficient was
reported to be 0.05±0.02 for $CaCO_3$; pre-heating could reduce its heterogeneous reactivity
towards $N_2O_5$ (Wagner et al., 2008), very likely due to the loss of surface adsorbed water and
surface OH groups. It should be noted that all the uptake coefficients measured by using
Knudsen cell reactors are based on the projected area of dust samples (Seisel et al., 2005;
Karagulian et al., 2006; Wagner et al., 2008).
Heterogeneous reactions of $N_2O_5$ with airborne mineral dust particles were also
investigated by several previous studies, with the first one being carried out by Mogili et al.
(2006b). In this study, in-situ FTIR measurements was carried out to determine $N_2O_5$ loss due
to reactions with dust particles in an environmental chamber at 290 K. The uptake coefficients
of $N_2O_5$, based on the BET area of dust particles, increase with RH for $SiO_2$, from
$(4.4\pm0.4)\times10^{-5}$ at <1% RH, to $(9.3\pm0.1)\times10^{-5}$ at 11% RH, $(1.2\pm0.2)\times10^{-4}$ at 19% RH, and
$(1.8\pm0.4)\times10^{-4}$ at 43% RH (Mogili et al., 2006b). In addition, $\gamma(N_2O_5)$ at <1% RH were
determined to be for $(1.9\pm0.2)\times10^{-4}$ for $CaCO_3$, $(9.8\pm0.1)\times10^{-4}$ for kaolinite, $(4.0\pm0.4)\times10^{-4}$ for
$\alpha$-$Fe_2O_3$, and $(1.9\pm0.2)\times10^{-4}$ for montmorillonite, respectively (Mogili et al., 2006b).




**Table 9:** Summary of previous laboratory studies on heterogeneous reactions of mineral dust with $N_2O_5$

| Dust | Reference | $T$ (K) | Concentration (molecule $cm^{-3}$) | Uptake coefficient | Techniques |
|---|---|---|---|---|---|
| Saharan dust | Seisel et al., 2005 | 298 | $(0.03-5)\times10^{12}$ | $\gamma_0$: $0.080\pm0.003$ and $\gamma_{ss}$: $0.013\pm0.003$ | KC, DRIFTS |
| | Karagulian et al., 2006 | $298\pm2$ | $(0.4-3.8)\times10^{12}$ | When $[N_2O_5]$ was $(4.0\pm1.0)\times10^{11}$ molecule $cm^{-3}$, $\gamma_0 = 0.30\pm0.08$ and $\gamma_{ss} = 0.20\pm0.05$; when $[N_2O_5]$ was $(3.8\pm0.5)\times10^{12}$ molecule $cm^{-3}$, $\gamma_0 = 0.090\pm0.026$ and $\gamma_{ss} = 0.059\pm0.016$. | KC |
| | Wagner et al., 2008 | $296\pm2$ | KC: $(3.0-11.0)\times10^{9}$; AFT: $(5-20)\times10^{12}$ | KC measurements: $\gamma_0 = \gamma_{ss} = 0.037\pm0.012$; AFT measurements: $0.026\pm0.004$ at 0% RH, $0.016\pm0.004$ at 29% RH, and $0.010\pm0.004$ at 58% RH. | KC-MS, AFT-CLD |
| | Tang et al., 2012 | $297\pm1$ | $(0.5-30)\times10^{12}$ | $0.02\pm0.01$, independent of RH (0-67%) | AFT-CRDS |
| ATD | Karagulian et al., 2006 | $298\pm2$ | $(0.4-3.8)\times10^{12}$ | When $[N_2O_5]$ was $(4.0\pm1.0)\times10^{11}$ molecule $cm^{-3}$, $\gamma_0 = 0.20\pm0.06$ and $\gamma_{ss} = 0.11\pm0.03$; when $[N_2O_5]$ was $(3.8\pm0.5)\times10^{12}$ molecule $cm^{-3}$, $\gamma_0 = 0.064\pm0.019$ and $\gamma_{ss} = 0.016\pm0.004$. | KC |
| | Wagner et al., 2008 | $296\pm2$ | $(3.3-10.4)\times10^{9}$ | $\gamma_0 = \gamma_{ss} = 0.022\pm0.008$ | KC-MS |
| | Wagner et al., 2009 | $296\pm2$ | $(10-44)\times10^{12}$ | $0.0098\pm0.0010$ at 0% RH and $0.0073\pm0.0007$ at 29% RH | AFT-CLD |
| | Tang et al., 2014c | $297\pm1$ | $(11-22)\times10^{12}$ | $(7.7\pm1.0)\times10^{-3}$ at 0% RH, $(6.0\pm2.0)\times10^{-3}$ at 17% RH, $(7.4\pm0.7)\times10^{-3}$ at 33% RH, $(4.9\pm1.3)\times10^{-3}$ at 50% RH, and $(5.0\pm0.3)\times10^{-3}$ at 67% RH. | AFT-CRDS |
| $CaCO_3$ | Karagulian et al., 2006 | $298\pm2$ | $(0.4-3.8)\times10^{12}$ | When $[N_2O_5]$ was $(4.0\pm1.0)\times10^{11}$ molecule $cm^{-3}$, $\gamma_0 = 0.12\pm0.04$ and $\gamma_{ss} = 0.021\pm0.006$; when $[N_2O_5]$ was $(3.8\pm0.5)\times10^{12}$ molecule $cm^{-3}$, $\gamma_0 = 0.033\pm0.010$ and $\gamma_{ss} = 0.0062\pm0.0018$. | KC |
| | Mogili et al., 2006b | 290 | $(2-3)\times10^{15}$ | $(1.9\pm0.2)\times10^{-4}$ at <1% RH | EC |



Atmospheric Chemistry and Physics Discussions — Open Access

| | Reference | $T$ (K) | concentration range | description | technique |
|---|---|---|---|---|---|
| | Wagner et al., 2008 | 296±2 | $(1.7\text{-}4.5)\times10^{9}$ | $\gamma_0 = 0.05\pm0.02$ | KC-MS |
| | Wagner et al., 2009 | 296±2 | $(1\text{-}40)\times10^{12}$ | $0.0048\pm0.0007$ at 0% RH, $0.0053\pm0.0010$ at 29% RH, $0.0113\pm0.0016$ at 58% RH, and $0.0194\pm0.0022$ at 71% RH. | AFT-CLD |
| SiO$_2$ | Mogili et al., 2006b | 290 | $(2\text{-}3)\times10^{15}$ | $(4.4\pm0.4)\times10^{-5}$ at <1% RH, $(9.3\pm0.1)\times10^{-5}$ at 11% RH, $(1.2\pm0.2)\times10^{-4}$ at 19% RH, and $(1.8\pm0.4)\times10^{-4}$ at 43% RH. | EC |
| | Wagner et al., 2009 | 296±2 | $(0.5\text{-}30)\times10^{12}$ | $0.0086\pm0.0006$ at 0% RH and $0.0045\pm0.0005$ at 29% | AFT-CLD |
| | Tang et al., 2014a | 296±2 | $(10\text{-}50)\times10^{12}$ | $(7.2\pm0.6)\times10^{-3}$ at $(7\pm2)$% RH, $(5.6\pm0.6)\times10^{-3}$ at $(26\pm2)$% RH, and $(5.3\pm0.8)\times10^{-3}$ at $(40\pm3)$% RH. | AFT-CLD |
| kaolinite | Karagulian et al., 2006 | 298±2 | $(0.4\text{-}3.8)\times10^{12}$ | When [N$_2$O$_5$] was $(4.0\pm1.0)\times10^{11}$ molecule cm$^{-3}$, $\gamma_0 = 0.16\pm0.04$ and $\gamma_{ss} = 0.021\pm0.006$; when [N$_2$O$_5$] was $(3.8\pm0.5)\times10^{12}$ molecule cm$^{-3}$, $\gamma_0 = 0.14\pm0.04$ and $\gamma_{ss} = 0.022\pm0.006$. | KC |
| | Mogili et al., 2006b | 290 | $(2\text{-}3)\times10^{15}$ | $(9.8\pm0.1)\times10^{-4}$ at <1% RH | EC |
| natural limestone | Karagulian et al., 2006 | 298±2 | $(0.4\text{-}3.8)\times10^{12}$ | When [N$_2$O$_5$] was $(4.0\pm1.0)\times10^{11}$ molecule cm$^{-3}$, $\gamma_0 = 0.43\pm0.13$ and $\gamma_{ss} = 0.043\pm0.013$; when [N$_2$O$_5$] was $(3.8\pm0.5)\times10^{12}$ molecule cm$^{-3}$, $\gamma_0 = 0.011\pm0.003$ and $\gamma_{ss} = 0.0022\pm0.0006$. | KC |
| montmorillonite | Mogili et al., 2006b | 290 | $(2\text{-}3)\times10^{15}$ | $(1.8\pm0.2)\times10^{-4}$ at <1% RH | EC |
| illite | Tang et al., 2014c | 297±1 | $(8\text{-}24)\times10^{12}$ | $0.091\pm0.039$ at 0% RH and $0.093\pm0.008$ at 17% RH, $0.072\pm0.021$ at 33% RH, $0.049\pm0.006$ at 50% RH, and $0.039\pm0.012$ at 67% RH. | AFT-CRDS |
| TiO$_2$ | Tang et al., 2014d | 296±2 | $(10\text{-}50)\times10^{12}$ | $(1.83\pm0.32)\times10^{-3}$ at $(5\pm1)$% RH, $(2.01\pm0.27)\times10^{-3}$ at $(12\pm2)$% RH, $(1.02\pm0.20)\times10^{-3}$ at $(23\pm2)$% RH, $(1.29\pm0.26)\times10^{-3}$ at $(33\pm2)$% RH, $(2.28\pm0.51)\times10^{-3}$ at $(45\pm3)$% RH, and $(4.47\pm2.04)\times10^{-3}$ at $(30\pm3)$% RH. | AFT-CLD |
| Fe$_2$O$_3$ | Mogili et al., 2006b | 290 | $(2\text{-}3)\times10^{15}$ | $(4.0\pm0.4)\times10^{-4}$ at <1% RH | EC |



An atmospheric pressure aerosol flow tube was deployed by Wagner et al. (2008, 2009)
to investigate heterogeneous reactions of $N_2O_5$ with Saharan dust, ATD, calcite, and $SiO_2$
aerosol particles at 296±2 K, and $N_2O_5$ decays in the flow tube were detected by using a
modified chemiluminescence method. Slightly negative dependence of $\gamma(N_2O_5)$ on RH was
observed for Saharan dust, ATD, and $SiO_2$ aerosol particles. $\gamma(N_2O_5)$ was determined to be
0.026±0.004 at 0% RH, 0.016±0.004 at 29% RH, and 0.010±0.004 at 58% RH for Saharan dust
(Wagner et al., 2008), 0.0086±0.0006 at 0% RH and 0.0045±0.0005 at 29% for $SiO_2$ (Wagner
et al., 2009), and 0.0098±0.0010 at 0% RH and 0.0073±0.0007 at 29% RH for ATD (Wagner
et al., 2009), respectively. In contrast, $\gamma(N_2O_5)$ increases with RH for $CaCO_3$, from
0.0048±0.0007 at 0% RH to 0.0194±0.0022 at 71% RH (Wagner et al., 2009). It should be
pointed out that in the original paper (Wagner et al., 2008) the uptake coefficients for Saharan
dust were based on the aerosol surface area concentrations after the shape factor correction was
applied. In order to keep consistence with other studies, $\gamma(N_2O_5)$ reported by Wagner et al.
(2008) have been recalculated in this review without taking into account the shape factor of
Saharan dust.
Tang and co-workers systematically investigated the dependence of $\gamma(N_2O_5)$ on RH and
dust mineralogy, using aerosol flow tubes with $N_2O_5$ measured by a modified
chemiluminescence method (Tang et al., 2012; Tang et al., 2014c) or cavity ring-down
spectroscopy (Tang et al., 2014a; Tang et al., 2014e). Within experimental uncertainties,
$\gamma(N_2O_5)$ was determined to be 0.02±0.01 for Saharan dust (Tang et al., 2012), independent of
RH (0-67%) and initial $N_2O_5$ concentration ($5\times10^{11}$ to $3\times10^{13}$ molecule cm$^{-3}$). Products
analysis suggests that $N_2O_5$ is converted to particulate nitrate after heterogeneous reaction with
Saharan dust, and that the formation of $NO_2$ in the gas phase is negligible (Tang et al., 2012).
It has also been shown that if pretreated with high levels of gaseous $HNO_3$, heterogeneous
reactivity of Saharan dust towards $N_2O_5$ would be substantially reduced (Tang et al., 2012). A



strong negative effect of RH on $\gamma(N_2O_5)$ was found for uptake onto illite, with $\gamma(N_2O_5)$
decreasing from 0.091±0.039 at 0% RH to 0.039±0.012 at 67% RH. The negative effect of RH
is much smaller for ATD, with $\gamma(N_2O_5)$ determined to be 0.0077±0.0010 at 0% RH and
0.0050±0.0003 at 67% RH (Tang et al., 2014c). $\gamma(N_2O_5)$ on $SiO_2$ particles decreases from
0.0072±0.0006 at (7±2)% RH to 0.0053±0.0008 at (40±2)% RH (Tang et al., 2014a), also
showing a weak negative RH dependence. RH exhibits complex effects on heterogeneous
reaction of $N_2O_5$ with $TiO_2$ particles, and the reported $\gamma(N_2O_5)$ first decreases with RH from
$(1.83\pm0.32)\times10^{-3}$ at (5±1)% RH to $(1.02\pm0.20)\times10^{-3}$ at (23±2)% RH, and then increases with
RH to $(4.47\pm2.04)\times10^{-3}$ at (60±3)% RH (Tang et al., 2014d). Analysis of optically levitated
single micrometer sized $SiO_2$ particles using Raman spectroscopy during their reaction with
$N_2O_5$ (Tang et al., 2014a) suggests that $HNO_3$ formed in this reaction can partition between gas
and particle phases, with partitioning largely governed by RH.

Figure 17 summarizes $\gamma(N_2O_5)$ onto Saharan dust reported by previous work. $\gamma(N_2O_5)$

reported by the three studies using Knudsen cell reactors (Seisel et al., 2005; Karagulian et al.,
2006; Wagner et al., 2008) show large variation, with $\gamma_{ss}(N_2O_5)$ ranging from 0.013±0.003 to
0.20±0.05. This comparison demonstrates that sample preparation methods could largely
influence reported uptake coefficients using particles supported on a substrate, even though
they all used Knudsen cell reactor (as discussed in Section 2.2.1). In addition, significant
surface saturation was observed by Seisel et al. (2005) and Karagulian et al. (2006), but not by
Wagner et al. (2008). For the same reason, $\gamma(N_2O_5)$ reported by two Knudsen studies
(Karagulian et al., 2006; Wagner et al., 2008) exhibit significant discrepancy for Arizona Test
Dust (and reasonably good agreement is found for $CaCO_3$). Instead, the two aerosol flow tube
studies (Wagner et al., 2008; Tang et al., 2012) show good agreement in $\gamma(N_2O_5)$ onto Saharan
dust considering experimental uncertainties, though RH was found to have a slightly negative
effect by Wagner et al. (2008) while no significant effect of RH was observed by Tang et al.




(2012). Since cavity ring-down spectroscopy used by Tang et al. (2012) to detect $N_2O_5$ is more
sensitive and selective than the chemiluminescence method used by Wagner et al. (2008), in
this work we choose to use the uptake coefficient (0.02±0.01) reported by Tang et al. (2012),
as recommended by the IUPAC task group, to assess $\tau_{het}(N_2O_5)$ in the troposphere.

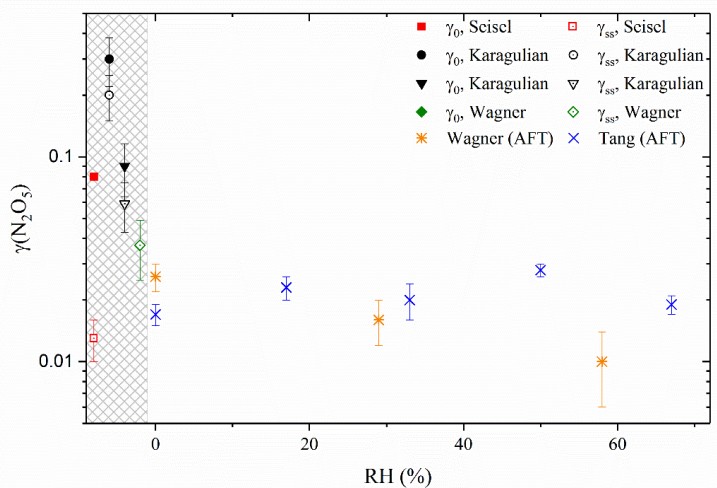


**Figure 17.** Uptake coefficients of $N_2O_5$ for Saharan dust, as reported by previous studies.
Knudsen cell studies were all carried out under vacuum conditions (i.e. 0% RH), and for better
readability these results are plotted in the region of RH <0% (shadowed region). Karagulian et
al. (2006) reported $\gamma_0$ and $\gamma_{ss}$ at two different $N_2O_5$ concentrations (circles: ~$4\times10^{11}$ molecule
$cm^{-3}$; triangles: ~$4\times10^{12}$ molecule $cm^{-3}$); $\gamma_0$ and $\gamma_{ss}$ reported by Wagner et al. (2008) using a
Knudsen cell reactor are equal and thus overlapped with each other in Figure 17.

It is somehow unexpected that $\gamma(N_2O_5)$ onto $SiO_2$ reported by the first two studies

(Mogili et al., 2006b; Wagner et al., 2009), both using aerosol samples, differ by about two
orders of magnitude. A third study (Tang et al., 2014a), using an aerosol flow tube, concluded
that this discrepancy is largely due to the fact that $SiO_2$ particles are likely to be porous. Mogili
et al. (2006b) and Wagner et al. (2009) used BET surface area and the Stokes diameter to



calculate the aerosol surface area, respectively. If BET surface area is used, $\gamma(N_2O_5)$ reported
by Tang et al. (2014a) show good agreement with those determined by Mogili et al. (2006b);
if mobility diameters are used to derive aerosol surface area, they agree well with those reported
by Wagner et al. (2009). Nevertheless, some discrepancies still remain: Wagner et al. (2009)
and Tang et al. (2014a) suggested a small negative dependence of $\gamma(N_2O_5)$ on RH, and Mogili
et al. (2006b) found that $\gamma(N_2O_5)$ significantly increase with RH. In addition, $\gamma(N_2O_5)$ onto
$CaCO_3$ aerosol particles at <1% RH, as reported by Mogili et al. (2006b) and Wagner et al.
(2009), differ by a factor of >20. It is not yet clear if the difference in calculating surface area
(BET surface area versus Stokes diameter based surface area) could explain such a large
difference, and further work is required to resolve this issue.

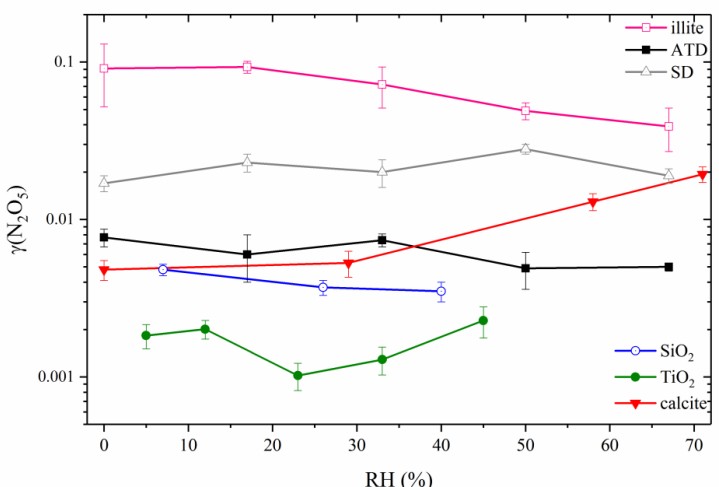


**Figure 18.** Uptake coefficients of $N_2O_5$ for Saharan dust (Tang et al., 2012), ATD (Tang et al.,
2014c), illite (Tang et al., 2014c), $CaCO_3$ (Wagner et al., 2009), $SiO_2$ (Tang et al., 2014a), and
$TiO_2$ (Tang et al., 2014e), as reported by aerosol flow tube studies.

Aerosol flow tubes have been deployed to investigate heterogeneous interactions of

$N_2O_5$ with different types of mineral dust, with reported $\gamma(N_2O_5)$ summarized in Figure 18.





Two distinctive features can be identified. First, different minerals exhibit very different
heterogeneous reactivity towards $N_2O_5$. $\gamma(N_2O_5)$ at <10% RH increase from $(1.83\pm0.32) \times 10^{-3}$
for $TiO_2$ to $0.091\pm0.039$ for illite, spanning over almost two orders of magnitude. Second, RH
(and thus surface adsorbed water) plays important and various roles in uptake kinetics. For
example, increasing RH significantly suppresses $N_2O_5$ uptake onto illite but largely enhances
its uptake onto $CaCO_3$, while it does not show a significant effect for Saharan dust. In this paper
$\gamma(N_2O_5)$ onto Saharan dust is used to assess the significance of heterogeneous reaction of $N_2O_5$
with mineral dust. Mineralogy of Asian dust is different from Saharan dust, and thus their
heterogeneous reactivity (and probably the effect of RH) towards $N_2O_5$ can be different.
Considering that Asian dust is transported over East Asia with high levels of NOx and $O_3$
(Zhang et al., 2007; Geng et al., 2008; Shao et al., 2009; Ding et al., 2013; Itahashi et al., 2014)
and thus also $N_2O_5$ (Brown et al., 2016; Tham et al., 2016; Wang et al., 2016), heterogeneous
reaction of $N_2O_5$ with Asian dust deserves further investigation.
Using $\gamma(N_2O_5)$ of 0.02, $\tau_{het}(N_2O_5)$ are estimated to be ~10 h, ~1 h, and ~6 min for dust
loading of 10, 100, and 1000 $\mu m\ m^{-3}$, respectively. $N_2O_5$ lifetimes in the troposphere is typically
in the range of several minutes to several hours, as shown in Table 1. Therefore, heterogeneous
uptake by mineral dust could contribute significantly to and in some regions even dominate
tropospheric $N_2O_5$ removal. Since uptake of $N_2O_5$ leads to the formation of nitrate, it can also
substantially modify chemical composition and physicochemical properties of mineral dust.
A global modelling study (Dentener and Crutzen, 1993) suggested that including
heterogeneous reaction of $N_2O_5$ with tropospheric aerosol particles with $\gamma(N_2O_5)$ equal to 0.1
could reduce modelled yearly average global NOx burden by 50%. It is found by other global
and regional modelling studies (Evans and Jacob, 2005; Chang et al., 2016) that modelled NOx
and $O_3$ concentrations agree better with observations if $\gamma(N_2O_5)$ parameterization based on new
laboratory results is adopted. In the study by Evans and Jacob (2005), $\gamma(N_2O_5)$ was set to be



0.01 for mineral dust, independent of RH. A recent modelling study (Macintyre and Evans,
2010) suggests that simulated NOx, $O_3$, and OH concentrations are very sensitive to the choice
of $\gamma(N_2O_5)$ in the range of 0.001-0.02, which significantly overlaps with the range of laboratory
measured $\gamma(N_2O_5)$ for mineral dust particles. Therefore, in order to better assess the impacts of
heterogeneous reaction of $N_2O_5$ with mineral dust on tropospheric oxidation capacity, $\gamma(N_2O_5)$
and its dependence on mineralogy and RH should be better understood.

Mineralogy and composition of mineral dust aerosol particles in the ambient air are

always more complex than those for dust samples used in laboratory studies. Measurements of
$NO_3$, $N_2O_5$, and other trace gases and aerosols in the troposphere enable steady-state $NO_3$ and
$N_2O_5$ lifetimes to be determined and $\gamma(N_2O_5)$ onto ambient aerosol particles to be derived
(Brown et al., 2006; Brown et al., 2009; Morgan et al., 2015; Phillips et al., 2016). It will be
very beneficial to investigate $N_2O_5$ uptake (and other reactive trace gases as well) by ambient
mineral dust aerosol. Recently such experimental apparatus, based on the aerosol flow tube
technique, has been developed and deployed to directly measure $\gamma(N_2O_5)$ onto ambient aerosol
particles (Bertram et al., 2009a; Bertram et al., 2009b). To our knowledge these measurements
have never been carried out in dust-impacted regions yet, though they will undoubtedly
improve our understanding of heterogeneous reaction of $N_2O_5$ with mineral dust in the
troposphere.



**Table 10:** Summary of previous laboratory studies on heterogeneous reactions of mineral dust with $NO_3$ radicals

| Dust | Reference | $T$ (K) | Concentration (molecule cm$^{-3}$) | Uptake coefficient | Techniques |
|---|---|---|---|---|---|
| Saharan dust | Karagulian and Rossi, 2005 | 298±2 | (0.7–4.0)×10$^{10}$ | $\gamma_0$ = 0.23±0.20 and $\gamma_{ss}$ = 0.12±0.08 when [$NO_3$]$_0$ = (7.0±1.0)×10$^{11}$ cm$^{-3}$; $\gamma_0$ = 0.16±0.05 and $\gamma_{ss}$ = 0.065±0.012 when [$NO_3$]$_0$ = (4.0±1.0)×10$^{12}$ cm$^{-3}$. | KC |
| | Tang et al., 2010 | 296±2 | (0.4–1.6)×10$^{10}$ | $\gamma(NO_3)/\gamma(N_2O_5)$ was reported to be 0.9±0.4, independent of RH (up to 70%). | CRDS |
| $CaCO_3$ | Karagulian and Rossi, 2005 | 298±2 | (0.4–3.8)×10$^{12}$ | $\gamma_0$ = 0.13±0.10 and $\gamma_{ss}$ = 0.067±0.040 when [$NO_3$]$_0$ = (7.0±1.0)×10$^{11}$ cm$^{-3}$; $\gamma_0$ = 0.14±0.05 and $\gamma_{ss}$ = 0.014±0.004 when [$NO_3$]$_0$ = (4.0±1.0)×10$^{12}$ cm$^{-3}$. | KC |
| kaolinite | Karagulian and Rossi, 2005 | 298±2 | (0.4–3.8)×10$^{12}$ | $\gamma_0$ = 0.11±0.08 and $\gamma_{ss}$ = 0.14±0.02 when [$NO_3$]$_0$ = (7.0±1.0)×10$^{11}$ cm$^{-3}$; $\gamma_0$ = 0.12±0.04 and $\gamma_{ss}$ = 0.065±0.012 when [$NO_3$]$_0$ = (4.0±1.0)×10$^{12}$ cm$^{-3}$. | KC |
| limestone | Karagulian and Rossi, 2005 | 298±2 | (0.4–3.8)×10$^{12}$ | $\gamma_0$ = 0.12±0.08 and $\gamma_{ss}$ = 0.034±0.016 when [$NO_3$]$_0$ = (7.0±1.0)×10$^{11}$ cm$^{-3}$; $\gamma_0$ = 0.20±0.07 and $\gamma_{ss}$ = 0.022±0.005 when [$NO_3$]$_0$ = (4.0±1.0)×10$^{12}$ cm$^{-3}$. | KC |
| ATD | Karagulian and Rossi, 2005 | 298±2 | (0.4–3.8)×10$^{12}$ | $\gamma_0$ = 0.2±0.1 and $\gamma_{ss}$ = 0.10±0.016 when [$NO_3$]$_0$ = (7.0±1.0)×10$^{11}$ cm$^{-3}$; $\gamma_0$ = 0.14±0.04 and $\gamma_{ss}$ = 0.025±0.007 when [$NO_3$]$_0$ = (4.0±1.0)×10$^{12}$ cm$^{-3}$. | KC |






### 3.6.2 NO$_3$ radicals

To our knowledge only two previous studies have explored heterogeneous uptake of NO$_3$ radicals by mineral dust particles. Heterogeneous reaction of NO$_3$ radicals with mineral dust was investigated for the first time at 298±2 K, using a Knudsen cell reactor (Karagulian and Rossi, 2005). Products observed in the gas phase include N$_2$O$_5$ (formed in the Eley-Rideal reaction of NO$_3$ with NO$_2$ on the dust surface) and HNO$_3$ (formed in the heterogeneous reaction of N$_2$O$_5$ and subsequently released into the gas phase) (Karagulian and Rossi, 2005). Surface deactivation occurred for all types of dust particles investigated. Dependence of uptake kinetics on the initial NO$_3$ concentration was observed (Karagulian and Rossi, 2005). When [NO$_3$]$_0$ was $(7.0\pm1.0)\times10^{11}$ cm$^{-3}$, the initial and steady-state uptake coefficients ($\gamma_0$ and $\gamma_{ss}$) were determined to be 0.13±0.10 and 0.067±0.040 for CaCO$_3$, 0.12±0.08 and 0.034±0.016 for natural limestone, 0.11±0.08 and 0.14±0.02 for kaolinite, 0.23±0.20 and 0.12±0.08 for Saharan dust, and 0.2±0.1 and 0.10±0.06 for ATD, respectively. When [NO$_3$]$_0$ was $(4.0\pm1.0)\times10^{12}$ cm$^{-3}$, $\gamma_0$ and $\gamma_{ss}$ were determined to be 0.14±0.05 and 0.014±0.004 for CaCO$_3$, 0.20±0.07 and 0.022±0.005 for natural limestone, 0.12±0.04 and 0.050±0.014 for kaolinite, 0.16±0.05 and 0.065±0.012 for Saharan dust, and 0.14±0.04 and 0.025±0.007 for ATD, respectively.

In the second study (Tang et al., 2010), a novel relative rate method was developed to investigate heterogeneous uptake of NO$_3$ and N$_2$O$_5$ by mineral dust. Changes in NO$_3$ and N$_2$O$_5$ concentrations due to reactions with dust particles (loaded on filters) were simultaneously detected by cavity ring-down spectroscopy. Experiments were carried out at room temperature (296±2 K) and at different RH up to 70%. $\gamma(\text{NO}_3)/\gamma(\text{N}_2\text{O}_5)$ was reported to be 0.9±0.4 for Saharan dust particles, independent of RH within the experimental uncertainties (Tang et al., 2010). In addition, even though very low levels of NO$_3$ and N$_2$O$_5$ (a few hundred pptv) were used, surface deactivation was still observed for both species (Tang et al., 2010).





With the reported $\gamma(NO_3)/\gamma(N_2O_5)$ ratio of 0.9 (Tang et al., 2010), $\gamma(NO_3)$ of 0.018 is
thus adopted to evaluate $\tau_{het}(NO_3)$ due to its heterogeneous uptake by mineral dust, based on
the $\gamma(N_2O_5)$ value of 0.02 (Section 3.6.1). Using Eq. (6), mineral dust mass concentrations of
10, 100, and 1000 $\mu m\ m^{-3}$ result in $\tau_{het}(NO_3)$ of ~9 h, ~52 min, and ~5 min, respectively. Field
measurements, as summarized in Table 1, suggest that tropospheric $NO_3$ lifetimes are typically
several minutes. Therefore, uptake by mineral dust is unlikely to be a significant sink for $NO_3$
in the troposphere, except for regions which are close to dust sources and thus heavily impacted
by dust storms. Similar conclusions were drawn by Tang et al. (2010a) who used an uptake
coefficient of 0.009 which is a factor of 2 smaller than the value used here. 3D GEOS-Chem
model simulations suggest that modelled $O_3$ appears to be insensitive to the choice of $\gamma(NO_3)$
in the range of 0.0001 to 0.1 (Mao et al., 2013b). To conclude, heterogeneous reaction with
mineral dust is not an important sink for tropospheric $NO_3$ radicals unless in regions with heavy
dust loadings.
**4. Summary and outlook**
It has been widely recognized that heterogeneous reactions with mineral dust particles
can significantly affect tropospheric oxidation capacity directly and indirectly. These reactions
can also change the composition of dust particles, thereby modifying their physicochemical
properties important for direct and indirect radiative forcing. In the past two decades there have
been a large number of laboratory (as well as field and modelling) studies which have examined
these reactions. In this paper we provide a comprehensive and timely review of laboratory
studies of heterogeneous reactions of mineral dust aerosol with OH, $NO_3$, and $O_3$ as well as
several other reactive species (including $HO_2$, $H_2O_2$, HCHO, HONO, and $N_2O_5$) which are
directly related to OH, $NO_3$, and $O_3$. Lifetimes of these species with respect to heterogeneous
uptake by mineral dust are compared to their lifetimes due to other major loss processes in the
troposphere in order to provide a quick assessment of atmospheric significance of




heterogeneous reactions as sinks for these species. In addition, representative field and
modelling work is also discussed to further illustrate the roles these heterogeneous reactions
play in tropospheric oxidation capacity. As shown in Section 3, these studies have significantly
improved our understanding of the effects of these reactions on tropospheric oxidation capacity.
Nevertheless, there are still a number of open questions which cannot be answered by
laboratory work alone but only by close collaboration among laboratory, field, and modelling
studies. Several major challenges, and strategies we proposed to address these challenges, are
outlined below.
1) Mineral dust in the troposphere are in fact mineralogically complex and its
mineralogy vary with dust sources and also residence time in the troposphere (Claquin et al.,
1999; Ta et al., 2003; Zhang et al., 2003; Nickovic et al., 2012; Journet et al., 2014; Scanza et
al., 2015). Different minerals can exhibit large variabilities in heterogeneous reactivity towards
trace gases, as shown by Tables 4-10. However, Tables 4-10 also reveal that simple oxides
(e.g., $SiO_2$ and $Al_2O_3$) and $CaCO_3$ have been much more widely investigated compared to
authentic dust samples (probably except ATD) and clay minerals which are the major
components of mineral dust aerosol particles (Claquin et al., 1999). The relative importance of
clay minerals will be increased after long-range transport due to their smaller sizes compared
to $SiO_2$ and $CaCO_3$. Therefore, more attention should be paid in future work to heterogeneous
reactions of clay minerals and authentic dust samples.
2) In the last several years, important roles that RH (and thus surface adsorbed water)
plays in heterogeneous reactions of mineral dust have been widely recognized by many studies
and discussed in a recent review paper (Rubasinghege and Grassian, 2013). Tables 4-10 show
that most of previous studies have been conducted at RH <80%, and heterogeneous reactivity
at higher RH largely remain unknown. In addition, effects of RH on heterogeneous reactions
of mineral dust with a few important reactive trace gases, such as $HO_2$ radicals (Bedjanian et



al., 2013b; Matthews et al., 2014) and $O_3$ (Sullivan et al., 2004; Chang et al., 2005; Mogili et
al., 2006a), are still under debate. It has been known that heterogeneous processing can modify
chemical composition and hygroscopicity of mineral dust particles (Tang et al., 2016a), and at
evaluated RH aged dust particles may consist of a solid core and an aqueous shell (Krueger et
al., 2003b; Laskin et al., 2005a; Liu et al., 2008b; Shi et al., 2008; Li and Shao, 2009; Ma et
al., 2012). Under such circumstances, reactions are no longer limited to particle surface but
instead involve gas, liquid, and solid phases and their interfaces, and hence mutual influence
among chemical reactivity, composition, and physiochemical properties has to be taken into
account (Tang et al., 2016a).

3) Temperature in the troposphere varies from <200 K to >300 K. However, most of

laboratory studies of heterogeneous reactions of mineral dust were carried out at room
temperature (around 296 K). Once lifted into the atmosphere, mineral dust aerosol is mainly
transported in the free troposphere in which temperature is much lower than that at the ground
level. Some work has started to examine the influence of temperature on heterogeneous uptake
by mineral dust (Michel et al., 2003; Xu et al., 2006; Xu et al., 2010; Wu et al., 2011; Xu et al.,
2011; Romanias et al., 2012b; Romanias et al., 2012a; Zhou et al., 2012; Bedjanian et al., 2013b;
El Zein et al., 2013a; El Zein et al., 2013b; Romanias et al., 2013; Wu et al., 2013b; El Zein et
al., 2014; Hou et al., 2016; Zhou et al., 2016). It has been found temperature may have
significant effects on some reactions. However, to the best of our knowledge, no study has
explored the influence of temperature on heterogeneous reactions of airborne mineral dust
particles.

4) Laboratory studies may not entirely mimic actual heterogeneous reactions in the

troposphere due to several reasons. First of all, laboratory studies are typically carried out with
time scales of <1 min to several hours, compared to lifetimes of a few days for mineral dust in
the troposphere. Secondly, it is not uncommon that concentrations of reactive trace gases used




in laboratory work are several orders of magnitude larger than those in the troposphere. These
two aspects can make it non-trivial to extrapolate laboratory results to the real atmosphere. In
addition, dust samples used in laboratory studies, even when authentic dust samples are used,
do not exactly mimic the complexity of ambient dust particles in composition and mineralogy.
Very recently a new type of experiments, sometimes called "laboratory work in the field", can
at least partly provide solutions to this challenge. For example, an aerosol flow tube has been
deployed to explore heterogeneous uptake of $N_2O_5$ by ambient aerosol particles at a few
locations (Bertram et al., 2009a; Bertram et al., 2009b; Ryder et al., 2014), revealing the roles
of RH and particle composition in heterogeneous reactivity of ambient aerosol particles. To
our knowledge, this technique has not been used to investigate heterogeneous uptake of $N_2O_5$
by ambient mineral dust aerosol. This technique can also be extended to examine
heterogeneous reactions of ambient aerosol particles with other reactive trace gases, especially
those whose heterogeneous reactions are anticipated to be efficient (e.g., $HO_2$ and $H_2O_2$).
5) Decrease in heterogeneous reactivity due to surface passivation has been observed
by many studies using dust powders supported by substrates. On the other hand, increase in
heterogeneous reactivity, due to conversion of solid particles to aqueous droplets with solid
cores (caused by formation of hygroscopic materials), has also been reported. In addition, it
has been widely recognized that the co-presence of two or more reactive trace gases may
change the rates of heterogeneous reactions of each individual gases (Li et al., 2006; Raff et
al., 2009; Liu et al., 2012; Rubasinghege and Grassian, 2012; Wu et al., 2013a; Zhao et al.,
2015; Yang et al., 2016a), typically termed as synergistic effects. Parameterization of these
complex processes is very difficult, and lack of sophisticated bulk parameterizations impedes
us from a quantitative assessment of their atmospheric significance via modelling studies.
Kinetic models have been developed to integrate physical and chemical processes in and
between different phases (Pöschl et al., 2007; Shiraiwa et al., 2012; Berkemeier et al., 2013),





and these models have been successfully used to investigate multiphase chemistry of aqueous

aerosol particles and cloud droplets (Shiraiwa et al., 2011; Arangio et al., 2015; Pöschl and

Shiraiwa, 2015). Future efforts devoted to development and application of comprehensive

kinetic models to study heterogeneous and multiphase reactions of mineral dust particles would

largely improve our understanding in the field.

6) It has been found that UV and visible radiation can substantially enhance the

heterogeneous reactivity of mineral dust towards several trace gases, including but not limited

to $H_2O_2$, $O_3$, and HCHO, and in some cases even reactivate mineral surfaces which have been

passivated (Cwiertny et al., 2008; Chen et al., 2012; George et al., 2015). In addition, photolysis

of materials (such as nitrate) formed on mineral surface can also be sources for some trace

gases (Nanayakkara et al., 2013; Gankanda and Grassian, 2014; Nanayakkara et al., 2014).

Although the effects of photo-radiation in heterogeneous reactions with mineral dust have been

recognized for more than one decade, it largely remains unclear to which extent these reactions

are photo-enhanced under ambient solar radiation and thus quantitative evaluation of impacts

of heterogeneous photochemistry on tropospheric oxidation capacity is lacking.

7) There still exists a considerably large gap between laboratory work and modelling

studies used to explain field measurements and predict future changes. One reason is that the

communication and collaboration between laboratory and modelling communities, though

enhanced in the past few decades, are still not enough and should be further encouraged and

stimulated in future. Furthermore, many laboratory studies have been designed from the

perspective of classical chemical kinetics such that although experimental results are beautiful,

they are difficult to be parameterized and then included in models. As mentioned,

heterogeneous reactivity is highly dependent on temperature, RH, co-presence of other trace

gases and mutual influences among these factors. Given that most models are capable of

resolving/assimilating meteorological variables and trace gas concentrations at high temporal




resolution, multivariate analysis and integrated numerical expressions are encouraged to be
conducted in laboratory studies so as to better characterize heterogeneous chemistry and its
climate and environmental effects in numerical models. Therefore, it is suggested that when a
laboratory study is designed, it should be kept in mind how experimental results can be used
by modelling studies. On the other hand, modelling work is encouraged to include new
laboratory results in numerical simulations and to identify missing reactions and key
parameters which deserve further laboratory investigation. Field campaigns which are
specifically designed to assess the impacts of mineral dust aerosol on tropospheric oxidation
capacity have been proved to be very beneficial (de Reus et al., 2000; Galy-Lacaux et al., 2001;
Seinfeld et al., 2004; Tang et al., 2004; de Reus et al., 2005; Umann et al., 2005; Arimoto et
al., 2006; Song et al., 2007), and more campaigns of this types should be organized. Overall,
as urged by a few recent articles (Kolb et al., 2010; Abbatt et al., 2014), the three-legged stool
approach (laboratory studies, field observations, and modelling studies) adopted by
atmospheric chemistry research for a long time should be emphasized, and mutual
communication and active collaboration among these three "legs" should be further enhanced.
**Acknowledgement**
The preparation of this manuscript was inspired by the first International Workshop on
Heterogeneous Atmospheric Chemistry (August 2015, Beijing, China) endorsed by the
International Global Atmospheric Chemistry (IGAC) Project, and Mingjin Tang and Tong Zhu
would like to thank all the participants for their valuable presentations and discussion. Financial
support provided by Chinese National Science Foundation (41675140 and 21522701), Chinese
Academy of Sciences international collaborative project (132744KYSB20160036), and State
Key Laboratory of Organic Geochemistry (SKLOGA201603A) is acknowledged. Mingjin
Tang is also sponsored by Chinese Academy of Sciences Pioneer Hundred Talents Program.




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
