# Peer review of "Heterogeneous reactions of mineral dust aerosol: implications for"

_Atmospheric Chemistry and Physics, 2017_

## Author Comment (AC1) · 4 Jun 2017

Two referees had some very helpful comments during the access review stage, and these comments will be addressed together with other comments raised later.

---

## Referee Comment (RC1) · Anonymous Referee #3 · 18 Jul 2017

This is a very interesting review focusing on the heterogeneous reactions of mineral dust aerosol with trace gases in the atmosphere. It presents a comprehensive and critical review of laboratory studies of heterogeneous uptake of OH, NO3, O3, and related species by mineral dust. The point of view which has been chosen here i.e., assessing the importance of the heterogeneous processes by comparing the associated lifetimes with other major loss processes and by discussing relevant field and modelling studies is very interesting and brings real added-value to already published reviews.

I really enjoyed reading this manuscript and would therefore recommend its publication in Atmospheric Chemistry and Physics. I have only one comment on which I would like

to draw the authors' attention to.

As one of the major target of that review is to derive lifetimes associated to heterogeneous processes, which is a valuable information, I would encourage the authors to put more emphasis on the needed/missing input information and more specifically on the need to use uptake coefficient derived under steady state conditions. Clearly this point is already more or less addressed in this review but without being properly emphasized. I believe that devoting a full paragraph to this issue in section 1.2 ("Introduction to heterogeneous kinetics") would be the way to go, and then for every targeted compound to highlight what is known and unknown with respect to long exposure times (i.e., steady state conditions), as initial uptake coefficients should not be used to derive atmospheric lifetimes for mineral dusts.

Also, when the authors derive these lifetimes, they fix one gas phase concentration and then do the calculations. However, for many of the processes discussed here, the lifetime will change with concentration and therefore the associated lifetimes will be spatially different. For instance, an uptake process could be slow at high concentration (i.e., at ground level) but significantly faster under reduced concentration (i.e., at higher altitude). Maybe the authors could bear that in mind when assessing lifetimes for some of the compounds, and especially ozone.

Minor points

The introduction, and justification of that review, is maybe a bit lengthy and could be reduced without loss of information.

The simplified figure 1 is still somewhat difficult to follow.

Why plotting the uptake coefficient versus the gas phase concentration on log-log scale (e.g., figure 13)? Is there a justifications for that? There are different ways of linearizing the adsorption isotherms and extract meaningful information.
* * *
2017.

---

## Referee Comment (RC2) · Anonymous Referee #4 · 4 Aug 2017

The authors summarized heterogeneous reactions of mineral aerosols and emphasized its implications for oxidation capacity in the troposphere on the basis of substantial publications. Generally, this is an interesting topic although a lot of review articles in this field have been published, followed by the first work reported by Usher et al. (2003). Especially, the authors tried to compare heterogeneous uptake lifetime of oxidative species (O3, H2O2, HONO, HCHO, and N2O5) to ones by other loss pathways in the atmosphere, which is the valuable information to the researchers. Finally, the authors supposed mineralogy of dusts, RH, temperature could play the important roles in the heterogeneous process, and recommended that simulated experiments should be performed under more actual conditions.

[Figure]

Specifically, the manuscript suffered from some small flaws: (1) As a review-type article, it's better if the authors supply time span of the literatures, since many review paper have been published in this field.

(2) In the fraction of "1.1 Mineral dust in the atmosphere", I found it is little relationship to oxidation capitation in the troposphere.

(3) The authors should list a total table to compare the loss lifetime of the key species by the heterogeneous process and gas-phase process.

(4) Although the paper was well organized and written, I still found some English errors, such as: Line 41 "in the atmospheres" , line 80 "and etc", line 222 ". . .in reporting and interpreting kinetic data", line 247 ". . .the first major primary source", line 1742 "the roles these heterogeneous reactions play in. . .".

---

## Author Comment (AC2) · 18 Aug 2017

Comments by Referees are in blue. Our replies are in black. Changes to the manuscript are highlighted in red both in here and in the revised manuscript.

**Reply to Ref #3**

This is a very interesting review focusing on the heterogeneous reactions of mineral dust aerosol with trace gases in the atmosphere. It presents a comprehensive and critical review of laboratory studies of heterogeneous uptake of OH, NO3, O3, and related species by mineral dust. The point of view which has been chosen here i.e., assessing the importance of the heterogeneous processes by comparing the associated lifetimes with other major loss processes and by discussing relevant field and modelling studies is very interesting and brings real added-value to already published reviews.

I really enjoyed reading this manuscript and would therefore recommend its publication in Atmospheric Chemistry and Physics. I have only one comment on which I would like to draw the authors' attention to.

**Reply:** We would like to thank ref #3 for his/her highly positive comments on our manuscript. All the comments have been properly addressed in the revised manuscript, as detailed below.

As one of the major target of that review is to derive lifetimes associated to heterogeneous processes, which is a valuable information, I would encourage the authors to put more emphasis on the needed/missing input information and more specifically on the need to use uptake coefficient derived under steady state conditions. Clearly this point is already more or less addressed in this review but without being properly emphasized. I believe that devoting a full paragraph to this issue in section 1.2 ("Introduction to heterogeneous kinetics") would be the way to go, and then for every targeted compound to highlight what is known and unknown with respect to long exposure times (i.e., steady state conditions), as initial uptake coefficients should not be used to derive atmospheric lifetimes for mineral dusts.

**Reply:** It is true that initial uptake coefficients should not be used to calculate atmospheric lifetimes due to heterogeneous uptake onto mineral dust. Nevertheless, as discussed in Section 2.2.2, steady-state uptake coefficients reported by laboratory studies depend largely on experimental conditions, such as trace gas concentrations and the mass of particle samples. Therefore, with the knowledge available up to now, we feel it very difficult to have definite

answers to the important question raised by ref #3.

Also, when the authors derive these lifetimes, they fix one gas phase concentration and then do the calculations. However, for many of the processes discussed here, the lifetime will change with concentration and therefore the associated lifetimes will be spatially different. For instance, an uptake process could be slow at high concentration (i.e., at ground level) but significantly faster under reduced concentration (i.e., at higher altitude). Maybe the authors could bear that in mind when assessing lifetimes for some of the compounds, and especially ozone.

**Reply:** We absolutely agree with ref #3 that a single uptake coefficient is not enough to describe the kinetics of a heterogeneous reaction. In the revised manuscript (line 643-649), we have included the following sentence to mention this caveat: "We also acknowledge that a single uptake coefficient may not always be enough to describe the kinetics of a heterogeneous reaction of mineral dust, because 1) uptake kinetics may change with reaction time, as discussed in Section 2.2; 2) uptake kinetics are also affected by particle mineralogy and composition, RH, temperature, the co-presence of other reactive trace gases, and etc.; and 3) for some reactive trace gases, such as $O_3$, the uptake coefficients may strongly depend on their concentrations."

Minor points

The introduction, and justification of that review, is maybe a bit lengthy and could be reduced without loss of information.

**Reply:** Indeed the introduction section in our manuscript is quite long. Nevertheless, since we aim to provide a comprehensive review of this topic, we feel it is necessary to try our best to mention all the aspects in the introduction. In addition, because in the last two decades there have been a few excellent reviews in this field, we would like to emphasize why writing our current review paper is justified and what distinguish our review paper from previous ones.

The simplified figure 1 is still somewhat difficult to follow.

**Reply:** As suggested, we have simplified Figure 1 in the revised manuscript to make it easy to follow.

Why plotting the uptake coefficient versus the gas phase concentration on log-log scale (e.g., figure 13)? Is there a justifications for that? There are different ways of linearizing the adsorption isotherms and extract meaningful information.

**Reply:** This is because both $O_3$ concentrations and $\gamma(O_3)$ shown in this figure (and Figure 12 as well) span over a few orders of magnitude. In the revised manuscript (line 1244-1245) we have added one sentence to explain why we use log-log scale: "Both $O_3$ concentrations and $\gamma(O_3)$ are plotted on the logarithm scale because their values span over a few orders of magnitude. "

---

## Author Comment (AC3) · 18 Aug 2017

Comments by Referees are in blue. Our replies are in black. Changes to the manuscript are highlighted in red both in here and in the revised manuscript.

**Reply to Ref #4**

The authors summarized heterogeneous reactions of mineral aerosols and emphasized its implications for oxidation capacity in the troposphere on the basis of substantial publications. Generally, this is an interesting topic although a lot of review articles in this field have been published, followed by the first work reported by Usher et al. (2003). Especially, the authors tried to compare heterogeneous uptake lifetime of oxidative species (O3, H2O2, HONO, HCHO, and N2O5) to ones by other loss pathways in the atmosphere, which is the valuable information to the researchers. Finally, the authors supposed mineralogy of dusts, RH, temperature could play the important roles in the heterogeneous process, and recommended that simulated experiments should be performed under more actual conditions.

**Reply:** We would like to thank ref #4 very much for recommending our manuscript for publication. His/her comments, which largely helped us improve our manuscript, have been properly addressed in our revised manuscript, as detailed below.

Specifically, the manuscript suffered from some small flaws: (1) As a review-type article, it's better if the authors supply time span of the literatures, since many review paper have been published in this field.

**Reply:** That is a good point. In the revised manuscript (line 235-236) we have clarified the literature covered in this review paper: "Following this in Section 3, we review previous laboratory studies of heterogeneous reactions of mineral dust particles with these eight reactive trace gases, and we have tried our best to cover all the journal articles (limited to those in English) published in this field."

(2) In the fraction of "1.1 Mineral dust in the atmosphere", I found it is little relationship to oxidation capitation in the troposphere.

**Reply:** The topic of our manuscript is heterogeneous reactions of mineral dust and its implications for tropospheric oxidation capacity; therefore, it is necessary to give an introduction to tropospheric mineral dust aerosol and its environmental and climatic impacts. Nevertheless, we can understand that ref #4 found it not very relevant to tropospheric oxidation capacity, and this is largely because in the original manuscript the impact of mineral dust on

tropospheric oxidation capacity was not emphasized. In the revised manuscript (line 98-101) we have highlighted it by referring to the pioneering work by Dentener et al. (1996):"According to this study, heterogeneous reactions with mineral dust could largely impact tropospheric photochemical oxidation cycles, resulting in up to 10% decreases in $O_3$ concentrations in dust source regions and nearby."

(3) The authors should list a total table to compare the loss lifetime of the key species by the heterogeneous process and gas-phase process.

**Reply:** We agree with ref #4 that it could be very informative to provide a table which summarizes lifetimes of key species with respect to gas phase and heterogeneous reactions. Such information has been provided in relevant sections/subsections in our manuscript. However, lifetimes of these species due to gas phase and heterogeneous loss processes are highly spatially and temporally variable, depends on the concentrations of other species they react with; therefore, it is very difficult to use a table to summarize such information in a comprehensive way. If such as a table is not comprehensive, it may cause overs-simplification and thus can be misleading. As a result, we prefer not to provide such a table at this moment, though we entirely agree that such a table, if presented in a comprehensive manner, would be very useful.

(4) Although the paper was well organized and written, I still found some English errors, such as: Line 41 "in the atmospheres", line 80 "and etc", line 222 ". . .in reporting and interpreting kinetic data", line 247 ". . .the first major primary source", line 1742 "the roles these heterogeneous reactions play in. . .".

**Reply:** We thank ref #4 for carefully reading our manuscript. All the typos have been corrected in our revised manuscript.

---

## Author Comment (AC4) · 18 Aug 2017

Comments by Referees are in blue. Our replies are in black. Changes to the manuscript are highlighted in red both in here and in the revised manuscript.

**Reply to Ref #1**

I have a few comments at this stage:

1. The authors need to have some introduction on dust and its composition. So the reader can understand why those laboratory work are discussed on different types of aerosol surfaces and how they are relevant to dust.

**Reply:** As suggested, in the revised manuscript (line 75-77) we have provided additional information on mineralogy of tropospheric mineral dust aerosols: "According to a recent global modeling study (Scanza et al., 2015), major minerals contained by tropospheric mineral dust particles include quartz, illite, montmorillonite, feldspar, kaolinite, calcite, hematite, and gypsum."

2. One major issue not discussed here, is the coating of mineral dust. It is well known that mineral dust may be coated by inorganic and organic acids during its lifetime in atmosphere. Consequently, this will involve aqueous chemistry in that liquid layer, particularly with dissolved substance from mineral dust. I don't think this can be reproduced from laboratory measurements, but it is something happening in the atmosphere and may cause a huge difference on gamma from those laboratory measurements. This should be discussed in the text.

**Reply:** We agree with ref #1 that mineral dust particles in the troposphere can be much more complex than those used in laboratory studies. In the revised manuscript (line 461-469) we have discuss this aspect: "In addition to these two important issues, it should also be mentioned that single minerals (e.g., illite, calcite, and quartz) and authentic dust samples (e.g., Saharan dust and Arizona test dust) may not necessarily reflect mineral dust particles found in the troposphere. After emitted into the troposphere, mineral dust particle will undergo heterogeneous reactions and cloud processing (Usher et al., 2003a; Tang et al., 2016a), forming soluble inorganic and organic materials coated on dust particles (Sullivan et al., 2007; Sullivan and Prather, 2007; Formenti et al., 2011; Fitzgerald et al., 2015). Therefore, heterogeneous reactivity of ambient mineral dust particles can be largely different from those used in laboratory studies."

3. Line 277-281, it should be pointed out that OH-regeneration could be also partly due to instrumental interference, see this paper for example (http://www.atmos-chem-phys.net/12/8009/2012/acp-12-8009-2012.html).

**Reply:** We agree. In the revised manuscript (line 294-300) we have added one sentence to mention the OH measurement interference and its implication: "Nevertheless, in a recent study (Mao et al., 2012), the proposed new OH regeneration mechanism is thought to be at least partly caused by unrecognized instrumental interference in OH measurements (Mao et al., 2012). A community effort is now started to assure the data quality of the OH measurement under different conditions especially for the chemical complex areas (http://www.fz-juelich.de/iek/iek8/EN/AboutUs/Projects/HOxROxWorkingGroup/HOxWorkshop2015_node.html)."

4. Figure 3 is a bit confusing. What is shown is actually not decay but rather increase. This can be better presented.

**Reply:** This figure shows the measured concentration of X. Therefore, surface deactivation would result in reduced loss of X due to heterogeneous uptake and thus increase in measured [X]. In the revised manuscript (line 578-585), we have revised and expanded the figure caption to better explain this figure, and we have also slightly modified this figure accordingly.

5. Line 722, the work from de Reus et al. 2005 suggests that the product of HO2 uptake on dust aerosols are likely H2O instead of H2O2. This should be added to the text.

**Reply:** The referee is right. In the revised manuscript (line 843-848) we have added a few sentences to discuss the work by de Reus et al. (2005): "In the modeling work carried out by de Reus et al. (2005), $\gamma(HO_2)$ was assumed to be 0.2 for heterogeneous uptake onto Saharan dust particles. If no $H_2O_2$ is formed in heterogeneous reaction of $HO_2$ with Saharan dust, modeled $H_2O_2$ concentrations would agree well with measurements; in contrast, if heterogeneous uptake of $HO_2$ radicals were assumed to produce $H_2O_2$, modeled $H_2O_2$ concentrations would be much larger than measured values."

---

## Author Comment (AC5) · 18 Aug 2017

Comments by Referees are in blue. Our replies are in black. Changes to the manuscript are highlighted in red both in here and in the revised manuscript.

**Reply to Ref #2**

This review on heterogeneous reactions of mineral dust aerosols (desert dusts, SiO2, Al2O3, TiO2) is very well written, useful for atmospheric chemists and recommended to be published after some revisions commented below.

**Reply:** We would like to thank ref #2 very much for recommending our manuscript for publication. His/her comments, which largely helped us improve our manuscript, have been properly addressed in our revised manuscript, as detailed below.

1. In this manuscript, logical uncertainty resides around the experimentally obtained values of uptake coefficients of benchmark sample particles based on assumed surface areas, and uptake coefficients of real ambient dust particles to be used for the calculation of their heterogeneous reaction rates and lifetimes in the atmosphere. Experimentally obtained values of γ depends on the assumed surface area, and those based on BET area and geometrical area are in general different in more than a couple of orders of magnitude as reported in this review. It is recommended to clarify the idea which area is more relevant to the heterogeneous reaction on the ambient aerosol particles. How is the real surface area of ambient dust particles, and how to calculate the best heterogeneous reaction rates based on reported γ values? In p.23 (line 486-487), although it is described "The surface area actually available for heterogeneous uptake falls between two extreme cases and varies for different studies," this description is not enough to give a solid idea to readers.

**Reply:** As pointed out by ref #2 as well as in our initial manuscript, uptake coefficients based on geometrical area and the BET area can differ by a few orders of magnitude. Nevertheless, up to now there is no universally accepted method to estimate the surface area actually available for heterogeneous uptake, and the statement "The surface area actually available for heterogeneous uptake falls between two extreme cases and varies for different studies" is the only one we can make.

In fact in our initial manuscript we discussed a few methods (such as the KML model and the LMD model) which have been proposed in order to better estimate the surface area available for heterogeneous uptake. In the revised manuscript, as suggested by ref #2, we have

expanded our discussion on the KML model, the LMD model, and etc.

It is suggested to add a new section on "characteristics of ambient mineral dust particles" for example at the beginning of "2 Background" to describe; 1) their chemical characteristics such as relevance of desert dust and benchmark minerals, SiO2, Al2O3, and TiO2, and 2) their physical characteristics such as surface area, porosity, etc. which are relevant to estimate the heterogeneous reaction rates. Quantitative descriptions on these topics are very useful for readers.

**Reply:** In the revised manuscript (line 75-77) we have provided additional information on mineralogy of tropospheric mineral dust aerosols: "According to a recent global modeling study (Scanza et al., 2015), major minerals contained by tropospheric mineral dust particles include quartz, illite, montmorillonite, feldspar, kaolinite, calcite, hematite, and gypsum."

In addition, we have also provided further information on physical characteristics of mineral dust particles. For examples, in line 77-80 of the revised manuscript, we have included the following sentence for particle size: "Formenti et al. (2011) summarized published measurements of tropospheric mineral dust particles, and the size of mineral dust particles depends dust sources and transport, with typical volume median diameters being a few micrometers or larger." In line 495-497 of the revised manuscript, we have included the following sentence for particle shape: "It has been reported that the median aspect ratios are in the range of 1.6-17 for Saharan dust particles (Chou et al., 2008; Kandler et al., 2009) and 1.4-1.5 for Asian dust particles (Okada et al., 2001)."

Some detailed description of the methodology to obtain surface area of dust particles, how to obtain geometrical area of non-spherical particles and BET area, would be helpful.

**Reply:** In line 512-513 of the revised manuscript, we have expanded the sentence to make it more clear how to obtain geometrical area of non-spherical particles: "In these experiments the surface area available for heterogeneous uptake is assumed to be either the projected area of dust particles (usually referred to the geometrical area of dust particles, equal to the geometrical surface area of the sample holder) or the BET surface area of the dust sample."

BET surface area measurements are widely used in characterization of solid particles. Instead of providing a detailed description on how to measure BET surface area, in line 514-515 of the revised manuscript, we have included a sentence to refer interested readers to proper

references: "Description of methods used in measuring BET surface area of solid particles can be found elsewhere (Sing, 2014; Naderi, 2015)."

Also some detailed description on transport of gaseous molecules within interior space of the powdered sample, and KML model would be helpful.

**Reply:** As suggested, in the revised manuscript (line 520-525) we have provided a description on reaction and transport of gas molecules within powdered samples: "When gas molecules are transported towards the top layer of the powdered sample, they may collide with the surface of particles on the top layer, be adsorbed, and undergo heterogeneous reaction; they may also be transported within the interior space and then collide and react with particles in the underlying layers. The depth gas molecules can reach depends on the microstructure of the powdered sample (e.g., how compactly particles are stacked) as well as their reactivity towards the surface."

Furthermore, in line 536-538 of the revised manuscript, we have included another sentence to further introduce the concept of the KML model: "An "effectiveness factor" was determined and used in the KML model to account for the contribution of underlying layers to the observed heterogeneous uptake."

2. It is interesting to compare the uptake coefficients of dust with more commonly available soil and ambient aerosol. Although some description is available for HCHO in p. 72 (line 1309-1310), further comparison of γ for HO2 by ambient common aerosols, and others should be described whenever the data are available.

**Reply:** Studies on heterogeneous reactions of reactive trace gases with ambient mineral dust aerosol are very limited, and most of them (if not all) have been discussed in our manuscript. There are many studies on heterogeneous reactions of reactive trace gases (mainly $HO_2$, $H_2O_2$ and $N_2O_5$) with other types of ambient aerosol particles, but their chemical composition and thus heterogeneous reactivity may be very different from these of mineral dust; therefore, such comparison may not be proper.

In addition to heterogeneous reaction of HCHO with soil, in our original manuscript (line 1403-1408) we have also discussed heterogeneous uptake of HONO onto soil samples.

A very recent study (Moon et al., 2017) examined heterogeneous uptake of $HO_2$ radicals by $TiO_2$ particles. In the revised manuscript (line 784-780), the study by Moon et al. have been

discussed, and Table 4 has also been expanded to include the major result reported by Moon et al. (2017).

3. p. 47 (line 889): Specific wavelengths should be described for "UV illumination".

**Reply:** The wavelength range is 315-400 nm, and this information has been included in the revised manuscript.

---

## Author Comment (AC6) · 18 Aug 2017

Please find our revised manuscript in the supplement.

Please also note the supplement to this comment:
https://www.atmos-chem-phys-discuss.net/acp-2017-458/acp-2017-458-AC6-supplement.pdf
* * *